# QUANTUM MACHINE LEARNING ADVANTAGES BEYOND HARDNESS OF EVALUATION

**Riccardo Molteni**[*,1,2]     **Simon C. Marshall**[1,2]     **Vedran Dunjko**[1,2]

[1] $\langle aQa^L \rangle$ Applied Quantum Algorithms, Universiteit Leiden
[2] LIACS, Universiteit Leiden, Niels Bohrweg 1, 2333 CA Leiden, Netherlands
[*] r.molteni@liacs.leidenuniv.nl

## ABSTRACT

Recent years have seen rigorous proofs of quantum advantages in machine learning, particularly when data is labeled by cryptographic or inherently quantum functions. These results typically rely on the infeasibility of classical polynomial-sized circuits to evaluate the true labeling function. While broad in scope, these results however reveal little about advantages stemming from the actual learning process itself. This motivates the study of the so-called identification task, where the goal is to "just" identify the labeling function behind a dataset, making the learning step the only possible source of advantage. The identification task also has natural applications, which we discuss. Yet, such identification advantages remain poorly understood. So far they have only been proven in cryptographic settings by leveraging random-generatability, the ability to efficiently generate labeled data. However, for quantum functions this property is conjectured not to hold, leaving identification advantages unexplored. In this work, we provide the first proofs of identification learning advantages for quantum functions under complexity-theoretic assumptions. Our main result relies on a new proof strategy, allowing us to show that for a broad class of quantum identification tasks there exists an exponential quantum advantage unless BQP is in a low level of the polynomial hierarchy. Along the way we prove a number of more technical results including the aforementioned conjecture that quantum functions are not random generatable (subject to plausible complexity-theoretic assumptions), which shows a new proof strategy was necessary. These findings suggest that for many quantum-related learning tasks, the entire learning process—not just final evaluation—gains significant advantages from quantum computation.

## 1  INTRODUCTION

A central question in quantum machine learning (QML) is understanding which learning problems inherently require a quantum computer for efficient solutions. As shown in (Huang et al., 2021), access to data enables classical learners to efficiently evaluate some hard-to-compute functions that are otherwise intractable for polynomial time classical randomized algorithms. Most of the known examples of learning problems where data does not aid in computing the target function involve cryptographic functions, for which random labeled samples can be generated efficiently using classical algorithms. While the proofs may be more involved, the intuition remains simple: if a classical machine could generate the data, then the data itself cannot be what makes a hard problem easy. In the context of QML, where the greatest advantages are intuitively expected for "fully quantum" functions (i.e., functions that are BQP-complete[1]), the focus on cryptographic tasks was somewhat unsatisfactory. This limitation was addressed in (Gyurik & Dunjko, 2023) by employing stronger computational complexity assumptions and, crucially, by leveraging the fact that standard learning definitions require the learned model to *evaluate* new data points.

However, a different learning condition could be considered where the learning algorithm is solely required to *identify* the target labeling function from a set of labeled data. In other words, we

---

[1]More precisely, the class BQP does not contain complete problems. Throughout this paper, whenever we refer to complete problems, we will instead consider the class PromiseBQP.

consider supervised learning problems where the learner receives labeled examples $(x, f(x))$ and it is required to output a description of a consistent $f$, up to PAC error. In this scenario, the classical learner would only need to recover the description of the unknown target function, without needing to also evaluate it on new input points. From a fundamental standpoint, analyzing the hardness of the identification task in learning problems involving quantum functions helps clarify the source of learning separations. From a practical viewpoint, particularly in expected applications of quantum machine learning, identifying the data-generating function can in fact be the primary goal, such as in Hamiltonian learning or in tasks related to finding order parameters (Gyurik & Dunjko, 2023), as we will comment in Appendix G. Crucially, as was proved in (Gyurik & Dunjko, 2023), separations for the identification problem cannot exist without assumptions on the hypothesis class (intuitively, the classical learner can always outputs the circuit for the whole quantum learner with hardwired data as the output). The main goal of this paper is then to determine the conditions under which a learning separation for fully quantum functions can already emerge from the identification step.

Classical hardness of the identification problem has already been established for certain learning tasks, such as the learnability of DNF formulas (Kearns & Vazirani, 1994) and cryptographic functions (Gyurik & Dunjko, 2023; Jerbi et al., 2024). The key components in these hardness proofs are: $(1)$ the existence of a succinct representation of the target function that enables efficient extraction of the relevant property, and $(2)$ the property of "random generatability". In this paper, we use the term "random generatability" to refer to the ability to efficiently generate labeled samples $(x, f(x))$ for random inputs $x$. We also refer to the characteristic functions of BQP languages as "quantum functions" (informally, these are functions whose values can be estimated efficiently by a quantum circuit, but not by any classical algorithm, see Def. 10). In Appendix G we list a series of practical scenarios where quantum functions naturally arise. Crucially, for learning tasks involving quantum functions, we cannot rely on a simple representation of the target function, and we also show that these functions are not randomly generatable. Consequently, our hardness result for the identification task requires a fundamentally different proof strategy from those used in prior works. We discuss this in more detail in Appendix A. The main contributions of the paper are the following.

1. We show that quantum functions are not random generatable unless BQP is in the second level of the polynomial hierarchy. This result has consequences on quantum generative modelling as well, see Corollary 2.

2. We introduce the task of verifiable-identification, where the algorithm must reject invalid datasets, and solve identification correctly otherwise, and prove classical verifiable-identification for quantum target function is impossible, unless BQP is in $\mathsf{BPP}^{\mathsf{NP}}$.

3. We identify sufficient conditions on learning algorithm and concept class such that "mere" classical identification implies approximate-verifiable identification (the algorithm must reject datasets unless most of the samples are labeled according to the same concept) within the polynomial hierarchy. This implies identification is classically intractable unless BQP is in $\mathsf{BPP}^{\mathsf{NP}^{\mathsf{NP}^{\mathsf{NP}}}}$ (two levels "up" in the hierarchy compared to the verifiable case in the point above).

4. We provide examples of physically motivated learning tasks that satisfy the above condition and for which there exists a quantum learning algorithm that solves the identification task, thereby yielding a formal quantum–classical separation (see Corollary 1).

The relationship between our proofs is illustrated in Figure 1 in Section 5. We note that the results in the main text are presented under the heuristic form of the complexity-theoretic assumptions, since our analysis follows a distribution-specific scenario. In Appendix C we describe how classical hardness can be obtained under the exact assumption $\mathsf{BQP} \not\subseteq \mathsf{BPP}^{\mathsf{NP}^{\mathsf{NP}^{\mathsf{NP}}}}$ in the case learnability is required under every distribution.

## 2 PRELIMINARIES

We now provide here the precise definitions of the two tasks we are addressing in this work. Appendix B provides the definitions of quantum functions and the complexity-theoretic background relevant to our hardness results, along with a glossary of the most frequently used symbols in Table 1.

## 2.1 DEFINITION OF RANDOM GENERATABILITY

Given a uniform family of functions $f = \{f_n\}_{n \in \mathbb{N}}$, with $f_n : \{0,1\}^n \to \{0,1\}$, we say that $f$ is "random-generable" under a distribution $\mathcal{D}$ if there exists a uniform (randomized) algorithm capable of producing samples $(x, f_n(x))$ with $x$ sampled from $\mathcal{D}$ efficiently for any $n$. The concept of functions that permit an efficient procedure to generate random samples $(x, f(x))$ was introduced in (Arrighi & Salvail, 2006) under the term "random verifiability". In this paper, we refer to the same property as "random generatability" to avoid potential confusion with the "verifiable case" of the identification task described in Def. 7. Specifically, we consider two cases: exact random generatability in Def. 1, where the algorithm is not allowed to make any errors, and approximate random generatability in Def. 2, where the algorithm outputs samples from a distribution only close to the true target distribution.

**Definition 1** (Exact random generatability). Let $f = \{f_n\}_{n \in \mathbb{N}}$ be a uniform family of functions[2], with $f_n : \{0,1\}^n \to \{0,1\}$. $f$ is exact random generatable under the input distribution $\mathcal{D}$ if there exists an efficient uniform (randomized) algorithm $\mathcal{A}_{\mathcal{D}}$ such that given as input a description of $f_n$ for any $n$ and a random string $r$ is such that with probability 1:

$$\mathcal{A}_{\mathcal{D}}(f_n, r) = (x_r, f_n(x_r)). \tag{1}$$

When $r$ is chosen uniformly at random from the set of all random strings with length polynomial in $n$, then $(x_r, f_n(x_r)) \sim \pi^f(\mathcal{D})$, where $\pi^f(\mathcal{D})$ samples $x$ from the target distribution $\mathcal{D}$ over $x \in \{0,1\}^n$ and assigns the corresponding label $f_n(x_r) \in \{0,1\}$.

**Definition 2** (Approximate random generatability). Let $f = \{f_n\}_{n \in \mathbb{N}}$ be a uniform family of functions, with $f_n : \{0,1\}^n \to \{0,1\}$. $f$ is approximately random generatable under the distribution $\mathcal{D}$ if there exists an efficient uniform (randomized) algorithm $\mathcal{A}_{\mathcal{D}}$ such that given as input a description of $f_n$ for any $n$, a random string $r$ and an error value $\epsilon$ outputs:

$$\mathcal{A}_{\mathcal{D}}(f_n, 0^{1/\epsilon}, r) = (x_r, f_n(x_r)) \tag{2}$$

with $(x_r, f_n(x_r)) \sim \pi_\epsilon^f(\mathcal{D})$ when $r$ is sampled uniformly at random from the distribution of all the random strings (with polynomial size). In particular, $\pi_\epsilon^f(\mathcal{D})$ is a probability distribution over $\{0,1\}^n \times \{0,1\}$ which satisfies: $\|\pi_\epsilon^f(\mathcal{D}) - \pi^f(\mathcal{D})\|_{TV} \leq \epsilon$, where $\pi^f(\mathcal{D})$ is the same distribution defined in Def. 1.

As the total variation distance between two distributions $p$ and $q$ over $x \in \{0,1\}^n$ is defined as $\|p - q\|_{TV} = \frac{1}{2} \sum_x |p(x) - q(x)|$, the algorithm $\mathcal{A}$ in Def. 2 is allowed to make mistakes both with respect to the probability distribution of the outputted $x$s (e.g. they may not be generated from the uniform distribution) and with respect to the assigned labels.

## 2.2 PAC FRAMEWORK

The results in this paper are expressed in the standard terms of the efficient *probably approximately correct* (PAC) learning framework (Kearns & Vazirani, 1994; Mohri, 2018). In the case of supervised learning, a learning problem in the PAC framework is defined by a *concept class* $\mathcal{F} = \{\mathcal{F}_n\}_{n \in \mathbb{N}}$, where each $\mathcal{F}_n$ is a set of *concepts*, which are functions from some *input space* $\mathcal{X}_n$ (in this paper we assume $\mathcal{X}_n$ is a subset of $\{0,1\}^n$) to some *label set* $\mathcal{Y}_n$ (in this paper we assume $\mathcal{Y}_n$ is $\{0,1\}$, unless stated otherwise). The learning algorithm receives on input samples of the target concepts $T = \{(x_\ell, f(x_\ell))\}_\ell$, where $x_\ell \in \mathcal{X}_n$ are drawn according to *target distributions* $\mathcal{D} = \{\mathcal{D}_n\}_{n \in \mathbb{N}}$. Finally, the learning algorithm has a hypothesis class $\mathcal{H} = \{\mathcal{H}_n\}_{n \in \mathbb{N}}$ associated to it, and the learning algorithm should output a *hypothesis* $h \in \mathcal{H}_n$ – which is another function from $\mathcal{X}_n$ to $\mathcal{Y}_n$– that is in some sense "close" (see Eq. (3) below) to the concept $f \in \mathcal{F}_n$ generating the samples in $T$.

**Definition 3** (Efficient probably approximately correct learnability). A concept class $\mathcal{F} = \{\mathcal{F}_n\}_{n \in \mathbb{N}}$ is *efficiently PAC learnable* under target distributions $\mathcal{D} = \{\mathcal{D}_n\}_{n \in \mathbb{N}}$ if there exists a hypothesis class $\mathcal{H} = \{\mathcal{H}_n\}_{n \in \mathbb{N}}$ and a (randomized) learning algorithm $\mathcal{L}$ with the following property: for every $f \in \mathcal{F}_n$, and for all $0 < \epsilon < 1/2$ and $0 < \delta < 1/2$, if $\mathcal{L}$ receives in input a training set $T$ of size greater than $M \in \mathcal{O}(\text{poly}(n))$, then with probability at least $1 - \delta$ over the random samples in $T$

---

[2]Committing slight abuse of notation, we will use $f_n$ to denote both the function itself and its description.

and over the internal randomization of $\mathcal{L}$, the learning algorithm $\mathcal{L}$ outputs a specification[3] of some $h \in \mathcal{H}_n$ that satisfies:

$$\Pr_{x \sim \mathcal{D}_n}\left[h(x) \neq f(x)\right] \leq \epsilon. \tag{3}$$

Moreover, the learning algorithm $\mathcal{L}$ must run in time $\mathcal{O}(\mathrm{poly}(n, 1/\epsilon, 1/\delta))$. If the learning algorithm runs in classical (or quantum) polynomial time, and the hypothesis functions can be evaluated efficiently on a classical (or quantum) computer, we refer to the concept class as *classically learnable* (or *quantumly learnable*, respectively).

In classical machine learning literature, hypothesis functions are generally assumed to be efficiently evaluable by classical algorithms, as the primary goal is to accurately label new data points. In this paper, however, we focus on learning separations that arise from the inability of a classical algorithm to identify the target concept that labels the data, rather than from the hardness of evaluating the concept itself. Therefore, for our purposes we consider hypothesis functions that may be computationally hard to evaluate classically. Specifically, we will consider target concept functions, as defined in Def. 10, that are related to (Promise) BQP complete languages. As discussed in (Gyurik & Dunjko, 2023), obtaining a learning separation for the identification problem in this case requires restricting the hypothesis class available to the classical learning algorithm. Otherwise, a classical learner can always outputs the circuit for the whole quantum learner with hardwired data as the output. A common approach is then to define the hypothesis class as exactly the same set of functions contained in the concept class that label the data. The classical learning algorithm will then have to correctly identify which is the concept, among the ones in the concept class, that labeled the data. As we will make clear later, the task is close to a variant of the PAC learning framework in Def. 3, called proper PAC learning.

**Definition 4** (Proper PAC learning (Kearns & Vazirani, 1994)). A concept class $\mathcal{F} = \{\mathcal{F}_n\}_{n \in \mathbb{N}}$ is efficiently proper PAC learnable under the distribution $\mathcal{D} = \{\mathcal{D}_n\}_{n \in \mathbb{N}}$ if it satisfies the definition of PAC learnability given in Def. 3, with the additional requirement that the learning algorithm $\mathcal{L}$ uses a hypothesis class $\mathcal{H}$ identical to the concept class, i.e., $\mathcal{H} = \mathcal{F}$.

We provide now the definition of the two types of concept classes considered in our main Theorem 10.

$c$**-distinct concept class** In the first scenario, we require the concept class to consist of BQP-complete functions, as defined in Def. 10, that disagree on a sufficiently large fraction of inputs. More specifically, we define a $c$-distinct concept class as follows.

**Definition 5** ($c$-distinct concept class). Let $\mathcal{F} = \{f^\alpha : \{0,1\}^n \rightarrow \{0,1\} \mid \alpha \in \{0,1\}^m\}$ be a concept class. We say $\mathcal{F}$ is a $c$-distinct concept class if

$$\forall \alpha_1, \alpha_2 \in \{0,1\}^m, \alpha_1 \neq \alpha_2 \quad \exists S \subset \{0,1\}^n, |S|/2^n \geq c \quad \text{s.t.} \quad \forall x \in S \ \ f^{\alpha_1}(x) \neq f^{\alpha_2}(x). \tag{4}$$

We note that in the case of concept classes containing PromiseBQP functions as defined in Def. 13, for the definition of $c$-distinct concepts to be meaningful, we require that the set $S$ is a subset of the inputs specified in the promise. In the Appendix F.2, we provide an example of a $0.5$-distinct concept class where the concepts are all (Promise)BQP functions.

**Average-case-smooth concept class** We now consider concept classes where the label space is equipped with a metric such that if two concepts are close under the PAC conditions, then their corresponding labels $\alpha_1$ and $\alpha_2$ are also close with respect to the metric on the label space. We formalize this notion of closeness in the definition below.

**Definition 6** (Average-case-smooth concept class). Let $\mathcal{F} = \{f^\alpha : \{0,1\}^n \rightarrow \{0,1\} \mid \alpha \in \{0,1\}^m\}$ be a concept class. We say that $\mathcal{F}$ is *average-case-smooth* if there exists a distance function $d : \{0,1\}^m \times \{0,1\}^m \rightarrow \mathbb{R}^+$ defined over the labels $\alpha \in \{0,1\}^m$ and a $C \geq 0$ such that $\forall \alpha_1, \alpha_2 \in \{0,1\}^m$:

$$\mathbb{E}_{x \sim \mathrm{Unif}(0,1^n)}|f^{\alpha_1}(x) - f^{\alpha_2}(x)| \geq C \, d(\alpha_1, \alpha_2). \tag{5}$$

---

[3]The hypotheses (and concepts) are specified according to some enumeration $R : \cup_{n \in \mathbb{N}}\{0,1\}^n \rightarrow \cup_n \mathcal{H}_n$ (or, $\cup_n \mathcal{C}_n$) and by a "specification of $h \in \mathcal{H}_n$" we mean a string $\sigma \in \{0,1\}^*$ such that $R(\sigma) = h$ (see (Kearns & Vazirani, 1994) for more details).

We note that in machine learning, it is often the case that closeness of functions in parameter space implies closeness in the PAC sense. However, in Def. 6, we require the reverse: that closeness at the function level implies closeness in parameter space. Nevertheless, in the Appendix F.2, we provide an example of an average-case-smooth concept class of quantum implementable functions.

## 2.3 DEFINITIONS OF THE IDENTIFICATION TASKS

Imagine we are given a machine learning task defined by a concept class $\mathcal{F}$ composed of $2^m$ target functions $f^\alpha \in \mathcal{F}$, each one specified by a vector alpha $\alpha \in \{0,1\}^m$ with $m$ scaling polynomially with input size $n$. Formally, that means that there exists a function $e : S \subseteq \{0,1\}^m \to \mathcal{F}$ which is bijective and such that $e(\alpha) = f^\alpha \in \mathcal{F}$. By abuse of notation, in this paper we often use $\alpha$ to refer to $e(\alpha)$ when it is clear from the context that we mean the function $f^\alpha$, rather than the vector $\alpha \in \{0,1\}^m$. Given a training set of inputs labeled by one of these concepts, the goal of the *identification learning algorithm* is to "recognize" the target concept which labeled the data and output the corresponding[4] $\alpha$. We address two closely related but subtly different versions of identification tasks, for which we provide precise definitions below. In the first one, we assume the learning algorithm can decide if a dataset is *invalid*, i.e. if it is not consistent with any of the concepts in the concept class. In particular, we adopt the notion of consistency as defined in Definition 3 of (Kearns & Vazirani, 1994) and regard a dataset as valid if every point in it is labeled in accordance with a concept. In Appendix A.4 we explain that this version of the task is closely related to the learning framework of the so-called consistency model found in the literature (Kearns & Vazirani, 1994; Mohri, 2018; Schapire, 1990).

**Definition 7** (Identification task - verifiable case). Let $\mathcal{F} = \{f^\alpha(x) \in \{0,1\} \mid \alpha \in \{0,1\}^m\}$ be a concept class[5]. A *verifiable* identification algorithm is a (randomized) algorithm $\mathcal{A}_B$ such that when given as input a set $T = \{(x_\ell, y_\ell)\}_{\ell=1}^B$ of at least $B$ pairs $(x,y) \in \{0,1\}^n \times \{0,1\}$, an error parameter $\epsilon > 0$ and a random string $r \in R$, it satisfies:

- If $\nexists \, \alpha \in \{0,1\}^m$ such that $y_\ell = f^\alpha(x_\ell)$ holds for all $(x_\ell, y_\ell) \in T$, then $\mathcal{A}_B$ outputs "*invalid dataset*".

- If the samples in $T$ come from the distribution $\mathcal{D}$, i.e. $x_\ell \sim \mathcal{D}$, and there exists an $\alpha \in \{0,1\}^m$ such that $y_\ell = f^\alpha(x_\ell) \in \{0,1\}$ holds for all $(x_\ell, y_\ell) \in T = \{(x_\ell, y_\ell)\}_{\ell=1}^B$, then with probability $1 - \delta$ it outputs:

$$\mathcal{A}_B(T, \epsilon, r) = \tilde{\alpha}, \quad \tilde{\alpha} \in \{0,1\}^m, \tag{6}$$

  satisfying $\mathbb{E}_{x \sim \mathcal{D}} |f^\alpha(x) - f^{\tilde{\alpha}}(x)| \leq \epsilon$.

  We say that $\mathcal{A}_B$ solves the identification task for a concept class $\mathcal{F}$ under the input distribution $\mathcal{D}$ if the algorithm works for any values of $\epsilon, \delta \geq 0$. The success probability $1 - \delta$ is over the training sets where the input points are sampled from the distribution $\mathcal{D}$ and the internal randomization of the algorithm. The required minimum size $B$ of the input set $T$ scales as poly($n, 1/\epsilon, 1/\delta$), while the running time of the algorithm scales as poly($B, n$).

It will be instructive to think about the algorithm $\mathcal{A}_B$ as a mapping that, once $\epsilon$ and $r$ are fixed, takes datasets $T \subseteq \{(x,y) \mid x \in \{0,1\}^n, \ y \in \{0,1\}\}$ with $|T| \geq B$, and outputs labels $\alpha \in \{0,1\}^m$. The verifiability condition will play a crucial role in our hardness result for the verifiable case of the identification task. Nevertheless, in our main hardness result, we will show how to relax the verifiability condition on the learning algorithm and provide hardness result for the existence of a proper PAC learner algorithm which does not reject any input dataset but satisfies the following additional assumptions. We call such learning algorithm an *approximate-correct* identification algorithm.

**Definition 8** (Identification task - non verifiable case ). Let $\mathcal{F} = \{f^\alpha : \{0,1\}^n \to \{0,1\} \mid \alpha \in \{0,1\}^m\}$ be a concept class. An *approximate-correct* identification algorithm is a (randomized) algorithm $\mathcal{A}_B$ such that when given as input a set $T = \{(x_\ell, y_\ell)\}_{\ell=1}^B$ of at least $B$ pairs $(x,y) \in$

---

[4]More precisely, the algorithm is allowed to output any $\tilde{\alpha}$ which is close in PAC condition with $\alpha$. See Def. 7 and 8.

[5]Here and throughout the paper, we assume that the concepts are labeled by a vector $\alpha$ in $\{0,1\}^m$. However, it is not required that $\alpha$ spans the entire set of bitstrings in $\{0,1\}^m$. The key requirement is that $m$ is sufficiently large to ensure that every concept in the concept class can be assigned a unique vector in $\{0,1\}^m$

$\{0, 1\}^n \times \{0, 1\}$, an error parameter $\epsilon > 0$ and a random string $r \in R$ satisfies the definition of a proper PAC learner (see Def. 4) along with the following additional properties:

1. If for any $\alpha$ all the $(x_\ell, y_\ell) \in T$ are such that $y_\ell \neq f^\alpha(x_\ell)$ then there exists an $\epsilon_1$ such that for all $\epsilon \leq \epsilon_1$ and all $r \in R$:
$$\mathcal{A}_B(T, \epsilon_1, r) \neq \alpha. \tag{7}$$
In other words, the algorithm will never output a totally incorrect $\alpha$, i.e. an $\alpha$ inconsistent with all the inputs in the dataset.

2. If $T = \{(x_\ell, y_\ell)\}_{\ell=1}^B$ is composed of different inputs $x_\ell$ and if there exists an $\alpha$ such that $y_\ell = f^\alpha(x_\ell)$ for all $(x_\ell, y_\ell) \in T$, then there exists a threshold $\epsilon_2$ such that for any $\epsilon \leq \epsilon_2$ there exists at least one $r \in R$ for which:
$$\mathcal{A}_B(T, \epsilon_2, r) = \alpha_2 \tag{8}$$
With the condition: $\mathbb{E}_{x \sim \text{Unif}(\{0,1\}^n)}|f^\alpha(x) - f^{\alpha_2}(x)| \leq \frac{1}{3}$.
Therefore, if the dataset is fully consistent with one of the concept $\alpha$, then there is at least one random string for which the identification algorithm will output a $\tilde{\alpha}$ closer than $\frac{1}{3}$ in PAC condition to the true labelling $\alpha$.

We say that $\mathcal{A}_B$ solves the identification task for a concept class $\mathcal{F}$ under the input distribution $\mathcal{D}$ if the algorithm works for any value of $\epsilon, \delta \geq 0$. The required minimum size $B$ of the input set $T$ is assumed to scale as $\text{poly}(n, 1/\epsilon, 1/\delta)$, while the running time of the algorithm scales as $\text{poly}(B, n)$. Moreover, the $\epsilon_1$ and $\epsilon_2$ required values scale at most inverse polynomially with $n$.

Appendix H examines the two assumptions made about the approximately correct algorithm and presents a concept class where just a standard proper PAC learner meets both conditions.

# 3 HARDNESS RESULTS FOR RANDOM GENERATABILITY OF QUANTUM FUNCTIONS

We now show our results on the hardness of random generability of quantum functions based on the assumptions that BQP is not contained in the second level of the PH, or, in case of approximate random generability, heuristically decided within HeurBPP by an efficient classical machine with a NP oracle. Classical hardness for exact random generatability of quantum functions can be proved also based on a different complexity-theoretic assumption, i.e. $\text{BPP}/\text{samp}^{\text{BQP}} \not\subseteq \text{BPP}$ (we refer to Appendix B and the works in Marshall et al. (2024); Huang et al. (2021) for the precise definition of the class $\text{BPP}/\text{samp}^{\text{BQP}}$, but roughly speaking, this class requires the sample generator to be a polynomial-time quantum computer). For completeness, we also prove this latter result in the Appendix D.

In words, Theorem 1 and 2 state that if $x$ are labeled by a BQP function $f$, then even classically generating random correctly labeled examples $(x, f(x))$ is hard. It is important to note that, in general, the hardness of computing a function does not imply the hardness of generating random labeled samples from it. For instance, while computing the discrete logarithm modulo a large prime is believed to be classically intractable, it is nevertheless possible to efficiently generate evaluations of this function on uniformly random inputs classically. In this sense, our result does not follow trivially from the classical intractability of the target quantum functions.

**Theorem 1** (Exact Random generatability implies $\text{BQP} \subseteq \text{P}^{\text{NP}}$). *Let $f = \{f_n\}_{n \in \mathbb{N}}$ be a family of* BQP *functions as in Def. 10. If there exists a classical randomized poly-time uniform algorithm that generates samples $(x, f_n(x))$ correctly with probability 1 as in Def.1, with $x \sim \text{Unif}(\{0, 1\}^n)$, then* $\text{P}^{\text{NP}}$ *contains* BQP.

*Proof sketch.* The whole proof can be found in the Appendix D.2.1, here we give the main idea. Suppose there exists an algorithm $\mathcal{A}$ which satisfies Def. 1 for a BQP family of function $f$ and for the uniform distribution over the inputs $x \in \{0, 1\}^n$. For a fixed function $f_n : \{0, 1\}^n \rightarrow \{0, 1\}$, the algorithm $\mathcal{A}$ maps a random string $r \in \{0, 1\}^{\text{poly}(n)}$ to a tuple of $(x_r, f_n(x_r))$, i.e. $\mathcal{A}(f_n, .) : \{0, 1\}^{\text{poly}(n)} \rightarrow \{0, 1\}^n \times \{0, 1\}$. Then, in order to prove Theorem 1 we construct an algorithm $\mathcal{A}'$ which can decide the BQP language associated to $f_n$ by using $\mathcal{A}$ and an NP oracle. Such algorithm

$\mathcal{A}'$ will essentially invert $\mathcal{A}$ on a given input $x_{\tilde{r}}$ to find a corresponding valid random string $\tilde{r}$ and then computes $f_n(x_{\tilde{r}})$ using $\mathcal{A}(f_n, \tilde{r})$. Specifically, by using the NP oracle, $\mathcal{A}'$ can find the random string $\tilde{r}$ associated to $x_{\tilde{r}}$ for which $\mathcal{A}(f_n, \tilde{r}) = (x_{\tilde{r}}, f_n(x_{\tilde{r}}))$ in polynomial time. Importantly, finding the string $\tilde{r}$ can be achieved using an NP oracle since $\mathcal{A}$ operates in classical polynomial time, and thus it can efficiently verify the correct string $\tilde{r}$. This concludes the proof as, by Def. 10, $f$ correctly decides an arbitrary BQP language and $\mathcal{A}'$ can evaluate any $f_n$ on every $x_{\tilde{r}}$ by just running $\mathcal{A}(f_n, \tilde{r})$. $\square$

In the next theorem, we allow the algorithm $\mathcal{A}$ to make mistakes both on the distribution followed by the outputted $x$ and we also allow errors on some of the outputted $(x, f_n(x))$.

**Theorem 2** (Approximate Random generatability implies $(\mathsf{BQP}, \mathsf{Unif}) \in \mathsf{HeurBPP}^{\mathsf{NP}}$)**.** *Let $f = \{f_n\}_{n \in \mathbb{N}}$ be a family of* BQP *functions which is the characteristic function of a language $\mathcal{L} \in$ BQP as in Def. 10, and let* Unif *be the uniform distribution over $\{0,1\}^n$. If there exists a polynomial time algorithm $\mathcal{A}$ which satisfies Def. 2 for the uniform input distribution* Unif*, then $(\mathcal{L}, \mathsf{Unif}) \in$ HeurBPP$^{\mathsf{NP}}$.*

*Proof sketch.* The full proof can be found in Appendix D.2.2, here we outline the main idea. We present an algorithm $\mathcal{A}' \in \mathsf{HeurBPP}^{\mathsf{NP}}$ which can heuristically decide the language $\mathcal{L}$ with respect to the uniform distribution over the inputs, i.e. it satisfies Eq. (10) for the input distribution $\mathcal{D} = \mathsf{Unif}$. Let $f = \{f_n\}_n$ be the family of functions associated to the BQP language as in Def. 10. The algorithm $\mathcal{A}'$ follows directly from the one described in the proof of Theorem 1. For a fixed function $f_n$ and error parameter $\epsilon$, the algorithm $\mathcal{A}(f_n, 0^{1/\epsilon}, .)$ in Def. 2 still maps random strings $r \in \{0,1\}^{\mathrm{poly}(n)}$ to tuples $(x_r, f_n(x_r)) \in \{0,1\}^n \times \{0,1\}$. Then, on an input $x_{\tilde{r}}$, $\mathcal{A}'$ still inverts the algorithm $\mathcal{A}$ in order to obtain a random string $\tilde{r}$ such that $\mathcal{A}(f_n, 0^{1/\epsilon}, \tilde{r}) = (x_{\tilde{r}}, f_n(x_{\tilde{r}}))$. This time however, it samples multiple such random strings uniformly at random (this can be done in polynomial time using the result from Bellare et al. (2000)). By doing so, we can guarantee that taking the average of the corresponding $f_n(x_{\tilde{r}_i})$, obtained from $\mathcal{A}(f_n, 0^{1/\epsilon}, \tilde{r}_i) = (x_{\tilde{r}_i}, f_n(x_{\tilde{r}_i}))$ for different $\tilde{r}_i$, will correctly classify the input point $x_{\tilde{r}}$ with high probability. More precisely, as stated in Def. 2, the algorithm $\mathcal{A}$ outputs samples $(x, f_n(x))$ which follow a distribution $\mathcal{L}_\epsilon$ $\epsilon$-close in total variation with the exactly labeled, uniformly sampled over $x \in \{0,1\}^n$ one. It follows (see full proof in the Appendix D.2.2) that the maximum fraction of point $x$ which $\mathcal{A}'$ may misclassify is upper bounded by a linear function of $\epsilon$. $\square$

Although this is tangential to our present discussion, in Appendix D.2.2 we present Corollary 2 of the last theorem, which proves that a certain class of quantum generative models called expectation value samplers (introduced in Romero & Aspuru-Guzik (2021), and proven universal in Barthe et al. (2024) and generalized in Shen et al. (2024)) is classically intractable.

Additionally, our result exhibits a family of distributions that is classically hard to approximate to within nontrivial error. Previous hardness results obtain strong conclusions under widely accepted, purely classical complexity assumptions, such as the non-collapse of PH. Under these widely accepted assumptions, these results rule out efficient classical algorithms for either exact Huang et al. (2025) simulation or approximation beyond an exponentially small additive error Bouland et al. (2018); Movassagh (2019); Bouland et al. (2022). It is important to note that Huang et al. (2025) constructs, for each input size, a single distribution from an explicit family that is provably hard to sample from, whereas Bouland et al. (2018); Movassagh (2019); Bouland et al. (2022) study conditional settings in which each input instance specifies a different output distribution. Our work instead relies on a comparatively less explored assumption, namely that BQP is heuristically not contained in PH and, under this premise, we obtain a correspondingly stronger conclusion: no efficient classical algorithm can approximate the joint distribution $(x, f(x))$ for uniformly random $x$ within inverse-polynomial total variation distance $\epsilon$ (similar to Huang et al. (2025), we construct one hard distribution per input size), whenever $f$ is a BQP function.

## 4 HARDNESS RESULTS FOR THE VERIFIABLE IDENTIFICATION PROBLEM

We now address the second problem studied in this paper, specifically the hardness of the identification task defined in Def. 7.

**Theorem 3** (Identification hardness for the verifiable case ). *Let $\mathcal{F} = \{f^\alpha : \{0,1\}^n \rightarrow \{0,1\} \mid \alpha \in \{0,1\}^m, m = \text{poly}(n)\}$ be a concept class such that there exists a $f^\alpha \in \mathcal{F}$ which is the characteristic function of a language $\mathcal{L} \in \text{BQP}$ as in Def. 10, and let $\text{Unif}$ be the uniform distribution over the $x \in \{0,1\}^n$. If there exists a verifiable identification algorithm $\mathcal{A}_B$ for the uniform input distribution $\text{Unif}$ as given in Def. 7, then $(\mathcal{L}, \text{Unif}) \in \text{HeurBPP}^{\text{NP}}$.*

*Proof sketch.* The full proof can be found in the Appendix E.1, here we outline the proof sketch. The core idea is to show that if there exists an algorithm $\mathcal{A}_B$ that satisfies Def. 7 for the concept class $\mathcal{F}$ in the theorem, then there exists a polynomial time algorithm $\mathcal{A}'$ which, using $\mathcal{A}_B$ and an NP oracle, takes as input any $\alpha \in \{0,1\}^m$ (which uniquely specifies a concept[6]) and outputs a dataset of $B$-many inputs labeled by a concept $f^{\tilde{\alpha}}$ in agreement with $f^\alpha$ under the PAC condition of Eq. 3. We recall here that, given an error parameter $\epsilon$ and a random string $r$, the algorithm $\mathcal{A}_B$ takes as input any set $T$ of $B$ pairs $(x, y) \in \{0,1\}^n \times \{0,1\}$ and, if and only if the set $T$ is consistent with one of the target concepts $\alpha$, outputs with high probability a $\tilde{\alpha} \in \{0,1\}^m$ close in PAC condition to $\alpha$. Note that the crucial "if and only if" condition stems directly from the ability of $\mathcal{A}_B$ to detect invalid datasets, as described in Def. 7. We can then leverage this to construct the algorithm $\mathcal{A}'$ which correctly classifies the $\mathcal{L}$ language associated to $f^\alpha$. Specifically, on any input $x \in \{0,1\}^n$, the algorithm $\mathcal{A}'$ first inverts $\mathcal{A}_B$ on the target $\alpha$ in order to obtain a dataset $T = \{(x_\ell, y_\ell)\}_{\ell=1}^B$ of $B$ input points such that $\mathcal{A}_B(T, \epsilon, .) = \alpha$. This is possible by using an NP oracle to search for a dataset $T$ such that $\mathcal{A}_B(T, \epsilon, \cdot) = \alpha$, leveraging the efficiency of the algorithm $\mathcal{A}_B$. It then labels the input $x$ based on consistency with the training set $T$ generated in the previous step. By selecting an appropriate number of inputs $B$, it is possible to bound the probability that the dataset $T$ is consistent with a concept $f^{\tilde{\alpha}}$ heuristically close to the target $f^\alpha$, thereby bounding the probability that the label assigned to $x$ corresponds to $f^\alpha(x)$. $\square$

In Appendix A.4, we observe that the verifiable identification task effectively captures the consistency learning model commonly used in the literature. Another compelling reason to consider learning algorithms within the verifiable case is that quantum learners *can* verify whether a dataset is valid for a given $\alpha$. Specifically, given an input dataset $T$, the quantum learning algorithm outputs the guessed $\alpha$ and then can check whether every point in the dataset are correctly labeled by $f^\alpha$, similar to the process for efficiently evaluable hypotheses. Assuming $f^\alpha$ is quantum evaluable, and the dataset is polynomial in size, the verification procedure runs in polynomial time. A following question is then whether in order to verify a dataset we necessarily need a quantum computer for the BQP functions defined in Def. 10. We address this question in the following proposition, which asserts that for a singleton concept class[7], determining whether a dataset is valid is possible if the concept can be evaluated in the class P/poly (see Def. 15 in Appendix B for a definition of P/poly).

**Proposition 1** (Hardness of verification - singleton case). *Let $\mathcal{F} = \{f : \{0,1\}^n \rightarrow \{0,1\}\}$ be a singleton concept class. If there exists an efficient algorithm $\mathcal{A}_B$ such that for every set $T = \{(x_\ell, y_\ell)\}_{\ell=1}^B$, with $B$ polynomial in $n$, $x_\ell \in \{0,1\}^n$ such that $x_i \neq x_j \, \forall \, i, j \in \{1, ..., B\}$ and $y_\ell \in \{0,1\}$, satisfies the following:*

- *If $\exists (x_\ell, y_\ell) \in T, \, y_\ell \neq f^\alpha(x_\ell)$, $\mathcal{A}_B(T)$ outputs "invalid dataset".*

*Then there exists a polynomial time non-uniform algorithm which computes $f(x)$ for every $x$. In particular, there exists an algorithm in P/poly which computes $f(x)$.*

*Proof.* For every input size $n$, let us take as polynomial advice the $B - 1$ samples $T^{B-1} = \{(x_\ell, f(x_\ell))\}_{\ell=1}^{B-1}$, for any sequence of different $x_\ell \in \{0,1\}^n$. Then, on any input $x \in \{0,1\}^n$, the algorithm in P/poly would run $\mathcal{A}_B$ on the dataset $T_x = \{(x, 0)\} \cup T^{B-1}$ and decide $x$ based on the corresponding output of $\mathcal{A}_B$. $\square$

---

[6]We assume that each $\alpha$ provides an unambiguous specification of the concept $f^\alpha$ (possibly as a quantum circuit, i.e. that a quantum circuit can, given $\alpha$, evaluate $f^\alpha$ in polynomial time).

[7]A singleton concept class is a concept class that consists of only one concept.

We specify that we require the inputs $x_\ell$ to be distinct in the training set $T$ to strengthen the result in Proposition 1. Without this condition, the result would be trivial, as one could provide $\mathcal{A}_B$ with $B$ identical copies of $(x, 0)$ as input and correctly decide each $x$ using a polynomial-time uniform algorithm by simply examining the corresponding output. In case of exponential-sized concept classes containing a BQP function which uniquely labels a set of polynomial number of inputs, the verifiability property of the identification algorithm can be used to prove that BQP is contained in the class P/poly.

**Theorem 4.** *Let $\mathcal{F} = \{f^\alpha : \{0, 1\}^n \to \{0, 1\} \mid \alpha \in \{0, 1\}^m, m = \mathrm{poly}(n)\}$ be a concept class such that there exists at least one function $f^\alpha \in \mathcal{F}$ that decides a language $\mathcal{L} \in$ BQP, and for which there exists a polynomial-sized subset $S \subset \{0, 1\}^n$ such that $f^\alpha$ is uniquely determined by its labels on $S$. If there exists a verifiable identification algorithm $\mathcal{A}_B$ as given in Def. 7, then BQP $\subseteq$ P/poly.*

*Proof.* For every input size $n$, let us take as polynomial advice the samples from the subset $S$ which uniquely determine $f^\alpha$ and the corresponding labels, i.e. $T^S = \{(x_\ell, f(x_\ell))\}_{\ell=1}^{|S|}$, such that $y_\ell = f^\alpha(x_\ell)$ for every $x_\ell \in S$. We then consider exactly that dataset as advice and we label any new input $x \in \{0, 1\}^n$ selecting the $y \in \{0, 1\}$ which keeps the dataset valid. More precisely, on any input $x \in \{0, 1\}^n$, the algorithm in P/poly would run $\mathcal{A}_B$ on the dataset $T_x = \{(x, 0)\} \cup T^S$ and label $x$ so that the algorithm $\mathcal{A}_B$ accepts the dataset. $\square$

## 5 HARDNESS RESULT FOR THE NON-VERIFIABLE IDENTIFICATION PROBLEM

We now present the main result of our paper, which establishes hardness for the non-verifiable identification task under the assumption that it is solvable by an approximately correct algorithm. The theorem states that, assuming BQP is not contained in a low level of PH, then no efficient classical learner can solve the identification problem for a broad class of concept classes. In particular, the result applies to concept classes that either include $c$-distinct concepts, as defined in Def. 8 with $c \geq \frac{1}{3}$, or satisfy the average-case smoothness property given in Def. 6.

**Theorem 5** (Hardness of the identification task - non verifiable case). *Let $\mathcal{F} = \{f^\alpha : \{0, 1\}^n \to \{0, 1\} \mid \alpha \in \{0, 1\}^m\}$ be a concept class containing at least a BQP function $f$ as in Def. 10 associated to a language $\mathcal{L} \in$ BQP. Assume further that $\mathcal{F}$ is either a $c$-distinct concept class with $c \geq 1/3$ or an average-case-smooth concept class. If the non-verifiable identification task for $\mathcal{F}$, as defined in Def. 8, is solvable by a classically efficient approximate-correct identification algorithm $\mathcal{A}_B$, then it follows that $(\mathcal{L}, \mathsf{Unif}) \in \mathsf{HeurBPP}^{\mathsf{NP}^{\mathsf{NP}^{\mathsf{NP}}}}$.*

*Proof sketch.* The proof can be found in the Appendix F. The proof combines the intermediate result in Theorem 11 with the result in Theorem 13 for the $c$-distinct concept classes or in Theorem 14 for average-case-smooth concept classes. The intuition is the following. In Theorem 13 and Theorem 14 we show that for $c$-dinstict or average-case-smooth concept classes the existence of an approximate-correct identification algorithm in Def. 8 allows for the construction of an identification algorithm in the first level of the PH which is able to reject any dataset which contains a fraction of inputs greater than $\frac{1}{\beta}$ not labeled by $f^\alpha$. Then in Theorem 11 we prove that if such an algorithm exists, then on a given $\alpha$ we can invert it climbing up two more layers in the PH in order to obtain a dataset of inputs mostly consistent with $f^\alpha$. Because each of these datasets will contain only a fraction of $\frac{1}{\beta}$ inputs incorrectly labeled, we can guarantee that the fraction of misclassified inputs can be made polynomially small. Thus we can evaluate $f^\alpha$ in $\mathsf{HeurBPP}^{\mathsf{NP}^{\mathsf{NP}^{\mathsf{NP}}}}$. $\square$

A full scheme of the proof overview can be found in Figure1.

Finally, we show that there exists a concrete learning problem for which a quantum learner can successfully identify the unknown label function, while any efficient classical method would fail. In particular, in Appendix G.1 we provide an example of a natural $c$-distinct concept class with $c \geq \frac{1}{3}$ which then satisfies the assumption of Theorem 10 and lead to the following final result.

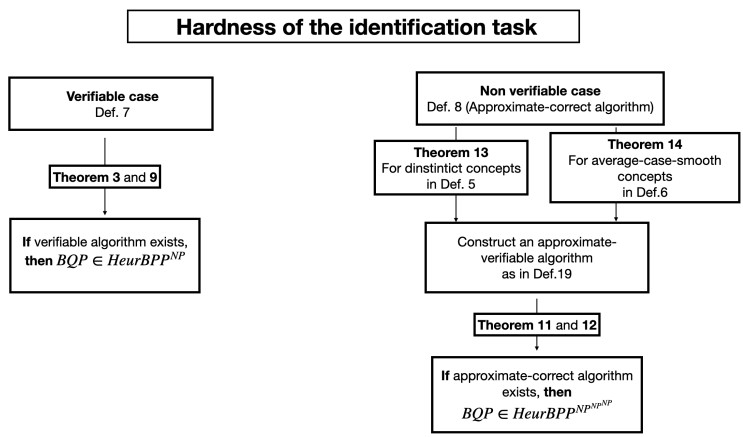

Figure 1: An overview of our proof strategy for the identification task.

**Corollary 1** (Informal, see Theorem 15 for a formal version). *There exists a natural learning problem involving quantum functions for which there exist a quantum learning advantage for the identification task (i.e. the target concepts are efficiently identifiable on a quantum computer but not on a classical computer) unless* BQP *is in the (Heuristic) polynomial hierarchy.*

The proof, along with the specific instance of the learning task that demonstrates the advantage, is provided in Appendix G.1.

## 6 DISCUSSION

In this paper, we have demonstrated learning separations for the identification task in the case of quantum target functions. In the case of concept classes consisting purely of quantum functions, provable quantum advantages in the PAC framework have so far been established only at the level of the evaluation step, i.e., when the learning algorithm is also required to efficiently evaluate the learned hypothesis. In this work, we instead analyze the hardness for a classical algorithm to identify a correct hypothesis within the given concept class itself (noting that, if no restriction is placed on the hypothesis class, a classical learner can always solve the task by outputting the circuit of the quantum learner with hardwired data, as shown in Gyurik & Dunjko (2023)). In this sense, our results provide a first quantum–classical separation for learning concept classes of quantum functions in the setting of proper PAC learning.

A natural question is whether our results extend to existing, well-studied learning tasks. In Appendix G, we examine in detail three scenarios where our results may be applicable. First, we show that our theorems directly apply to the task of learning observables introduced in (Molteni et al., 2024), for which a learning separation had previously been established only at the evaluation stage and we outline a potential benchmark against a classical dequantized algorithm. We then consider two physically relevant tasks, such as Hamiltonian learning and learning the order parameter, that can naturally be formulated as identification problems. In one common formulation of Hamiltonian learning, one is given access to measurements on the Gibbs state of an unknown Hamiltonian at a fixed temperature and asked to recover the Hamiltonian. This can be cast as an identification problem by defining a concept class whose elements are functions mapping measurement settings to expectation values on the Gibbs state of the target Hamiltonian. In this sense, Hamiltonian learning is closely related to our identification task, although there are several important differences, which we spell out in Appendix G. Importantly, since Hamiltonian learning is known to be classically solvable Anshu et al. (2020); Haah et al. (2024), this suggests that the additional structural assumptions we impose on concept classes for our hardness theorems ($c$-distinctness or average-case smoothness) are not easily relaxed. As a consequence of our results, a general hardness theorem for identification without such constraints, together with the classical tractability of Hamiltonian learning, would lead to unexpected consequences in complexity theory. We further elaborate on this, as well as on the connection to learning an unknown order parameter, in Appendix G.

## ACKNOWLEDGMENTS

The authors are thankful to Sofiene Jerbi for numerous discussions and suggestions to look into Reed-Solomon codes, and to Matthias Caro for discussions involving the packing numbers of certain quantum concept classes. This publication is part of the project Divide & Quantum (with project number 1389.20.241) of the research programme NWA-ORC which is (partly) financed by the Dutch Research Council (NWO). This work was also supported by the Dutch National Growth Fund (NGF), as part of the Quantum Delta NL programme. This work was also supported by the European Union's Horizon Europe program through the ERC CoG BeMAIQuantum (Grant No. 101124342).

## REPRODUCIBILITY STATEMENT

This manuscript presents exclusively theoretical results. To ensure reproducibility, we provide fully detailed proofs of all theorems claimed in the main text, which are included in the Appendix. Additionally, key proof ideas are outlined within the main body of the paper for further clarity.

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

# A RELATED WORKS

## A.1 PREVIOUS RESULTS ON LEARNING SEPARATIONS FOR QUANTUM FUNCTIONS

The study of different types of learning separations in quantum machine learning was initiated in Gyurik & Dunjko (2023), distinguishing between advantages in the identification step and those arising solely from the evaluation of the target function. For BQP-complete functions, the authors showed that learning separations straightforwardly follow under the assumption that there exists an input distribution $\mathcal{D}$ and a BQP language $L$ such that $(L, \mathcal{D}) \not\subseteq$ HeurP/poly. This separation relied on requiring the learning algorithm to evaluate the learned function on new inputs, thus considering the evaluation step only. The paper left open the question of whether a stronger result could be established by proving that even identifying the correct target function is classically hard for BQP-complete concepts, which we address in this paper. In Molteni et al. (2024), stronger learning separations were established for BQP functions under the widely studied assumption that BQP $\not\subseteq$ P/poly (exactly, not with respect to heuristic conditions as in Gyurik & Dunjko (2023)), by reverting to standard PAC learning where the learning process must be successful for all input distributions (as opposed to settings where the distribution is fixed). Additionally, there the authors proposed a quantum algorithm capable of correctly identifying and evaluating the target concept in the nontrivial case of a superpolynomially large concept class, as opposed to Gyurik & Dunjko (2023) where only polynomially large classes were studied. However, even in this case, the learning separation was demonstrated only for the evaluation step. Learning separations for superpolynomially large concept classes were also presented in Yamasaki et al. (2023), though they are based on assumptions about heuristic classes similar to Gyurik & Dunjko (2023) and were not focused on finding physically motivated problems, which was instead the goal in Molteni et al. (2024). In Appendix G.1, we show that our result directly applies to the physically relevant learning problem considered in Molteni et al. (2024).

## A.2 PREVIOUS RESULTS ON HARDNESS OF THE IDENTIFICATION TASK

Hardness results for the identification step are already known for certain tasks and it is important to highlight why the proof strategies used in those cases do not apply to the learning task considered in this paper, which involves quantum functions. In particular, hardness results for identification are already known in the context of learning DNF formulas (Kearns & Vazirani, 1994) and for specific cryptographic functions, such as modular exponentiation (Gyurik & Dunjko, 2023) and discrete cube root (Jerbi et al., 2024). In all of these examples, a key element underlying the hardness proofs is the existence of a representation of the target functions that allows efficient computation of properties of interest. In the case of learning DNF formulas, (Kearns & Vazirani, 1994) showed that DNF formulas cannot be learned in the proper PAC setting—that is, given samples of variable assignments along with their evaluations under a DNF formula, it is possible to reconstruct the original formula. While the proof in (Kearns & Vazirani, 1994) reduces the problem of learning DNF formulas to the NP-complete problem of graph coloring, the following argument provides what we see as an intuitive explanation of how the key element mentioned above plays a role in the proof. It is well known that checking the satisfiability of CNF formulas is NP-hard, whereas for DNF formulas this task is classically easy. Moreover, since DNF formulas are universal for Boolean functions, any CNF formula can be equivalently expressed as a DNF[8]. Of course, the transformation from CNF to DNF is itself NP-hard. However, if DNF formulas were learnable in the proper PAC sense, then a classical learner could, given samples from any target CNF, reconstruct the equivalent DNF representation and thereby decide satisfiability efficiently.

In the case of cryptographic functions the existence of an easy-to-compute representation implies a mapping between two representations of the same set of functions: one that is computationally intractable and another that is classically efficient to compute. If identifying the target concept in the second representation were classically efficient, it would also allow the evaluation of the target function in its hard-to-compute representation. Examples of this are provided in (Kearns & Valiant, 1994; Gyurik & Dunjko, 2023) based around the discrete cube root function which is intractable given the input, but has an equivalent representation as modular exponentiation by some (hard-to-compute) exponent. Furthermore, in the case of cryptographic functions, thanks to the additional property

---

[8]Note that in general the conversion requires an exponential size DNF. This is the reason why our intuition cannot be trivially sharpened into a proof.

of random generatability, a classical algorithm could easily generate labeled data and solve the identification problem using the hypothesis class corresponding to the easy-to-compute representation of the function. Once the correct concept is identified, the algorithm could then evaluate it, thereby violating the cryptographic assumptions of the considered set of functions.

For the BQP-complete functions analyzed in this paper, our findings on the hardness of random generatability already rule out the possibility of employing similar proof techniques to establish the hardness of identification. Finally, we observe that our proof does not rely on the existence of a simple representation for the target BQP functions. Rather, in our case, we directly tackle the property of the learning algorithm to find the label from data, by exploiting the fact this can be "inverted" in the PH. This approach is then really relying on the strongest properties of quantum functions as it could not work for cryptographic functions which typically are in the PH.

### A.3 RELATION TO PREVIOUS WORKS ON HARDNESS OF SAMPLING

In the work Aaronson (2014) the authors showed an equivalence between sampling and searching problem. In particular, they showed that if classical computers could solve every search problem quantum solvable, then classical computer could also sample from the output distribution of any quantum circuit. While this might seem related to our results about the hardness of random generability of quantum functions in Section 3, we note that the problems addressed are fundamentally different. In particular, our results on classical intractability do not regard the whole class of distribution realizable by quantum circuits (i.e, SampBQP), nor do we make claims about the hardness of that class. We only consider the distributions given by the tuple $(x, f(x))$ with $f$ the characteristic function of a BQP language. Such distributions do not constitute the whole of the class SampBQP, and thus the capacity to classically sample from them does not straightforwardly imply the capacity to compute (all) quantum functions. Indeed, because we are considering a much more restricted sampling capability, our results are in fact weaker. We do not claim that sampling from these distributions would imply FBQP = FBPP. Instead, we show a weaker result: if a classical algorithm could sample pairs $(x, f(x))$, then it would imply that $\text{BQP} \subseteq \text{HeurBPP}^{\text{NP}}$.

### A.4 CONSISTENCY MODEL LEARNING

As argued in the text, the ability of the learning algorithm to detect invalid dataset is crucial in the construction of proof of Theorem 9 for the verifiable identication task. Although it is a strong assumption (which we will remove in Section 5 for certain families of concept classes) we can still argue that the verifiable case of Def. 7 is of interest. An important reason is that Def. 7 perfectly captures the framework of learning in the consistency model present in the literature Balcan (2015); Schapire (2008); Mohri (2018); Kearns & Vazirani (1994). In Kearns & Vazirani (1994), a concept $f^\alpha$ is defined to be consistent with a dataset $T^\alpha = \{(x_\ell, y_\ell)\}_{\ell=1}^B$ if $y_\ell = f^\alpha(x_\ell)$ for every $\ell = 1, ..., B$. Based on that, we give the following definition of learning in the consistency model framework, which includes the ability of the learning model to distinguish invalid datasets. As we explain below, this assumption aligns with the general case found in the literature Kearns & Vazirani (1994); Schapire (1990); Mohri (2018), where only efficient hypothesis classes are considered.

**Definition 9** (Consistency model learning Balcan (2015); Schapire (2008)). We say that an algorithm $\mathcal{A}$ learns the class $\mathcal{F} = \{f^\alpha : \{0,1\}^n \to \{0,1\} \mid \alpha \in \{0,1\}^m, m = \text{poly}(n)\}$ in the consistency model if, given any set of labeled examples $T$, the algorithm produces a concept $f \in \mathcal{F}$ that is consistent with $T$, if such a concept exists, and outputs "there is no consistent concept" otherwise. The algorithm runs in polynomial time (in the size of $T$ and the size $n$ of the examples).

Given the definition of learning in the consistency model in Def. 9, it is clear that an algorithm $\mathcal{A}_B$ solves the identification task in Def. 7 for a given concept class $\mathcal{F}$ if and only if it learns $\mathcal{F}$ in the consistency model.

## B DEFINITIONS FROM COMPLEXITY THEORY

To enhance readability, we provide a glossary of the most commonly used symbols in this paper.

| Symbol | Meaning |
|---|---|
| $f^\alpha(x)$ | Concept function parameterized by $\alpha$ |
| $\mathcal{F}$ | Concept class (set of labeling functions $\mathcal{F} = \{f^\alpha \mid \alpha \in \{0,1\}^m\}$ ) |
| $\mathcal{D}$ | Distribution over input space of bitstrings $x \in \{0,1\}^n$ |
| Unif | Uniform distribution over input space of bitstrings $x \in \{0,1\}^n$ |
| BQP | Class of decision problems solvable by a quantum computer in polynomial time with bounded error |
| NP | Class of decision problems verifiable in polynomial time |
| PH | Polynomial Hierarchy |
| $P^{NP}$ | Class of problems solvable in P with access to an NP oracle |
| HeurBPP | Class of problems heuristically solvable in BPP |
| HeurBPP$^{NP}$ | Class of problems solvable in HeurBPP with access to an NP oracle |
| HeurBPP$^{NP^{NP^{NP}}}$ | Class of problems solvable in HeurBPP with access to an $NP^{NP^{NP}}$ oracle |

Table 1: Glossary of commonly used symbols and complexity classes.

## B.1 TOOLS FROM COMPLEXITY THEORY

All our hardness results are based on the assumption that the class BQP is not contained in (a low level of) the polynomial hierarchy (PH). See Appendix C for arguments supporting this assumptions. Specifically, given a BQP language $\mathcal{L}$, we define a function $f$ which "correctly decides" $\mathcal{L}$ as follows.

**Definition 10** (BQP function $f$). We refer to a function $f : \{0,1\}^n \to \{0,1\}$ as a BQP function if there exists a language $\mathcal{L} \in$ BQP such that $f$ is its characteristic function. In particular:

$$f(x) = \begin{cases} f(x) = 0 \text{ if } x \notin \mathcal{L} \\ f(x) = 1 \text{ if } x \in \mathcal{L} \end{cases}. \tag{9}$$

Where, as previously said, we use $f$ to refer to a uniform family of functions, one for each input size. In our hardness results, we assume the existence of a BQP language $\mathcal{L}$ such that, under the uniform distribution Unif over inputs $x \in \{0,1\}^n$, the pair $(\mathcal{L}, \text{Unif})$ is not contained in HeurBPP with access to NP oracles in the PH. We next provide a precise definition for the complexity class HeurBPP (for more details see Bogdanov et al. (2006)). To define heuristic complexity classes, we first need to consider the so-called distributional problems (which should not be confused with learning distributions in the context of unsupervised learning).

**Definition 11** (Distributional problem). A distributional problem $(\mathcal{L}, \mathcal{D})$ consists of a language $L \subseteq \{0,1\}^*$ and a family of distributions $\mathcal{D} = \{\mathcal{D}_n\}_{n \in \mathbb{N}}$ such that $\text{supp}(\mathcal{D}_n) \subseteq \{0,1\}^n$.

We are now ready to provide a precise definition of the class HeurBPP.

**Definition 12** (Class HeurBPP). A distributional problem $(\mathcal{L}, \mathcal{D})$ is in HeurBPP if there exists a polynomial-time classical algorithm such that for all $n$ and $\epsilon \geq 0$:

$$\Pr_{x \sim \mathcal{D}} \left[ \Pr[\mathcal{A}(x, 0^{1/\epsilon}) = \mathcal{L}(x) \geq \frac{2}{3} \right] \geq 1 - \epsilon, \tag{10}$$

in the above the inner probability is taken over the internal randomization of $\mathcal{A}_B$.

Finally, we observe that by slightly modifying the proofs in the theorems, our hardness results will also hold under a similar but distinct assumption. Specifically, in Appendix F we will prove our hardness results under the assumption that there exists a classically samplable distribution $\text{Unif}_{\mathcal{L}_P}$ such that $(\text{PromiseBQP}, \text{Unif}_{\mathcal{L}_P})$ is not contained in HeurBPP with access to NP oracles in the PH.

## B.2 PROMISE PROBLEMS

We note that our results can also be proven under similar but different assumptions. In particular, by slightly modifying the proofs our main hardness results hold even under the assumption that PromiseBQP problems cannot be solved in HeurBPP with access to NP oracles. We now provide the necessary definitions to prove these versions of our results, which we will do in the following sections. We notice that, unlike BQP, the class PromiseBQP allows for complete problems. We note that complete problems for PromiseBQP are known to emerge in a number of physical processes, we list few of them in Appendix G.4

**Definition 13** (PromiseBQP function $f$). We refer to a function $f : \{0,1\}^n \to \{0,1\}$ as a PromiseBQP function if there exists a language with a promise $\mathcal{L}_P = (\mathcal{L}_{\text{yes}}, \mathcal{L}_{\text{no}}) \subset \{0,1\}^n$ in PromiseBQP such that $f$ is the characteristic function of $\mathcal{L}_P$. In particular:

$$f(x) = \begin{cases} f(x) = 0 \text{ if } x \notin \mathcal{L}_{\text{yes}} \\ f(x) = 1 \text{ if } x \in \mathcal{L}_{\text{yes}} \end{cases} . \tag{11}$$

Additionally, we say that $f$ is a BQP-complete[9] function if the associated language $\mathcal{L}$ is complete for PromiseBQP.

Notice that for a PromiseBQP function $f$, $f(x) = 0$ both if $x$ belongs to the NO subset of the promise, i.e. $x \in \mathcal{L}_{\text{no}}$, and if $x$ does not belong to the promise subset at all, i.e. $x \notin \mathcal{L}_{\text{yes}} \cup \mathcal{L}_{\text{no}}$. In general the promise subset $\mathcal{L}_{\text{yes}} \cup \mathcal{L}_{\text{no}}$ contains an exponentially small fraction of all the possible input $x \in \{0,1\}^n$ and consequently $f = 0$ for the majority of $x \in \{0,1\}^n$. This means that correctly evaluating $f$ on average over the input, for instance under the uniform distribution over all $x \in \{0,1\}^n$, becomes trivial unless the error is exponentially small. We refer to Gyurik & Dunjko (2023) for a more detailed discussion on this. We will prove our hardness results by showing that a classical machine with access to an NP oracle can achieve achieve average-case correctness for evaluating a PromiseBQP function $f$ with respect a given input distribution. For PromiseBQP functions, to ensure the task remains non-trivial, we will state our hardness results with respect to the input distribution $\mathsf{Unif}_{\mathcal{L}_P}$ which is defined as the uniform distribution over a subset of the promise set of input $x \in \mathcal{L}_{\text{yes}} \cup \mathcal{L}_{\text{no}}$. In our hardness results, we assume the existence of a PromiseBQP complete language $\mathcal{L}_P$ such that there exists a distribution $\mathsf{Unif}_{\mathcal{L}_P}$ of hard-to-evaluate but classically samplable instances.

**Definition 14** (Distribution $\mathsf{Unif}_{\mathcal{L}_P}$). Let $\mathcal{L}_P = (\mathcal{L}_{\text{yes}}, \mathcal{L}_{\text{no}})$ be a PromiseBQP complete language. We call $\mathsf{Unif}_{\mathcal{L}_P}$, if it exists, the uniform distribution over a subset $\Pi \subset \mathcal{L}_{\text{yes}} \cup \mathcal{L}_{\text{no}}$ such that the following holds:

- $\mathsf{Unif}_{\mathcal{L}_P}$ is efficiently classically samplable.

For our results in Theorem 2 and Theorem 12 and 9 we will then assume the existence of a PromiseBQP language $\mathcal{L}_P$ for which there exists a $\mathsf{Unif}_{\mathcal{L}_P}$ such that $(\mathcal{L}_P, \mathsf{Unif}_{\mathcal{L}_P})$ is not contained in HeurBPP with access to NP oracles in the PH.

### B.3 ADVICE CLASSES

In this section we provide further useful definitions. We first define the complexity class P/poly which appears in Proposition 1 and Theorem 4.

**Definition 15** (Polynomial advice Arora & Barak (2009)). A problem $L : \{0,1\}^* \to \{0,1\}$ is in P/poly if there exists a polynomial-time classical algorithm $\mathcal{A}_B$ with the following property: for every $n$ there exists an advice bitstring $\alpha_n \in \{0,1\}^{\text{poly}(n)}$ such that for all $x \in \{0,1\}^n$:

$$\mathcal{A}_B(x, \alpha_n) = L(x). \tag{12}$$

We also provide the definition of the class $\mathsf{BPP/samp}^{\mathsf{BQP}}$, which will appear in Theorem 6.

**Definition 16** ($\mathsf{BPP/samp}^{\mathsf{BQP}}$). A problem $L : \{0,1\}^* \to \{0,1\}$ is in $\mathsf{BPP/samp}^{\mathsf{BQP}}$ if there exists a polynomial-time quantum algorithm $\mathcal{S}$ and a polynomial-time classical randomized algorithm $\mathcal{A}$ such that for every $n$:

- $\mathcal{S}$ generates random instances $x \in \{0,1\}^n$ sampled from the distribution $\mathcal{D}_n$.

- $\mathcal{A}$ receives as input $\mathcal{T} = \{(x_i, L(x_i)) \mid x_i \sim \mathcal{D}_n\}_{i=1}^{\text{poly}(n)}$ and satisfies for all $x \in \{0,1\}^n$:

$$\Pr\big(\mathcal{A}(x, \mathcal{T}) = L(x)\big) \geq \frac{2}{3}, \tag{13}$$

where the probability is taken over the internal randomization of $\mathcal{A}$ and $\mathcal{T}$.

---

[9]It is known that BQP do not have complete problems. Here, we will abuse notation and actually refer to PromiseBQP problems.

## C    EVIDENCE THAT BQP $\not\subset$ PH

In this section, we provide a brief discussion on the primary assumptions underlying our work, which are derivatives and versions of BQP $\not\subset$ PH. The first main result in this direction is given in (Aaronson, 2010), where an oracle separation between the relational version of the two classes was proven. Specifically, Aaronson proved that the relation problem *Fourier Fishing*, a variant of the *Forrelation* problem, exhibits an oracular separation $\text{FBQP}^A \not\subset \text{FBPP}^{\text{PH}^A}$. In the same paper Aaronson motivated the study of oracle separation for BQP and PH as lower bounds in a concrete computational model, claiming them as a natural starting point for showing evidence of BQP $\not\subset$ PH. Almost ten years later, in the remarkable work (Raz & Tal, 2022), the authors managed to prove an oracle separation for the decision versions of the classes. Namely they proved the existence of an oracle $A$ such that $\text{BQP}^A \not\subset \text{PH}^A$. Although an unconditional proof of separation between BQP and PH is not likely to appear anytime soon[10], the assumption BQP $\not\subset$ PH is generally considered reasonable.

Most of the results in this paper furthermore rely on the assumption that BQP languages cannot be decided correctly on average by algorithms in HeurBPP, even when given access to oracles for the polynomial hierarchy. This is a stronger assumption than simply BQP $\not\subset$ PH, but we believe not unreasonable. However, we note that a workaround would be possible by requiring a stronger notion of learnability, in particular by moving from the less common setting of distribution-specific PAC of Def. 3 to "standard" PAC, where the learner is required to output a consistent hypothesis for every input distribution. In this case, for any given input $x$, one can apply the classical identification algorithm to a distribution concentrated solely on $x$. To evaluate the target concept on this input, the same strategy used in the proof of Theorem 11 can be employed: namely, inverting the identification algorithm (which now works for the distribution focused on $x$) within PH. This would enable evaluation of the target function at any point $x$, contradicting the assumption that BQP $\not\subset$ PH.

If we choose to remain within the distribution-specific PAC setting, we however believe our assumptions are still widely accepted. Some evidence in this direction comes from sampling problems, where the proof of the best-known separation of SampP and SampBQP (Aaronson & Arkhipov, 2011; Marshall et al., 2024) is identical for the heuristic case and produces the same collapse. To be more exact, suppose that SampBQP can be decided in some heuristic level of the polynomial hierarchy, follow the proof of (Aaronson & Arkhipov, 2011) for a heuristic-equivalent theorem until we get to the definition of (Heur)$\text{GPE}_{\pm}$, at which point we note that the preexisting definition of $\text{GPE}_{\pm}$ is already heuristic, so our $\text{HeurGPE}_{\pm}$ is equal to $\text{GPE}_{\pm}$. Therefore, assuming that SampBQP is in some heuristic level of the sampling analogue of the polynomial hierarchy would still imply the collapse of the standard polynomial hierarchy to some level (standard assumptions from quantum supremacy arguments notwithstanding).

From these arguments, however, no analogous claims regarding the decision (and distributional) variants BQP and Heur PH can be proven using known techniques. For this reason, we stand by the following careful claim: there is no reason to believe that BQP is in Heur PH (relative to relevant distrubutions).

As a final remark, we do highlight that our results do not require considering the whole PH. Specifically it is important to note that for the hardness of random generatability of quantum functions in Section 3 we only need to assume that BQP is not in the second level of the PH, while the assumption extends up to the fourth level for the hardness of the identification problem in Section 5.

## D    PROOFS REGARDING RANDOM GENERATABILITY

### D.1    HARDNESS OF RANDOM GENERATABILITY BASED ON THE ASSUMPTION $\text{BPP}/\text{samp}^{\text{BQP}} \not\subseteq \text{BPP}$

In this section, we demonstrate that achieving exact random generatability for quantum functions would lead to $\text{BPP}/\text{samp}^{\text{BQP}} \subseteq \text{BPP}$. Similar to Theorem 1, we show here classical hardness of the task defined in Def. 1 for quantum functions, though based on a different complexity-theoretic assumption. The proof is straightforward and relies on the idea that if a classical machine could

---

[10]Note that any proof of BQP $\not\subset$ PH would immediately imply the hard-to-prove claim that BQP $\neq$ BPP.

efficiently generate samples for any quantum function, passing those samples as advice would offer no additional advantage.

**Theorem 6** (Exact RG implies $\mathsf{BPP}/\mathsf{samp}^{\mathsf{BQP}} \subseteq \mathsf{BPP}$). *If* $\mathsf{BPP}/\mathsf{samp}^{\mathsf{BQP}} \not\subseteq \mathsf{BPP}$, *then there exists a quantum function* $f : \{0,1\}^n \to \{0,1\}$ *as given Def. 13 which is not exact random verifiable as in Def. 1 with a classical algorithm* $\mathcal{A}$.

*Proof.* Suppose $\forall f$ in Def. 13 there exists a randomized algorithm $\mathcal{A}_f$ such that $\mathcal{A}_f(r) \to (x, f(x))$ with $x \sim \mathrm{Unif}(\{0,1\}^n)$, for $r$ sampled uniformly at random. Then every function $g \in \mathsf{BPP}/\mathsf{samp}^{\mathsf{BQP}}$ can be computed in BPP by first generating the samples $(x, g(x))$ using $\mathcal{A}_g$. $\qquad\square$

We now argue that the hypothesis of the Theorem 6 are indeed reasonable. Indeed, a separation between the class BPP and $\mathsf{BPP}/\mathsf{samp}^{\mathsf{BQP}}$ can be proven if we assume the existence of a sequence of quantum circuits $\{U_n\}_n$, one for each size $n$, such that the $Z$ measurement on the first qubit is hard to decide classically. A proof idea for the following theorem is stated in Huang et al. (2021), and we include here the whole proof for completeness.

**Theorem 7** ($\mathsf{BPP}/\mathsf{samp}^{\mathsf{BQP}} \not\subset \mathsf{BPP}$, unless $\mathsf{BPP} = \mathsf{BQP}$ for unary languages). *If there exists a uniform sequence of quantum circuits* $\{U_n\}_n$, *one for each size* $n = 1, 2, ...$ *such that the* $Z$ *measurement on the first qubit is hard to decide classically, then* $\mathsf{BPP}/\mathsf{samp}^{\mathsf{BQP}} \not\subset \mathsf{BPP}$.

*Proof.* As shown in Huang et al. (2021), such a sequence of circuits would define a unary language

$$L_{BQP} = \{1^n \mid \langle 0^n | (U_n)^\dagger Z U_n | 0^n \rangle \geq 0\} \tag{14}$$

that is outside BPP but inside BQP. The authors then consider a classically easy language $L_{\text{easy}} \in$ BPP and assume that for each input size $n$, there exists an input $a_n \in L_{\text{easy}}$ and an input $b_n \notin L_{\text{easy}}$. Then it is possible to define a new language:

$$L = \bigcup_{n=1}^\infty \{x \mid \forall x \in L_{\text{easy}}, \ 1^n \in L_{\text{hard}}, \ |x| = n\} \cup \{x \mid \forall x \notin L_{\text{easy}}, \ 1^n \notin L_{\text{hard}}, \ |x| = n\}. \tag{15}$$

For each size $n$, if $1^n \in L_{\text{BQP}}$, $L$ would include all $x \in L_{\text{easy}}$ with $|x| = n$. If $1^n \notin L_{\text{hard}}$, $L$ would include all $x \notin L_{\text{easy}}$ with $|x| = n$. By definition, if we can determine whether $x \in L$ for an input $x$ using a classical algorithm (BPP), we could also determine whether $1^n \in L_{\text{BQP}}$ by checking whether $x \in L_{\text{easy}}$. However, this must be impossible as we are assuming that $L_{\text{BQP}}$ cannot be decided classically. Hence, the language $L$ is not in BPP. On the other hand, for every size $n$, a classical machine learning algorithm can use a single training data point $(x_0, y_0)$ to decide whether $x \in L$ by what said above.

$\qquad\square$

We note here that the existence of unary languages in BQP but not in BPP is a well believed assumption.

### D.2 Proofs of the theorems from the main text concerning the hardness of random generatability

In this Section we provide the proves about hardness of random generability based on the fact that BQP is not in the $\mathsf{P}^{\mathsf{NP}}$ or $\mathsf{HeurBPP}^{\mathsf{NP}}$. We provide here an intuition on how the proofs work. Suppose there exists an efficient classical algorithm that samples pairs $(x, f(x))$ at random, where $x \, \mathcal{D}$ for some $\mathcal{D}$. Then, for every sampled $(x_r, f(x_r))$ there must exist a random string $r$ such that the algorithm, when run with randomness $r$, outputs that pair. Given any input $x$, we could then search for such an $r$ using an NP oracle (since the algorithm is efficient), and once the correct $r$ is found, we could evaluate the randomized algorithm on that random string to obtain $f(x)$. This would imply that $f(x)$ is classically computable with the help of an NP oracle, contradicting our complexity assumptions.

### D.2.1 HARDNESS OF EXACT RANDOM GENERATABILITY

**Theorem 1** (Exact Random generatability implies $\mathsf{BQP} \subseteq \mathsf{P}^{\mathsf{NP}}$). *Let $f = \{f_n\}_{n \in \mathbb{N}}$ be a family of $\mathsf{BQP}$ functions as in Def. 10. If there exists a classical randomized poly-time uniform algorithm that generates samples $(x, f_n(x))$ correctly with probability 1 as in Def.1, with $x \sim \mathsf{Unif}(\{0,1\}^n)$, then $\mathsf{P}^{\mathsf{NP}}$ contains $\mathsf{BQP}$.*

*Proof.* The existence of a randomized poly-time algorithm which satisfies Theorem 1 is equivalent to the existence of a uniform family of poly-sized algorithms $C(r)$ such that a random choice of $r \in \{0,1\}^{\mathrm{poly}(n)}$ outputs a tuple $(x, f(x))$ uniformly at random, i.e. :

$$\mathcal{C} = \{C(r) \mid r \in \{0,1\}^{\mathrm{poly(n)}}\} \tag{16}$$

Then, for every $x$ there exists at least one $r_x$ such that $C(r_x) = (x, f(x))$. In particular, for a given $x$ a $\mathsf{P}^{\mathsf{NP}}$ machine can in polynomial time find a corresponding $r_x$. Consider the set

$$P(\tilde{C}) = \{(u,x) \mid \exists v \in \{0,1\}^* \text{ s.t. } \tilde{C}(uv) = x\} \tag{17}$$

where $\tilde{\mathcal{C}}$ is the same family of algorithms $\mathcal{C}$ except that the last bit of the output (i.e., the value $f(x)$) is omitted and $\tilde{C}(r) \in \tilde{\mathcal{C}}$. Then $(u,x)$ is in $P(\tilde{C})$ if and only if $u$ is a prefix of some $r_x$ inverse of $x$ with respect to $\tilde{C}$. Clearly $P(\tilde{C})$ is in NP as a correct $u \in \{0,1\}^*$ can be verified in polynomial time as also the family $\tilde{\mathcal{C}}$ consists of polynomial sized algorithms. Then we can run the following algorithm in $\mathsf{P}^{\mathsf{NP}}$:

---
**Algorithm 1** Prefix Search Algorithm
---
1: **function** PREFIXSEARCH($x$)
2:      $u \leftarrow \varepsilon$;                                       ▷ where $\varepsilon$ denotes the empty string
3:      **for** $|r|$ times **do**
4:          **if** $(u1, x) \in P(\tilde{C})$ **then** $u \leftarrow u1$ **else** $u \leftarrow u0$
5:          **end if**
6:      **end for**
7:      **return** $u$
8: **end function**

---

As for every $x$ there exists at least one corresponding $r_x$, the above algorithm always succeeds in finding a correct $r_x$. Once the string $r_x$ is found, the $P$ machine can run $C$ on that string and evaluate $f(x)$. Since $f(x)$ can decide an arbitrary BQP language by Def. 10, it follows that $\mathsf{P}^{\mathsf{NP}}$ can also correctly decide every $x \in \{0,1\}^n$.

$\square$

### D.2.2 HARDNESS OF APPROXIMATE RANDOM GENERATABILITY

In this and the following sections, our proofs concerning the hardness of the approximate random generatability and of the identification task will rely on a well-known result about sampling witnesses of an NP relation using an NP oracle. Specifically, the authors of Bellare et al. (2000) showed that for a given NP language $L$ and relation $R$, where $L = \{x \mid \exists w \text{ such that } R[x,w] = 1\}$, it is possible to uniformly sample witnesses $w$ from the set $R_x = \{w \mid R[x,w] = 1\}$ for any $x \in L$. Since this result will be central to all our subsequent proofs, we include the main theorem from Bellare et al. (2000) here for reference.

**Theorem 8** (Theorem 3.1 in Bellare et al. (2000)). *Let $R$ be an NP-relation. Then there is a uniform generator for $R$ which is implementable in probabilistic, polynomial time with an NP-oracle.*

For convenience, we also report here the definition of approximate random generatability of Def. 2 in the main text.

**Definition 17** (Approximate random generatability (Def. 2 in the main text).). *Let $f = \{f_n\}_{n \in \mathbb{N}}$ be a uniform family of functions, with $f_n : \{0,1\}^n \rightarrow \{0,1\}$. $f$ is approximately random generatable under the distribution $\mathcal{D}$ if there exists an efficient non-uniform (randomized) algorithm $\mathcal{A}$ such that given as input a description of $f_n$ for any $n$, a random string $r$ and an error value $\epsilon$, outputs:*

$$\mathcal{A}(f_n, \epsilon, r) = (x_r, f_n(x_r)) \tag{18}$$

with $(x_r, f_n(x_r)) \sim \pi_\epsilon^f(\mathcal{D})$ when $r$ is sampled uniformly at random from the distribution of all the random strings. In particular, $\pi_\epsilon^f(\mathcal{D})$ is a probability distribution over $\{0,1\}^n \times \{0,1\}$ which satisfies: $\|\pi_\epsilon^f(\mathcal{D}) - \pi^f(\mathcal{D})\|_{TV} \leq \epsilon$, where $\pi^f(\mathcal{D})$ is the same distribution defined in Def. 1.

We can now state our hardness result for the approximate case of random generatability.

**Theorem 2** (Approximate Random generatability implies $(\mathsf{BQP}, \mathsf{Unif}) \in \mathsf{HeurBPP}^{\mathsf{NP}}$). *Let $f = \{f_n\}_{n\in\mathbb{N}}$ be a family of $\mathsf{BQP}$ functions which is the characteristic function of a language $\mathcal{L} \in \mathsf{BQP}$ as in Def. 10, and let $\mathsf{Unif}$ be the uniform distribution over $\{0,1\}^n$. If there exists a polynomial time algorithm $\mathcal{A}$ which satisfies Def. 2 for the uniform input distribution $\mathsf{Unif}$, then $(\mathcal{L}, \mathsf{Unif}) \in \mathsf{HeurBPP}^{\mathsf{NP}}$.*

*Proof.* The proof follows from the proof of Theorem 1. We will present the proof under the assumption of a distribution $\mathsf{Unif}_{\mathcal{L}_P}$, as defined in Def. 14, and a PromiseBQP-complete language $\mathcal{L}_P$ such that $(\mathcal{L}_P, \mathsf{Unif}_{\mathcal{L}_P}) \nsubseteq \mathsf{HeurBPP}^{\mathsf{NP}}$. The proof then generalizes straightforwardly to the case where we assume $(\mathcal{L} \in \mathsf{BQP}, \mathsf{Unif}) \nsubseteq \mathsf{HeurBPP}^{\mathsf{NP}}$, simply by considering the uniform distribution $\mathsf{Unif}$ over all inputs $x \in 0, 1^n$ instead of $\mathsf{Unif}_{\mathcal{L}_P}$.

Consider the same families of algorithms $\mathcal{C}$ and $\tilde{\mathcal{C}}$ introduced in the proof of Theorem 1. The existence of an algorithm $\mathcal{A}$ as in Def. 2 guarantees the existence of such families $\mathcal{C}$ and $\tilde{\mathcal{C}}$. Let $W = \{r_i\}_{i=1}^M$ be a set of $M$ random strings $r_i \in \{0,1\}^{|r|}$, with $|r| = \mathrm{poly}(n)$ and consider the set:

$$P(\tilde{C}, W) = \{x \in \{0,1\}^n \mid \exists r \in \{0,1\}^{|r|} \text{ s.t. } \tilde{C}(r) = x \ \& \ r \notin W\} \tag{19}$$

If the set $W$ has a size $M = \mathrm{poly}(n)$ then $P(\tilde{C}, W)$ can be decided in $\mathsf{NP}$ since both the conditions $\tilde{C}(r) = x$ and $r \notin W$ can be verified in polynomial time. Theorem 8 from Bellare et al. (2000) guarantees the existence of an algorithm $A_R$ such that, given the NP relation $R : \tilde{C}(r) = x$ and an input $x$, outputs, with high probability, a sample $r$ such that $\tilde{C}(r) = x$ uniformly at random among all the witnesses of $x$. Then the following algorithm runs in $\mathrm{P}^{\mathrm{NP}}$.

---

```
 1: function MULTIPREFIXSEARCH(x)
 2:     r ← ε                                         ▷ where ε denotes the empty string
 3:     W = {r}
 4:     for poly(|x|) times do
 5:         r ← A_R[x]
 6:         W = W + {r}
 7:     end for
 8:     for i : 1 < i < |W| times do
 9:         return r_i ∈ W
10:     end for
11: end function
```

---

Where we denote as $A_R[x]$ the algorithm in Bellare et al. (2000) which uniformly samples witnesses for the relation $R : \tilde{C}(r) = x$ corresponding to the input $x$. On an input $x$, the algorithm `MultiPrefixSearch` outputs a list of polynomial different random strings $\{r_x^i\}_i$ (if they exists) for which $\tilde{C}(r_x^i) = x \ \forall i$, sampled uniformly at random among all the random strings associated to $x$. We now construct the following algorithm $\mathcal{A}'$ acting on input $x$. Specifically, on any $x \in \{0,1\}^n$, $\mathcal{A}'$ first applies the algorithm `MultiPrefixSearch(x)`, then either:

- If `MultiPrefixSearch(x)` outputs an empty string, then the algorithm assigns a random value to $f(x)$

- Otherwise, the algorithm computes the lables $\mathcal{A}(r_x^i) = (x, f(x))_{r_x^i}$ for any of the random strings outputted by `MultiPrefixSearch(x)`. It then assigns to $x$ the label determined by the majority vote among all the $f(x)_{r_x^i}$ values obtained.

Now we show that the above algorithm $\mathcal{A}'$ is able to classify the language $\mathcal{L}$ correctly on average with respect to the uniform input distribution. More precisely, for each $x$ consider the sets $R_x^T =$

$\{r_x \mid \mathcal{A}(r_x) = (x, y) \text{ with } y = f(x)\}$ and $R_x^F = \{r_x \mid \mathcal{A}(r_x) = (x, y) \text{ with } y = f(x) \oplus 1\}$. Also, let $R_{\text{tot}}$ be the set of all the random strings of the algorithm $\mathcal{A}$, clearly $|R_{tot}| = \sum_{x \in \{0,1\}^n} |R_x^T| + |R_x^F|$. Notice that $|R_x^T|/|R_{tot}|$ precisely represents the probability that the classical algorithm $\mathcal{A}$ outputs the pair $(x, f(x))$. Similarly, the probability that $\mathcal{A}$ outputs a tuple containing $x$ is $|R_{tot}(x)|/|R_{tot}|$ where $|R_{tot}(x)| = |R_x^T| + |R_x^F|$.

By definition of difference in total variation between two distributions, i.e. $\|p-q\|_{TV} = \frac{1}{2} \sum_x |p(x) - q(x)|$, we have the following relation between the distribution $\pi_\epsilon$ generated by $\mathcal{A}$ and the target distribution $\text{Unif}_{\mathcal{L}_P}(\{0,1\}^n) \times f(\text{Unif}_{\mathcal{L}_P}(\{0,1\}^n))$ uniform over the input of the promise of the language $\mathcal{L}$:

$$\|\pi_\epsilon - \text{Unif}_{\mathcal{L}_P}(\{0,1\}^n) \times f(\text{Unif}_{\mathcal{L}_P}(\{0,1\}^n))\|_{TV} = \frac{1}{2} \sum_{x \in P} \left| \frac{|R_x^T|}{|R_{\text{tot}}|} - \frac{1}{|P|} \right| + \frac{1}{2} \sum_{x \in P} \frac{|R_x^F|}{|R_{tot}|}$$
$$+ \frac{1}{2} \sum_{x \notin P} \frac{|R_x^T| + |R_x^F|}{|R_{tot}|}, \tag{20}$$

where $P = \mathcal{L}_{\text{yes}} \cup \mathcal{L}_{\text{no}} \subset \{0,1\}^n$ denotes the set of $x$ belonging to the promise of for the language $\mathcal{L}$. We now bound the number of $x \in P$ for which the probability of $\mathcal{A}$ computing incorrectly them exceeds $1/3$, specifically we are interested in the size of the set $X_{\text{wrong}}$ such that:

$$X_{\text{wrong}} = \left\{ x \in P \mid \frac{|R_x^T|}{|R_{\text{tot}}(x)|} \leq \frac{2}{3} \right\}. \tag{21}$$

From Eq. (20), it follows:

$$\|\mathcal{L}_\epsilon - \text{Unif}(\{0,1\}^n) \times f(\text{Unif}(\{0,1\}^n))\|_{TV} \geq \frac{1}{2} \sum_{x \in P} \left| \frac{|R_x^T|}{|R_{\text{tot}}|} - \frac{1}{|P|} \right| \tag{22}$$

$$\geq \frac{1}{2} \sum_{x \in X_{\text{wrong}}} \left| \frac{\frac{2}{3}|R_{\text{tot}}(x)|}{|R_{\text{tot}}|} - \frac{1}{|P|} \right| \tag{23}$$

$$\sim \frac{1}{2} \sum_{x \in X_{\text{wrong}}} \left| \frac{2}{3}\frac{1}{|P|} - \frac{1}{|P|} \right| \tag{24}$$

$$= \frac{1}{2} \frac{|X_{\text{wrong}}|}{3 \cdot |P|}, \tag{25}$$

where we used the fact that as the output distribution $\mathcal{L}_\epsilon$ is close in TV with the uniform distribution over the $x \in \{0,1\}^n$ (and the corresponding labels $y = f(x)$), then $\frac{|R_{\text{tot}}(x)|}{|R_{\text{tot}}|} = \frac{1}{|P|} + \delta_x$ with the constraint $\sum_x |\delta_x| \leq \epsilon$. We can then assume $\delta_x << \epsilon$, and therefore neglect it in the derivation above. From Eq. (25), it directly follows that $|X_{\text{wrong}}| \leq 6\epsilon \cdot |P|$ and therefore the fraction of $x \in |P|$ for which $\mathcal{A}$ outputs the correct label with a probability less than $2/3$ is approximately $6\epsilon$.

Then, the algorithm $\mathcal{A}'$, with a NP oracle, correctly decides in BPP at least a fraction $1 - 6\epsilon$ of all the possible $x \in \{0,1\}^n$. When $x$ are sampled uniformly, algorithm $\mathcal{A}'$ is therefore able to decide the language $\mathcal{L}$ in HeurBPP$^{\text{NP}}$ with respect to the distribution $\text{Unif}_{\mathcal{L}_P}$ uniform over the set of $x$ belonging to the promise.

$\square$

**Corollary 2.** *The expectation value sampling (EVS) generative model, as defined in Barthe et al. (2024) is classically intractable.*

*Proof.* To not detract from the main topic we just provide a quick proof sketch. By intractable we mean that there cannot exist a classical algorithm which takes on input an arbitrary polynomial sized EVS $M$ specified by its parametrized circuit (see Barthe et al. (2024)) and outputs an $\epsilon$ approximation of the distribution outputted by $M$. To prove this we note that for any BQP function $f : \{0,1\}^n \to \{0,1\}$ we can construct a poly-sized EVS which produces samples of the form $(x, f(x))$, where $x$ is an $n$-bit bitsting sampled (approximately) uniformly (or according to any

desired classically samplable distribution). There are many ways to realize this: the easiest is to coherently read out the bits of input random continuous-valued variable using phase estimation (for this, this variable $x$ is uploaded using prefactors $2^k$ for each bit position $k$), forwards this to part of the output, and in a parallel register computes $f(x)$ and outputs that as well. This is a valid EVS, and sampling from it is equivalent to random generatability for the BQP function $f$. This then implies $(\mathcal{L}_P, \mathsf{Unif}_{\mathcal{L}_P}) \in \mathsf{HeurBPP}^{\mathsf{NP}}$. $\qquad\square$

We provide here an example of a $0.5$-distinct concept class, which therefore satisfies the condition of Theorem 13, where the concepts are all $(\mathsf{Promise}-)\mathsf{BQP}$ complete functions.

**Definition 18** ($0.5$-distinct concept class with BQP concepts)**.** Let $f$ be a PromiseBQP complete function as in Def. 13 on input $x \in \{0,1\}^n$. Then for any $f$ there exists a concept class $\mathcal{F}$ containing $2^n$ PromiseBQP-complete concepts defined as:

$$\mathcal{F} = \{f^\alpha : \{0,1\}^n \to \{0,1\} \mid \alpha \in \{0,1\}^n\} \tag{26}$$
$$f^\alpha : x \to f(x) \oplus (\alpha \cdot x) \tag{27}$$

where $\alpha \cdot x$ is the bitwise inner product modulo 2 of the vectors $\alpha$ and $x$.

Clearly for any $\alpha_1, \alpha_2 \in \{0,1\}^n$, $\alpha_1 \neq \alpha_2$, it holds that $f^{\alpha_1}(x) \neq f^{\alpha_2}(x)$ on exactly half of the input $x$, also they are all PromiseBQP complete functions as by evaluating one of them we can easily recover the value of $f(x)$.

# E  PROOFS OF THE THEOREMS CONCERNING THE IDENTIFICATION TASK

## E.1  PROOF OF HARDNESS FOR THE IDENTIFICATION TASK IN THE VERIFIABLE CASE

**Theorem 3** (Identification hardness for the verifiable case )**.** *Let* $\mathcal{F} = \{f^\alpha : \{0,1\}^n \to \{0,1\} \mid \alpha \in \{0,1\}^m, m = \mathrm{poly}(n)\}$ *be a concept class such that there exists a* $f^\alpha \in \mathcal{F}$ *which is the characteristic function of a language* $\mathcal{L} \in \mathsf{BQP}$ *as in Def. 10, and let* $\mathsf{Unif}$ *be the uniform distribution over the* $x \in \{0,1\}^n$. *If there exists a verifiable identification algorithm* $\mathcal{A}_B$ *for the uniform input distribution* $\mathsf{Unif}$ *as given in Def. 7, then* $(\mathcal{L}, \mathsf{Unif}) \in \mathsf{HeurBPP}^{\mathsf{NP}}$.

*Proof.* In this proof, we consider any training set $T = \{(x_\ell, y_\ell)\}_{\ell=1}^B$ as a sequence of concateneted bitstrings, i.e. $T = \underbrace{x_1 y_1 x_2 y_2 ... x_B y_B}_{B(n+1) \text{ bits}}$. Let $X = \{x_\ell\}_{\ell=1}^B$ be a set of $B$ inputs $x_\ell \in \{0,1\}^n$. We define the following set:

$$P(\mathcal{A}_B, \epsilon, B) = \{(X, \alpha) \mid \exists Y \in \{0,1\}^B, r \in \{0,1\}^{\mathrm{poly}(n)} \text{ s.t. } \mathcal{A}_B(T_{X,Y}, \epsilon, r) = \alpha \}, \tag{28}$$

where $Y = \{y_\ell\}_{\ell=1}^B$ is a collection of labels $y_\ell \in \{0,1\}$ such that $T_{X,Y} = \{(x_\ell, y_\ell)\}_{\ell=1}^B$. We are also going to assume that for a given $\alpha$ and $\epsilon$, the number of training sets for which there exists an $r$ such that $\mathcal{A}_B(T, \epsilon, r) = \alpha$ is greater than 0 and the majority of them are dataset labeled accordingly to one concept $\tilde{\alpha}$ which is $\epsilon$-close to $\alpha$ in PAC condition. The set $P(\mathcal{A}_B, \epsilon, B)$ contains then all the tuples $(X, \alpha)$ which satisfy the relation $R : \mathcal{A}_B(T_{X,Y}, \epsilon, r) = \alpha$. Since $\mathcal{A}_B$ runs in polynomial time (for $\epsilon \sim \frac{1}{\mathrm{poly}(n)}$ and $B \sim \mathrm{poly}(n)$), the relation $R$ belongs to NP. Theorem 8 from Bellare et al. (2000) guarantees the existence of an algorithm $A_R$ such that given the NP relation $R : \mathcal{A}_B(T_{X,Y}, \epsilon, r) = \alpha$ outputs on input $(X, \alpha)$ a tuple $(Y, r) = A_R[(X, \alpha)]$ which satisfies $\mathcal{A}_B(T_{X,Y}, \epsilon, r) = \alpha$, uniformly at random among all the possible tuples $(Y, r)$ which satisfies $R$.

We can now construct an algorithm $\mathcal{A}'$ which, using a NP oracle and the algorithm $A_R$ in Bellare et al. (2000), evaluates any concept $f^\alpha$ with an average-case error of at most $\epsilon'$ under the uniform distribution over inputs $x \in \{0,1\}^n$. In other words, the algorithm $\mathcal{A}'$ satisfies satisfies the heuristic condition in Eq. (10) with respect the target function $f^\alpha$. Firstly, since we are considering the verifiable case, we can implement the following function which, given a dataset consistent with one of the concept, tells if the label of an input $x \in \{0,1\}^n$ is consistent with the dataset or not.

---

1: **function** CHECK($x, \epsilon, T$)
2:     **if** $\mathcal{A}_B(\{(x,0)\} \cup T, \epsilon, r_0)$ =“invalid dataset”
3:         **return** 1
4:     **else return** 0
5: **end function**

---

We can now construct the algorithm $\mathcal{A}'$. Fix a desired average-case error $\epsilon'$. First consider an identification algorithm $\mathcal{A}_B$ in Def. 7 which runs with $\epsilon$ and $\delta$ such that $\epsilon = \frac{\epsilon'}{2}(1 - \delta)$. Using the algorithm $\mathcal{A}_B$ and the algorithm $R$ given in Bellare et al. (2000), we can construct the following algorithm $\mathcal{A}'$ which runs in polynomial time using an NP oracle.

---

1: **function** $\mathcal{A}'(\alpha, x, \epsilon, B)$
2:     **step 1.** Sample $B$ inputs $X = x_1, x_2, ..., x_B$ from the uniform distribution Unif.
3:     **step 2.** $(Y, r) \leftarrow A_R[(X, \alpha)]$
4:     **step 3.** Construct $T_{X,Y}$ by assigning each label in $Y$ to the corresponding point in $X$.
5:     **step 4. return** $y \leftarrow$ CHECK($x, \epsilon, T_{X,Y}$).
6: **end function**

---

The labels produced by $\mathcal{A}'$ ensure that all inputs in $\{(x, y)\} \cup T_{X,Y}$ are consistent with at least one concept labeled by $\tilde{\alpha}$. We will now demonstrate that the concept $f^{\tilde{\alpha}}$ is, with probability greater than $2/3$, heuristically close to the target $\alpha$, meaning that:

$$\mathbb{E}_{x \sim \mathrm{Unif}(\{0,1\}^n)} ||f^{\tilde{\alpha}}(x) - f^\alpha(x)|| \leq \epsilon'. \tag{29}$$

In Step 2 of the algorithm $\mathcal{A}'$, the labeling $Y = \{y_\ell\}_{\ell=1}^B$ is chosen such that $\mathcal{A}_B$, given the dataset $T_{X,Y}$ and parameter $\epsilon$, outputs $\alpha$ with a probability greater than 0. Specifically, the dataset is sampled uniformly at random from all possible datasets satisfying this condition. The probability that an outputted dataset $T_{X,Y}$ is consistent with a concept $f^{\tilde{\alpha}}$ that is not $\epsilon'$-close to $f^\alpha$ (as defined by Eq. (29)) depends on both the inputs in $X$ and the failure probability $\delta$ of the algorithm $\mathcal{A}_B$. For simplicity, we initially neglect the failure probability $\delta$ of $\mathcal{A}_B$. Even in this case, the dataset $T_{X,Y}$ could still be consistent with a concept $f^{\tilde{\alpha}}$ far from $f^\alpha$ if both $f^\alpha$ and $f^{\tilde{\alpha}}$ agree on all the $B$ inputs in $X$. By setting $\epsilon = \epsilon'/2$, the fraction of inputs on which a concept $f^{\tilde{\alpha}}$ (not satisfying Eq. (29)) disagrees with a concept that is $\epsilon$-close to $f^\alpha$ is at least $\epsilon$. If at least one of those inputs is in $X$, then such $f^{\tilde{\alpha}}$ cannot be outputted by $R[P(\mathcal{A}_B, \epsilon, B), (X, \alpha)]$ (we are still assuming the algorithm $\mathcal{A}_B$ has a zero failure probability, i.e. $\delta = 0$). The probability that a random $x$ lies in a fraction $\epsilon$ of all the $x \in \{0,1\}^n$ is clearly $\epsilon$. It follows that the probability of selecting $B$ random inputs where at least one shows a disagreement between a concept that is $\epsilon$-close to $f^\alpha$ and a concept that is more than $\epsilon'$ away from $f^\alpha$ is:

$$1 - (1 - \epsilon)^B \tag{30}$$

We want to bound this probability such that is above $\frac{2}{3}$. In order to obtain that, we need to extract a number $B$ of inputs greater than:

$$(1 - \epsilon)^B \leq \frac{1}{3} \tag{31}$$

$$B \log(1 - \epsilon) \leq -\log 3 \tag{32}$$

$$-2\epsilon B \leq -\log 3 \tag{33}$$

$$B \geq \frac{\log 3}{2\epsilon} \tag{34}$$

Where we used the fact that for $\epsilon \leq 0.7$, $-2\epsilon \leq \log(1 - \epsilon)$. Therefore, by selecting $B \sim \frac{1}{\epsilon}$ (noting that the $\epsilon$ has to scale as $\epsilon \sim \frac{1}{n}$ for the algorithm $A$ to remain efficient), we can ensure with a probability greater than $2/3$ that the dataset $T_{X,Y}$ produced in Step 2 of $\mathcal{A}'$ is consistent only with concepts that are $\epsilon'$-close to the target $\alpha$. We now take into account the failure probability of the algorithm $\mathcal{A}_B$. Because of this, it is still possible that there exists a random string $r_0$ such that, for a dataset $T_{X,Y}$ consistent with a concept $f^{\tilde{\alpha}}$ that is more than $\epsilon'$ away from $f^\alpha$, the algorithm

$A(T_{X,Y}, \epsilon, r_0)$ outputs $\alpha$, regardless of the $B$ inputs in the dataset. This happens with a probability $\delta$ over all the possible datasets. We can then take into account this probability of failure by asking that the probability in Eq. (30) is above $\frac{2}{3(1-\delta)}$. With this modification, the bound on the number of inputs $B$ becomes:

$$B \geq \frac{\log 3 + 2\delta}{\epsilon} \tag{35}$$

Notice that for $\delta, \epsilon \approx \frac{1}{n}$ the above quantity is still polynomial in $n$.

For every input $x$, the algorithm $\mathcal{A}'$ outputs, with probability greater than $\frac{2}{3}$, a label $y \in \{0, 1\}$ that aligns with one of the concepts $f^{\tilde{\alpha}}$ satisfying Eq. (29), i.e., concepts that are $\epsilon'$-close to $f^\alpha$. The maximum number of inputs $x \in \{0, 1\}^n$ that the algorithm can consistently misclassify is bounded above by $\epsilon'$. This corresponds to the scenario where all $\epsilon'$-close concepts disagree with $\alpha$ on the same subset of inputs.

$\square$

As discussed in Appendix B.2, we state here the second version of our result on the hardness of the verifiable identification task. In this version, we assume the existence of an input distribution $\mathsf{Unif}_{\mathcal{L}_P}$ as in Def. 14 and a PromiseBQP language $\mathcal{L}_P$ such that $(\mathcal{L}_P, \mathsf{Unif}_{\mathcal{L}_P}) \not\subseteq \mathsf{HeurBPP}^{\mathsf{NP}}$.

**Theorem 9** (Identification hardness for the verifiable case - second version). *Let $\mathcal{F} = \{f^\alpha : \{0, 1\}^n \to \{0, 1\} \mid \alpha \in \{0, 1\}^m, m = \mathrm{poly}(n)\}$ be a concept class such that there exists a $f^\alpha \in \mathcal{F}$ which is the characteristic function of a language $\mathcal{L}_P \in \mathsf{PromiseBQP}$ as in Def. 10, and let $\mathsf{Unif}_{\mathcal{L}_P}$ be the input distribution defined in Def. 14. If there exists a verifiable identification algorithm $\mathcal{A}_B$ for the input distribution $\mathsf{Unif}_{\mathcal{L}_P}$ as given in Def. 7, then $(\mathcal{L}_P, \mathsf{Unif}_{\mathcal{L}_P}) \in \mathsf{HeurBPP}^{\mathsf{NP}}$.*

*Proof.* The proof proceeds in direct analogy to the proof of Theorem 9. The only change regards the first step in Algorithm $\mathcal{A}'$. Specifically, instead of sampling the inputs $X = x_1, x_2, ..., x_B$ uniformly at random, now the algorithm $\mathcal{A}'$ will sample them from the distribution $\mathsf{Unif}_P$. We note that, by Def. 14, $\mathsf{Unif}_P$ is assumed to be classically efficiently samplable. $\square$

# F    Proofs of hardness for the identification task in the non-verifiable case

In this section we provide the full proof of our main result in Theorem 10. As outlined in the proof sketch in the main text, the proof is a combination of two other theorems that we also state and prove here. We first restate the main theorem, including both the possible assumptions- average case hardness of evaluating BQP or PromiseBQP languages- together.

**Theorem 10** (Hardness of the identification task - non verifiable case). *Let $\mathcal{F} = \{f^\alpha : \{0, 1\}^n \to \{0, 1\} \mid \alpha \in \{0, 1\}^m\}$ be a concept class containing at least a $\mathsf{BQP}$ (PromiseBQP) function $f$ as in Def. 10 (as in Def. 13) associated to a language $\mathcal{L} \in \mathsf{BQP}$ ($\mathcal{L}_P \in \mathsf{PromiseBQP}$). Assume further that $\mathcal{F}$ is either a $c$-distinct concept class with $c \geq 1/3$ or an average-case-smooth concept class. If the non-verifiable identification task for $\mathcal{F}$, as defined in Def. 8, is solvable by an approximate-correct identification algorithm $\mathcal{A}_B$ (Def. 19), then it follows that $(\mathcal{L}, \mathsf{Unif}) \in \mathsf{HeurBPP}^{\mathsf{NP}^{\mathsf{NP}^{\mathsf{NP}}}}$ (respectively, $(\mathcal{L}_P, \mathsf{Unif}_{\mathcal{L}_P}) \in \mathsf{HeurBPP}^{\mathsf{NP}^{\mathsf{NP}^{\mathsf{NP}}}}$).*

*Proof.* The proof combines the intermediate result in Theorem 11 (or Theorem 12) with the result in Theorem 13 for the $c$-distinct concept classes or in Theorem 14 for average-case-smooth concept classes. Specifically, Theorems 13 and 14 show that if an approximate-correct algorithm exists for a concept class $\mathcal{F}$ satisfying Def. 5 or Def. 6, then there exists an approximate-verifiable algorithm for $\mathcal{F}$ running in NP. Therefore, as a consequence of the intermediate result in Theorem 11 (Theorem 12), it follows that if such approximate-verifiable algorithm would exist, then $(\mathcal{L}, \mathsf{Unif}) \in \mathsf{HeurBPP}^{\mathsf{NP}^{\mathsf{NP}^{\mathsf{NP}}}}$ $((\mathcal{L}_P, \mathsf{Unif}_{\mathcal{L}_P}) \in \mathsf{HeurBPP}^{\mathsf{NP}^{\mathsf{NP}^{\mathsf{NP}}}})$. This comes from selecting $\mathsf{A} = \mathsf{NP}$ in Theorem 11 $\square$

In the following two subsections we will provide the proof of the results used in the proof of Theorem 10.

### F.1 Hardness of approximate-verifiable identification tasks

We first present an intermediate result, which concerns the hardness of the approximate-verifiable identification task. We define the approximate-verifiable task an identification task where the learning algorithm can reject datasets which contains more than a fraction $\beta$ of inputs incorrectly labeled.

**Definition 19** (Identification task - approximate-verifiable case)**.** Let $\mathcal{F} = \{f^\alpha : \{0,1\}^n \rightarrow \{0,1\} \mid \alpha \in \{0,1\}^m\}$ be a concept class. An *approximate-verifiable* identification algorithm is a (randomized) algorithm $\mathcal{A}_B^\beta$ such that when given as input a set $T = \{x_\ell, y_\ell\}_{\ell=1}^B$ of at least $B$ pairs $(x,y) \in \{0,1\}^n \times \{0,1\}$, an error parameter $\epsilon > 0$ and a random string $r \in R$, it satisfies the two following properties:

- **Soundness** If $\mathcal{A}_B^\beta(T, \epsilon, r) = \alpha$ then there exists a subset $T' \subset T$, with $|T'| \geq (1 - \frac{1}{\beta})B$ such that $y_\ell = f^\alpha(x_\ell)$ for all the $(x_\ell, y_\ell) \in T'$. In case there does not exist a subset $T' \subset T$ of $(1 - \frac{1}{\beta})B$ inputs consistent with any one of the concepts, then $\mathcal{A}_B^\beta$ outputs *"invalid dataset"*.

- **Completeness** If $T = \{(x_\ell, y_\ell)\}_\ell$ and $y_\ell = f^\alpha(x_\ell)$ for all $(x_\ell, y_\ell) \in T'$, then, for any $\epsilon \geq \frac{1}{3}$ and $\beta \geq 0$, the following condition holds:

$$\exists r \text{ s.t. } \mathcal{A}_B^\beta(T, \epsilon, r) = \alpha.$$

In addition to being a proper PAC learner: if the samples in $T$ come from the distribution $\mathcal{D}$, i.e. $x_\ell \sim \mathcal{D}$, and there exists an $\alpha \in \{0,1\}^m$ such that $T = \{(x_\ell, y_\ell)\}_{\ell=1}^B$, with $x_\ell \sim \mathcal{D}$ and $y_\ell = f^\alpha(x_\ell) \in \{0,1\}$ then with probability $1 - \delta$ outputs:

$$\mathcal{A}_B(T^\alpha, \epsilon, r) = \tilde{\alpha}, \quad \tilde{\alpha} \in \{0,1\}^m, \tag{36}$$

with the condition $\mathbb{E}_{x \sim \mathcal{D}}|f^\alpha(x) - f^{\tilde{\alpha}}(x)| \leq \epsilon$.

We say that $\mathcal{A}_B$ solves the identification task for a concept class $\mathcal{F}$ under the input distribution $\mathcal{D}$ if the algorithm works for any value of $\epsilon, \delta \geq 0$. The success probability $1 - \delta$ is over the training sets $T$ where the input points are sampled from the distribution $\mathcal{D}$ and the internal randomization of the algorithm. The required minimum size $B$ of the input set $T$ is assumed to scale as poly($n, 1/\epsilon, 1/\delta$), while the running time of the algorithm scales as poly($B, n, \beta$).

In the following theorem we prove hardness of the approximate-verifiable identification task. It will be useful to state our result assuming that the approximate-verifiable algorithm lies in a generic complexity class A and obtain hardness based on the assumption that BQP (or PromiseBQP in the second version of the theorem) is not in $\mathsf{HeurBPP}^{\mathsf{NP}^{\mathsf{NP}^{\mathsf{A}}}}$. In case A = P, and the approximate-verifiable algorithm is classically efficient, then $\mathsf{HeurBPP}^{\mathsf{NP}^{\mathsf{NP}^{\mathsf{A}}}} = \mathsf{HeurBPP}^{\mathsf{NP}^{\mathsf{NP}}}$.

**Theorem 11** (Approximate-verifiable identification implies $(\mathsf{BQP}, \mathsf{Unif}) \subset \mathsf{HeurBPP}^{\mathsf{NP}^{\mathsf{NP}}}$)**.** *Let* $\mathcal{F} = \{f^\alpha : \{0,1\}^n \rightarrow \{0,1\} \mid \alpha \in \{0,1\}^m\}$ *be a concept class containing at least a* BQP *function $f^\alpha$ as in Def. 10 associated to a language $\mathcal{L} \in$ BQP. If for any $\beta \geq 0$ there exists an approximate-verifiable identification algorithm $\mathcal{A}_B^\beta$ running in* A *for the input distribution* Unif *as in Def. 19 then* $(\mathcal{L}, \mathsf{Unif}) \in \mathsf{HeurBPP}^{\mathsf{NP}^{\mathsf{NP}^{\mathsf{A}}}}$.

*Proof.* Let the concept $f^\alpha \in \mathcal{F}$ be the BQP-function associated to a given BQP language. In this proof we are considering training sets $T_{X,Y} = \{(x_\ell, y_\ell)\}_{\ell=1}^B$ as sequences of concatenated bitstrings, i.e. $T_{X,Y} = \underbrace{x_1 y_1 x_2 y_2 ... x_B y_B}_{B(n+1) \text{ bits}}$ where $X = \{x_1, x_2, ..., x_B\}$ and $Y = \{y_1, y_2, ..., y_B\}$.

There are $2^{nB}$ possible sets $X = \{x_\ell\}_{\ell=1}^B$ of $B$ inputs $x \in \{0,1\}^n$. For any of these sets $X$, we can construct a dataset $T_{X,Y} = \{(x_\ell, y_\ell)\}_{\ell=1}^B$ by assigning a set of labels $Y = \{y_\ell\}_{\ell=1}^B$ to the inputs in $X$. We define now for each input $x \in \{0,1\}^n$ the set $T^\alpha(x)$ containing all the datasets such that:

$$T^\alpha(x) = \{T_{X,Y} = \{(x_\ell, y_\ell)\}_{\ell=1}^B \mid \exists r \text{ s.t. } \mathcal{A}_B^\beta(T_{X,Y}, r, 1/3) = \alpha \wedge x \in X\} \tag{37}$$

Because of the soundness property of the algorithm $\mathcal{A}_B^\beta$ in Def. 19, every set $T_{X,Y} \in T^\alpha(x)$ will have at least a fraction $1 - \frac{1}{\beta}$ of the $x$'s correctly labeled by $f^\alpha$. We can now construct a randomized

algorithm $M^\alpha(\mathcal{A}_B^\beta, x, s)$ which, using the approximate-verifiable algorithm $\mathcal{A}_B^\beta$ in Def. 19, on input $x$ sample one random dataset $T_{X,Y}$ from $T^\alpha(x)$ and label $x$ accordingly to the label of the corresponding $y$ in $T_{X,Y}$.

---

1: **function** $M^\alpha(\mathcal{A}_B^\beta, x, s)$
2:     **step 1.** Sample $T_{X,Y}$ from the set $T^\alpha(x)$ uniformly at random.
3:     **step 2.** Output $y$ from $(x, y) \in T_{X,Y}$
4: **end function**

---

We first show that the algorithm $M^\alpha$ belongs to $\mathsf{BPP}^{\mathsf{NP}^{\mathsf{NP}^\mathsf{A}}}$ if $\mathcal{A}_B^\beta$ runs in A. In the algorithm $M^\alpha$ the random string $s$ determines the set $T_{X,Y} \in T^\alpha(x)$ sampled. We then consider the following set:

$$P(\mathcal{A}_B, x) = \{\alpha \in \{0,1\}^m \mid \exists T_{X,Y} \text{ s.t. } T_{X,Y} \in T_\delta^\alpha(x)\}, \tag{38}$$

The set $P(\mathcal{A}_B, x)$ contains all $\alpha$ for which there exists a dataset $T_{X,Y}$ and a random string $r$ such that the identification algorithm $\mathcal{A}_B^\beta$ in Def. 19 outputs $\alpha$, i.e. $\mathcal{A}_B^\beta(T_{X,Y}, 1/3, r) = \alpha$, and $x$ appears in $T_{X,Y}$. Deciding if an $\alpha$ belongs to $P(\mathcal{A}_B^\beta, x)$ can be done in $\mathsf{NP}^{\mathsf{NP}^\mathsf{A}}$. The reason for that is that within $\mathsf{NP}^\mathsf{A}$ it is possible to decide if a given $T_{X,Y}$ belongs to $T^\alpha(x)$ since the algorithm $\mathcal{A}_B^\beta(T_{X,Y}, r, 1/3)$ runs in polynomial time. Therefore, the condition $T_{X,Y} \in T^\alpha(x)$, which ensures that $\alpha \in P(\mathcal{A}_B^\beta, x)$, can be verified in polynomial time with the help of an $\mathsf{NP}^\mathsf{A}$ oracle. We can then use Theorem 8 from Bellare et al. (2000) that guarantees the existence of an algorithm in $\mathsf{BPP}^{\mathsf{NP}}$ such that given any NP relation outputs a witness for a given $x$ uniformly at random. In particular, we apply Theorem 8 to sample a training set $T_{X,Y}$ from the set $T^\alpha(x)$ which can be regarded as witnesses for the relation $\alpha \in P(\mathcal{A}_B^\beta, x)$. This allows to perform the first step of the algorithm $M^\alpha$ in $\mathsf{BPP}^{\mathsf{NP}^{\mathsf{NP}^\mathsf{A}}}$. We have then proved that the algorithm $M^\alpha$ can be run in $\mathsf{BPP}^{\mathsf{NP}^{\mathsf{NP}^\mathsf{A}}}$.

We now show that the algorithm $M^\alpha(\mathcal{A}_B^\beta, x, s)$ can correctly evaluate the BQP function $f^\alpha$ on a large fraction of input points. In particular, we obtain that for any $\epsilon \geq 0$ the algorithm $M^\alpha$ is such that:

$$\Pr_{x \sim \mathsf{Unif}\{0,1\}^n} \left[ \Pr_s[M^\alpha(\mathcal{A}_B, x, s) \neq f^\alpha(x)] \geq \frac{2}{5} \right] \leq \epsilon. \tag{39}$$

In other words, the fraction of inputs $x \in \{0,1\}^n$ for which algorithm $M^\alpha$ misclassifies with probability greater than $2/5$ is less than $\epsilon$. Let us first consider the set $T^\alpha(x)$ for a given $x$. Because of the soundness property of the approximate-verifiable algorithm $\mathcal{A}_B^\beta$ in Def. 19, in every dataset $T_{X,Y} \in T^\alpha(x)$ the number of inputs $x \in X$ incorrectly labeled are at most $\frac{1}{\beta}B$. Since in total there can be $2^{nB}$ different sets $X$, the maximum budget for incorrectly labeled inputs across all the possible datasets is $\frac{B}{\beta}2^{nB}$. The algorithm $M^\alpha$ will misclassify a point $x$ with a probability greater than $\frac{2}{5}$ if the point appears with the incorrect label on more than $\frac{2}{5}$ of the datasets $T_{X,Y} \in T^\alpha(x)$. The total number of possible sets $X$ of $B$ points which contains $x$ is given by the total number of possible datasets minus the number of datasets which do not contains $x$, i.e.

$$2^{nB} - (2^n - 1)^B = 2^{nB} - 2^{nB}(1 - \frac{1}{2^n})^B \sim B2^{n(B-1)}. \tag{40}$$

Then, the maximum number $n_{\max}$ of different inputs $x$ for which $\Pr_s[M^\alpha(\mathcal{A}_B^\beta, x, s) \neq f^\alpha(x)] \geq \frac{2}{5}$ is

$$n_{\max}\frac{2}{5}2^{n(B-1)}B \leq \frac{1}{\beta}B2^{nB} \tag{41}$$

which gives $n_{\max} \leq \frac{5}{6\beta}2^n$ and therefore a fraction $\epsilon = \frac{5}{6\beta}$ of misclassified inputs. By choosing $\beta$ to be sufficiently small (e.g., inverse polynomial in size), we can make the error $\epsilon$ in Eq. (39) arbitrarily small.

When $x$ comes from the uniform distribution over the set of allowed inputs, this means that:

$$\Pr_{x \sim \mathsf{Unif}\{0,1\}^n} \left[ \Pr_s[M^\alpha(\mathcal{A}_B, x, s) \neq f^\alpha(x)] \geq \frac{2}{5} \right] \leq \epsilon$$

and thus $(\mathcal{L}, \mathsf{Unif}) = \mathsf{HeurBPP}^{\mathsf{NP}^{\mathsf{NP}^{A}}}$.

$\square$

As it was the case for the previous results, we can show hardness for the approximate-verifiable identification task based on the assumption that there exists a distribution $\mathsf{Unif}_{\mathcal{L}_P}$ and a PromiseBQP-complete language $\mathcal{L}_P$ not heuristically decidable given access to oracle in the PH.

**Theorem 12** (Approximate-verifiable identification implies $(\mathsf{PromiseBQP}, \mathsf{Unif}_{\mathcal{L}_P}) \subset \mathsf{HeurBPP}^{\mathsf{NP}^{\mathsf{NP}}}$)**.** *Let $\mathcal{F} = \{f^{\alpha} : \{0,1\}^n \to \{0,1\} \mid \alpha \in \{0,1\}^m\}$ be a concept class containing at least a PromiseBQP function $f$ as in Def. 13 associated to a language $\mathcal{L}_P \in \mathsf{PromiseBQP}$ and let $\mathsf{Unif}_{\mathcal{L}_P}$ be an associated input distribution as in Def. 14. If there exists an approximate-verifiable identification algorithm $\mathcal{A}_B$ running in A for the input distribution $\mathsf{Unif}_{\mathcal{L}_P}$ as in Def. 19 then $(\mathcal{L}_P, \mathsf{Unif}_{\mathcal{L}_P}) \in \mathsf{HeurBPP}^{\mathsf{NP}^{\mathsf{NP}^{A}}}$.*

*Proof.* The argument proceeds in direct analogy to the proof of Theorem 11, the only change regards the sets of trainingsets considered. In particular, we impose that the inputs $x_\ell$ in the training sets $T^\alpha(x)$ of Eq. (37) comes from the input distribution $\mathsf{Unif}_{\mathcal{L}_P}$. Since we assume efficient classical samplability of $\mathsf{Unif}_{\mathcal{L}_P}$ in Def. 14, this condition can be enforced within polynomial time. The rest of the proof follow the same steps as the proof of Theorem 11. $\square$

### F.2 CONSTRUCTION OF AN APPROXIMATE-VERIFIABLE ALGORITHM IN THE PH

We now present the second part of results needed to complete the proof of Theorem 10. Specifically, we show how to construct an approximate-verifiable algorithm which satisfies Def. 19 given an approximate-correct algorithm which satisfies the non-verifiable task in Def. 8. We will show this is possible by climbing up one layer in the PH for two types of concept classes: $c$-distinct and average-case-smooth.

**Theorem 13.** *If there exists an approximate-correct identification algorithm $\mathcal{A}_B$ as in Def. 8 for a $c$-distinct concept class $\mathcal{F}$, with $c \geq 1/3$, which works correctly for a dataset of size $B$, then there also exists an approximate-verifiable algorithm $\mathcal{A}_{\beta B}^{\beta}$ as in Def. 19 in NP which works correctly for a dataset of size $\beta B$.*

*Proof.* The proof works by explicitly constructing an algorithm $\mathcal{A}_{\beta B}^{\beta}$ that is within NP and that, given access to the approximate-correct algorithm $\mathcal{A}_B$ in Def. 8, satisfies the condition in Def. 19 of approximate-verifiable identification algorithm. In particular, the main objective of the proof is to show how an approximate-correct identification algorithm in Def. 8 can be used to construct an algorithm in NP which satisfies the completeness condition of Def. 19. We first present the main steps of the algorithm, then show that they can be performed within NP and finally check the correctness of the algorithm.

First we notice the following. If there exists a learning algorithm $\mathcal{A}_B$ which solves the identification task in Def. 8 for the concept class $\mathcal{F}$, then we can easily construct an algorithm $\bar{\mathcal{A}}_B$ that can solve the identification task even for the concept class $\bar{\mathcal{F}}$ defined as follows:

$$\bar{\mathcal{F}} = \{f^{\bar\alpha} : \{0,1\}^n \to \{0,1\} \mid \forall x \in \{0,1\}^n \;\; f^{\bar\alpha}(x) \oplus 1 = f^\alpha(x) \in \mathcal{F}\} \tag{42}$$

Specifically, the algorithm $\bar{\mathcal{A}}_B$ solves the identification task for the concept class $\bar{\mathcal{F}}$ by flipping the labels of all points in the input dataset before applying $\mathcal{A}_B$.

We are now ready to describe the algorithm $\mathcal{A}_{\beta B}^{\beta}$. We notice that having access to the algorithm $\mathcal{A}_B$ in Def. 8 we can implement the following function which, given a dataset $T = \{(x_\ell, y_\ell)\}_{\ell=1}^{\beta B}$ of $\beta B$ inputs and a label $\alpha$ corresponding to one of the concept, detects if there is a subset $T_B$ of $B$ of inputs not consistent with $\alpha$. Consider an algorithm $\mathcal{A}_B$ as in Def. 8 and let $\epsilon_c$ be $\epsilon_c = \min(\epsilon_1, \epsilon_2)$, with $\epsilon_1$ and $\epsilon_2$ the error values in the conditions of Eq. (68) and (69) of Def. 8. Then we define the `check` function as follows.

```
1: function CHECK(α, T)
2:     for T_B in T
3:         for r in R
4:             if Ā_B(T_B, ε_c, r) = ᾱ return "invalid dataset"
5: end function
```

The function `check` can be implemented by a polynomial time algorithm running in the second level of the PH. In order to prove it, consider the following set:

$$S_1(\alpha, \beta) = \{T = \{(x_\ell, y_\ell)\}_{\ell=1}^{\beta B} \mid \exists r \in R, T_B = \{(x_i, y_i)\}_{i=1}^{B}, \ T_B \subset T \text{ s.t. } \bar{A}_B(T_B, \epsilon_c, r) = \bar{\alpha} \} \tag{43}$$

The set $S_1(\alpha, \beta)$ contains all the datasets $T$ for which there exists at least one subset of $B$ inputs $T_B$ and a random string $r \in R$ such that $\bar{A}_B(T_B, \epsilon_c, r) = \bar{\alpha}$. Note that for the property of Eq. (68) in Def. 8 of approximate-correct identification algorithm, a dataset of $\beta B$ inputs entirely consistent with $\alpha$ is not contained in $S_1(\alpha, \beta)$. On a given input dataset $T = \{(x_\ell, y_\ell)\}_{\ell=1}^{\beta B}$, the algorithm `check` decides if the dataset $T$ belongs to $S_1(\alpha, \beta)$ or not. Deciding if a set $T$ belongs to $S_1(\alpha, \beta)$ can be done in NP as a YES instance can be verified in polynomial time if $\mathcal{A}_B$ runs in polynomial time.

We construct the algorithm $\mathcal{A}_{\beta B}^\beta$ as follows.

```
1: function 𝒜_{βB}^β(T, ε, r)
2:     α ← 𝒜_{βB}(T, ε, r)      ▷ with 𝒜_{βB} the approximate-correct identification algorithm in Def. 8
3:     if check(α, T) = "invalid dataset"
4:         return "invalid dataset"
5:     else return α
6: end function
```

We now show that, for $c$-distinct concept classes with $c \geq 1/3$, the algorithm $\mathcal{A}_{\beta B}^\beta$ described above rejects any dataset $T = \{(x_i, y_i)\}_{i=1}^{\beta B}$ with $B$ or more inputs not labeled by $f^\alpha$. Thus, the constructed $\mathcal{A}_{\beta B}^\beta$ satisfies the soundness condition of the approximate-verifiable algorithm as in Def. 19. First, if there exists a subset $T_B \subset T$ such that every point in it is labeled by $f^{\bar{\alpha}}$, then there exists a random string $r$ and a value $\epsilon \leq \epsilon_2$ such that $\bar{A}_B(T_B, \epsilon, r) = \bar{\alpha}'$ with the condition $\mathbb{E}_{x \sim \text{Unif}(\{0,1\}^n)} |f^{\bar{\alpha}}(x) - f^{\bar{\alpha}'}(x)| < \frac{1}{3}$. However, since by assumption all the concepts differ one from each other on more than $1/3$ of the inputs, then $\bar{A}_B(T_B, \epsilon, r) = \bar{\alpha}$. It follows that the function `check` will successfully detect and discard any input dataset that contains $B$ or more inputs not labeled by $\alpha$. Therefore, for any input dataset of $\beta B$ or more inputs the algorithm $\mathcal{A}_{\beta B}^\beta$ satisfies the conditions in Def. 19 of approximate-verifiable algorithm.

$\square$

As anticipated in the main text, we provide here an example of a 0.5-distinct concept class, which therefore satisfies the condition of Theorem 13, where the concepts are all (Promise)BQP complete functions.

**Definition 20** (0.5-distinct concept class with BQP concepts). Let $f$ be a PromiseBQP complete function as in Def. 13 on input $x \in \{0,1\}^n$. Then for any $f$ there exists a concept class $\mathcal{F}$ containing $2^n$ PromiseBQP-complete concepts defined as:

$$\mathcal{F} = \{f^\alpha : \{0,1\}^n \to \{0,1\} \mid \alpha \in \{0,1\}^n\} \tag{44}$$
$$f^\alpha : x \to f(x) \oplus (\alpha \cdot x) \tag{45}$$

where $\alpha \cdot x$ is the bitwise inner product modulo 2 of the vectors $\alpha$ and $x$.

Clearly for any $\alpha_1, \alpha_2 \in \{0,1\}^n$, $\alpha_1 \neq \alpha_2$, it holds that $f^{\alpha_1}(x) \neq f^{\alpha_2}(x)$ on exactly half of the input $x$, also they are all PromiseBQP complete functions as by evaluating one of them we can easily recover the value of $f(x)$.

**Theorem 14.** *If there exists an approximate-correct identification algorithm $\mathcal{A}_B$ as in Def. 8 for an average-case-smooth concept class $\mathcal{F}$, which works correctly for a dataset of size $B$, then there also exists an approximate-verifiable algorithm $\mathcal{A}_{\beta B}^{\beta}$ as in Def. 19 in NP which works correctly for a dataset of size $\beta B$.*

*Proof.* The proof is analogous to the one of Theorem 13. In particular we again leverage the existence of the algorithm $\bar{\mathcal{A}}_B$ to check if a dataset $T = \{(x_i, y_i)\}_{i=1}^{\beta B}$ of $\beta B$ inputs contains a fraction of $B$ inputs not consistent with a previously outputted $\alpha$. Since now the concept class satisfies the condition in Def. 6, in order to check if a set $T_B$ of $B$ inputs from $T$ are labeled by $f^{\bar{\alpha}}$ is sufficient to check if $d(\bar{\mathcal{A}}_B(T_B, \epsilon, r), \bar{\alpha}) \leq 1/3$, where $d$ is the distance function in Def 6. By the second property in Def. 8, if all the inputs in $T_B$ are consistent with $f^{\bar{\alpha}}$, then there exists a $r$ such that $d(\bar{\mathcal{A}}_B(T_B, \epsilon, r), \bar{\alpha}) \leq 1/3$ for any $\epsilon \leq \epsilon_2$ in Def. 8. The proof then follows the same structure as that of Theorem 13, with the only change being in the `check` function, which is now defined as follows.

---

1: **function** CHECK$(\alpha, T)$
2:     **for** $T_B$ **in** $T$
3:         **for** $r$ **in** $R$
4:             **if** $d(\bar{\mathcal{A}}_B(T_B, \epsilon_c, r), \bar{\alpha}) \leq 1/3$ **return** "invalid dataset"
5: **end function**

---

$\square$

As it was the case for the $c$-distinct concepts, even for the average-case-smooth concept class we present a concrete example of quantum functions which satisfies the average-case-smooth condition.

**Lemma 1** (Average-case-smooth quantum concept class). *Let $x \in \{0,1\}^n$, $m = \text{poly}(n)$, and $i(x) = \text{int}(x[1 : \lceil \log m \rceil])$, then the concept class:*

$$\mathcal{F} = \{f^{\alpha} : \{0,1\}^n \to \{0,1\} \mid \alpha \in \{0,1\}^n\} \tag{46}$$

$$f^{\alpha} : x \to \begin{cases} \alpha_{i(x)} & \text{if } i(x) \leq m \\ 0 & \text{else} \end{cases} \tag{47}$$

*is average-case-smooth and it can be efficiently implemented on a quantum computer.*

*Proof.* We first show that for any different $\alpha$ and $\alpha'$ the functions $f^{\alpha}$ and $f^{\alpha'}$ satisfies the definition of average-case-smoothness in Eq. ( 5). From the definition of $f^{\alpha}$ in Eq. (46), and recalling that the Hamming distance between two bitstrings $\alpha$ and $\alpha'$ is defined as $d(\alpha, \alpha') := \frac{1}{m}|\{i \in [m] : \alpha_i' \neq \alpha_i\}|$ we have that:

$$\mathbb{E}_{x \sim \text{Unif}(\{0,1\}^n)}|f^{\alpha}(x) - f^{\alpha'}(x)| = \frac{1}{2^n} \sum_{x \in \{0,1\}^n} |f^{\alpha}(x) - f^{\alpha'}(x)| \tag{48}$$

$$= \frac{1}{2^n} \sum_{i \in \{0,1\}^l} \sum_{z \in \{0,1\}^{n-l}} |f^{\alpha}(i|z) - f^{\alpha'}(i|z)| \tag{49}$$

$$= \frac{1}{2^n} \sum_{i \in \{0,1\}^l} |f^{\alpha}(i|0) - f^{\alpha'}(i|0)| \cdot 2^{n-l} \tag{50}$$

$$= \frac{1}{2^l} \sum_{i \in \{0,1\}^l} |f^{\alpha}(i|0) - f^{\alpha'}(i|0)| \tag{51}$$

$$= \frac{1}{2^l} \sum_{i=i}^{m} |\alpha_i - \alpha_i'| \tag{52}$$

$$= \frac{m}{2^l} d(\alpha, \alpha') \tag{53}$$

Where $i|z$ is the concatenation of the bitstring $i$ and $z$. Since $l = \lceil \log m \rceil$, we have $2^{l-1} < m \leq 2^l$, therefore:

$$\mathbb{E}_{x \sim \text{Unif}(\{0,1\}^n)}|f^{\alpha}(x) - f^{\alpha'}(x)| \geq \frac{1}{2} d(\alpha, \alpha') \tag{54}$$

So we proved that indeed the concept class $\mathcal{F}$ satisfies the average-case-smoothness in Def. 6.

We now present an easy implementation for the functions $f^\alpha$ on a quantum computer. For any concept $f^\alpha$, we encode $\alpha$ in an $m$-qubit computational basis state:

$$|\psi_\alpha\rangle = |\alpha_1\rangle \otimes ... \otimes |\alpha_m\rangle \tag{55}$$

Then the concept $f^\alpha$ can be implemented as:

$$f^\alpha(x) = \langle\psi_\alpha| O(x) |\psi_\alpha\rangle \tag{56}$$

Where for $i(x) \leq m$:

$$O(x) = I^{\otimes(i(x)-1)} \otimes Z \otimes I^{m-i(x)} \tag{57}$$

while if $i(x) > m$ we define $f^\alpha(x) = 0$.

$\square$

We notice that the functions considered in the above construction are obviously classically easy. However, one could consider the tensor product between the register $|\alpha\rangle$ and a second register undergoing any BQP hard computation, in direct analogy with the construction used in the proof of Theorem 15. That would construct a concept class of BQP function satisfying the average-case-smooth property, and therefore our Theorem 10 would apply.

## G    Some implications of our results

In this section, we discuss the connections between our theoretical findings with other well-known physically relevant tasks: Hamiltonian learning, the learning of parametrized observables and the learning of order parameters. We also present a physically motivated concept class to which our hardness results apply directly and where a quantum advantage is achieved. Moreover we provide a list of example of physical processes which give rise to BQP functions and practical settings where the identification task naturally arises.

For Hamiltonian learning, we clarify why an efficient classical algorithm for the identification task remains feasible. We then show how our results demonstrate the hardness of the identification task in learning parametrized observables. Finally, we discuss the connection to the task of learning of order parameters and the potential for quantum advantages therein.

### G.1    Hardness of identification for a physically motivated BQP-complete concept class

In the recent work Molteni et al. (2024) a physically motivated learning problem for which a learning separation was proved. Importantly, the classical hardness for the task was achieved by considering concepts being BQP-complete functions as in Def. 13 and then arguing that a classical learner would not be able to evaluate such concepts unless BQP $\subset$ P/poly. An interesting question is whether the classical hardness could arise already from identifying the target concept that labels the data, rather than only from evaluating it. Using the results in Section 5, we can answer this question in an affirmative way for a subclass of problems for which the hardness of evaluation holds. Let us restate the learning problem presented in Molteni et al. (2024) and show how our results in the previous section apply to it. The learning problem considered is described by the following concept class:

$$\mathcal{F}^{U,O} = \{f^{\boldsymbol{\alpha}}(\boldsymbol{x}) \in \mathbb{R} \mid \boldsymbol{\alpha} \in [-1,1]^m\} \tag{58}$$

$$\text{with:} \quad f^{\boldsymbol{\alpha}}(\boldsymbol{x}): \quad \boldsymbol{x} \in \{0,1\}^n \rightarrow f^{\boldsymbol{\alpha}}(\boldsymbol{x}) = \text{Tr}[\rho_U(\boldsymbol{x})O(\boldsymbol{\alpha})]$$

$$O(\boldsymbol{\alpha}) = \sum_{i=1}^m \boldsymbol{\alpha}_i P_i.$$

Where $\rho_U(\mathbf{x})$ is a "classically hard to compute" quantum state (i.e., certain properties of this state are hard to compute) depending on $\boldsymbol{x} \in \{0,1\}^n$. An example could be $\rho_U(\boldsymbol{x}) = U |\boldsymbol{x}\rangle \langle\boldsymbol{x}| U^\dagger$ with $U = e^{iH\tau}$ where $H$ is a local Hamiltonian whose time evolution is universal for BQP Feynman (1985). As each $P_i$ represents a $k$-local Pauli string, the function $f^{\boldsymbol{\alpha}}$ is a BQP-complete function

for some values of $\boldsymbol{\alpha}$. For example, for the $\boldsymbol{\alpha}_Z$ which corresponds to $O(\boldsymbol{\alpha}_Z) = Z \otimes I \otimes ... \otimes I$ the function $f^{\boldsymbol{\alpha}_Z}$ is exactly the function which corresponds to the time evolution problem in the literature Kitaev et al. (2002), which is known to be universal for BQP. This already motivates the classical hardness of the learning task under the assumption BQP $\not\subseteq$ P/poly as the classical learner is required to evaluate a correct hypothesis on new input data. We now argue that concept classes of the kind $\mathcal{F}^{U,O}$ are also not classically identifiable as in Def. 8. In order to show it, we will consider a particular construction for the quantum states $\rho_U(\boldsymbol{x})$ and observables $O(\boldsymbol{\alpha})$ such that the corresponding concept class $\mathcal{F}^{U,O}$ satisfies the definition of $c$-distinct concept class in Def. 5 with $c = 1/3$.

It is important to note that in Molteni et al. (2024), a quantum learning algorithm was proposed that solves the task by first identifying a sufficiently good $\alpha$, and then evaluating the model using the identified $\alpha$. As such, the quantum algorithm also addresses the identification task. Furthermore, as demonstrated in Molteni et al. (2024), the algorithm remains effective even in the presence of noise in the training data.

**Theorem 15.** *Under the assumption* $(\mathsf{BQP}, \mathsf{Unif}) \not\subseteq \mathsf{HeurBPP}^{\mathsf{NP}^{\mathsf{NP}^{\mathsf{NP}}}}$, *there exists a family of unitaries* $\{U(\boldsymbol{x})\}_{\boldsymbol{x}}$ *and observables* $\{O(\boldsymbol{\alpha})\}_{\boldsymbol{\alpha}}$ *such that the corresponding concept class* $\mathcal{F}^{U,O}$ *is not classically identifiable by an approximate-correct algorithm as in Def. 8, but is identifiable by a quantum learning algorithm.*

The key idea of the proof is to construct a particular family of quantum states $\rho_U$ and observables $O$ such that the corresponding concept class in Eq. (58) contains $c$-distinct concepts with $c \geq 1/3$. We achieve this by leveraging the Reed-Solomon code to construct functions that differ from each other on at least $1/3$ of all the input $x$. As this is still an instance of the same category of concept classes as studied in Molteni et al. (2024), the quantum learning algorithm from this work will solve the identification task, and also the corresponding evaluation task.

We give here the complete proof of Theorem 15.

*Proof.* Let us first introduce the set of $3n$ classical functions $\{f_j\}_{j=1}^{3n}$ defined as follows:

$$f_j(x) = \begin{cases} f_j(x) = 0 \text{ if } x \notin \mathcal{L}_j \\ f_j(x) = 1 \text{ if } x \in \mathcal{L}_j \end{cases} . \tag{59}$$

Where $\mathcal{L}_j = \{\boldsymbol{x_j}, \boldsymbol{x_{j+1}}, \ldots, \boldsymbol{x_{j+2^n/3n}}\}$ denotes the $j$-th block, a contiguous subset of $\frac{2^n}{3n}$ elements selected from the set of all possible $x \in \{0,1\}^n$, according to a predefined order. We then consider the Reed-Solomon code Wicker & Bhargava (1999) $RS(3|k|, |k|)$ defined over a field with $q = 2^m$ elements which maps a message $k$ of length $|k|$ into a codeword of length $3|k|$. Let us consider $m$ and $|k|$ such that $n = |k| \cdot m$. In this way the minimum distance between two different codewords is lower bounded by $d \geq 3|k| - k + 1 \geq 2|k| + 1 \geq |k|$. This corresponds to a Hamming distance greater than $d_H \geq |k| \cdot m$ between the bitstring representation of codewords corresponding to two different messages $k$ and $k'$. Next, we define the function $g : \{0,1\}^n \to \{0,1\}^{3n}$ as the function which maps the bitstring representation of the message $k$ into the bitstring representation of its codeword of the previously introduced $RS(3|k|, |k|)$ code. In this way, $d_H(g(k), g(k')) \geq n$ for every $k \neq k'$, where $d_H(g(k), g(k'))$ stands for the Hamming distance between the $3n$-bits given by the function $g$. Consider now the family of $2^n$ functions $\{h_k\}_{k=1}^{2^n}$ defined as:

$$h_k(x) = \sum_{j=1}^{3n} [g(k)]_j \cdot f_j(x) \tag{60}$$

It is clear that for different $k$ each function $h_k$ will differ from the others on at least $\frac{2^n}{3}$ distinct values of $x \in \{0,1\}^n$. We now want to implement the functions $h_k$ in the form of the concepts in $\mathcal{F}^{H,O}$. We can do so by considering a register of $3n \times n$ qubits. On each of the $3n$ registers the function $f_j$ in Eq. (59) is implemented by measuring the $Z$ observable on the first qubit after having prepared a corresponding state $\rho_j(\boldsymbol{x})$. In particular the unitary $U$ is the tensor product of each $U_j$ acting on the $j$-th register of $n$ qubits, with $j = 1, 2, ..., 3n$. Each $U_j$ is such that:

$$f_j(x) = \langle x | U_j^\dagger Z_1 U_j | x \rangle \tag{61}$$

where $Z_1$ represent the $Z$ Pauli operator acting on the first qubit of the $j$-th register. We note that since classical computations are in BQP such a unitary always exists. Let us consider then the following circuit composed of $3n + 1$ register of $n$ qubit each. On the first register the input dependent state $|\boldsymbol{x}\rangle$ undergoes an arbitrary BQP computation and the observable $Z$ is measured on the first qubit. The other registers implement the functions $f_j$ as described above. As they are classical functions, they can also be implemented quantumly. Next we define the required set of observables $\{O(\boldsymbol{\alpha})\}_{\boldsymbol{\alpha}}$ defined as:

$$O(\boldsymbol{\alpha}) = Z_1 \otimes \sum_{j=1}^{3n} \alpha_j Z_{nj+1}, \qquad (62)$$

where $Z_i$ represents the Pauli string consisting of identity operators on all qubits except for the $Z$ matrix acting on the $i$-th qubit. Each concept in $f^{\boldsymbol{\alpha}} \in \mathcal{F}$ is now expressed as $f^{\boldsymbol{\alpha}} = \text{Tr}[\rho(\boldsymbol{x})O(\boldsymbol{\alpha})]$, where $\rho(\boldsymbol{x})$ represents the quantum state formed by the tensor product of the previously described $3n + 1$ registers, and $O(\boldsymbol{\alpha})$ is defined as in Eq. (62). Specifically, $\rho(\boldsymbol{x})$ is defined as follows:

$$\rho(\boldsymbol{x}) = \bigotimes_{j=1}^{3n+1} \rho_j(x), \quad \rho_j = \begin{cases} \text{if } j = 1: \rho_1(\boldsymbol{x}) = |\psi\rangle \langle\psi|_1 \text{ with } |\psi\rangle = U_{BQP} |\boldsymbol{x}\rangle \\ \text{if } j \neq 1: \rho_j(\boldsymbol{x}) = |\psi\rangle \langle\psi|_j \text{ with } |\psi\rangle = U_j |\boldsymbol{x}\rangle \end{cases}. \qquad (63)$$

where $U_{BQP}$ represents any arbitrary BQP computation while $\{U_j\}_j$ are the unitaries associated to the functions $f_j$ as defined in Eq. (61).

If we now consider the concept class

$$\mathcal{F} = \{f^{\boldsymbol{\alpha}} \mid f^{\boldsymbol{\alpha}} = \text{Tr}[\rho(\boldsymbol{x})O(\boldsymbol{\alpha})], \ \boldsymbol{\alpha} = g(k) \ \forall k \in \{0,1\}^n\}, \qquad (64)$$

where the functions $f^{\boldsymbol{\alpha}}$ and $g$ are defined as above, we obtain a concept class $\mathcal{F}$ such that for any pair of different $\boldsymbol{\alpha}_1, \boldsymbol{\alpha}_2$ the corresponding concepts differ on at least $2^n/3$ different inputs $x \in \{0,1\}^n$. Therefore the constructed concept class is a $c$-distinct concept class as in Def. 5, with $c = 1/3$. We can then apply our Theorem 13 and construct an approximate-verifiable algorithm from an approximate-correct identification algorithm as in Def. 8 which correctly solves the identification task for $\mathcal{F}$ by using an NP oracle. Moreover, as there is at least a PromiseBQP complete concept for $\boldsymbol{\alpha} = \boldsymbol{0}$[11], for Theorem 11 if there exists an approximate-verifiable algorithm which runs in the second level of the polynomial hierarchy then BQP would be contained in the fourth level of the PH. We can therefore conclude that the constructed concept class $\mathcal{F}$ is not learnable even in the identification sense with an approximate algorithm which satisfies Def. 8. $\qquad \square$

We note here that, although the task defined by the concept class in Eq.( 58) is general and include many physical scenarios as motivated in Molteni et al. (2024), the specific choice of unitaries $\{U(\boldsymbol{x})\}_{\boldsymbol{x}}$ and observables $\{O(\boldsymbol{\alpha})\}_{\boldsymbol{\alpha}}$ used in the proof of Theorem 15 is purely technical, intended to ensure the concept class is $1/3$-distinct.

## G.2 HAMILTONIAN LEARNING

Hamiltonian learning is a well-known problem which, arguably surprisingly, admits an efficient classical solution. Moreover, the Hamiltonian learning problem can be easily framed as a standard learning problem within the PAC framework, revealing clear connections to the identification task discussed in this paper. Although the underlying concepts may initially seem inherently quantum, the existence of an efficient classical algorithm for solving the problem raises the question of why our hardness results do not extend to this setting. In its main variant, the task involves reconstructing an unknown Hamiltonian from measurements of its Gibbs state at a temperature $T$. Specifically, given an unknown local Hamiltonian of the form $H(\boldsymbol{\lambda}) = \sum_i \lambda_i P_i$, the goal of the Hamiltonian learning procedure is to recover $\boldsymbol{\lambda}$ from measurements of its Gibbs state $\rho(\beta, \boldsymbol{\lambda})$ at temperature $T$:

$$\rho(\beta, \boldsymbol{\lambda}) = \frac{e^{\beta H(\boldsymbol{\lambda})}}{\text{Tr}[e^{\beta H(\boldsymbol{\lambda})}]}$$

where $\beta = \frac{1}{T}$. In Anshu et al. (2020), the authors proposed a classical algorithm capable of recovering the unknown Hamiltonian parameterized by $\boldsymbol{\lambda}$ using a polynomial number of measurements of the

---

[11]We do note that, even for most values of $\boldsymbol{\alpha} \neq \boldsymbol{0}$, we still expect the corresponding concepts to be PromiseBQP-complete.

local Pauli terms $\{P_i\}_i$ that constitute the target Hamiltonian $H(\boldsymbol{\lambda}) = \sum_i \lambda_i P_i$. In Haah et al. (2024), an also time-efficient algorithm was proposed. The above task can be reformulated as an identification problem by considering the following concept class:

$$\mathcal{F}_\beta = \{f^{\boldsymbol{\lambda}}(\boldsymbol{x}) \in \mathbb{R} \ \mid \ \boldsymbol{\lambda} \in [-1,1]^m\} \tag{65}$$

$$\text{with:} \quad f^{\boldsymbol{\lambda}}(\boldsymbol{x}): \ x \in \mathcal{X} \subseteq \{0,1\}^n \to \text{Tr}[\rho(\beta, \boldsymbol{\lambda})O(\boldsymbol{x})]$$

$$O(\boldsymbol{x}) = \sum_{i=1}^{\text{poly}(n)} x_i P_i.$$

Then given polynomially many training samples of the kind $T = \{(\boldsymbol{x}_\ell, f^{\boldsymbol{\lambda}}(\boldsymbol{x}_\ell))\}_\ell$ it is possible to recover the expectation values of the local Pauli terms $\{P_i\}_i$ and therefore the identification problem can be solved using the learning algorithm in Haah et al. (2024) for learning the vector $\boldsymbol{\lambda}$.

We now provide a list of incompatibilities between the task of Hamiltonian learning and the identification task considered in our results, and examine them to try to find the crux of the reason why Hamiltonian learning remains classically feasible.

- **Hamiltonian learning is a regression problem.** It is important to highlight that the concepts in Eq. (65) are not binary functions, as they are in our theorems, but rather form a regression problem. We note that this categorical difference alone is also not the end of the explanation. It is in fact possible to "binarize" the concepts in the following way:

$$f^{\boldsymbol{\lambda}}(\boldsymbol{x}, i): \ (\boldsymbol{x}, i) \in \subseteq \{0,1\}^n \times \{0,1\}^{\log n} \to \text{bin}\Big[\text{Tr}[\rho(\beta, \boldsymbol{\lambda})O(\boldsymbol{x})], i\Big] \tag{66}$$

where $\text{bin}\Big[\boldsymbol{y}, i\Big]$ returns the $i$-th bit of $\boldsymbol{y}$. A Hamiltonian learning algorithm can still be used to determine the unknown $\boldsymbol{\lambda}$. This can be done, for example, by considering an input distribution that focuses on exploring the most significant bits of the extreme inputs where the observable $O(\boldsymbol{x})$ reduces to a single Pauli string, specifically inputs $\boldsymbol{x} \in \{0,1\}^n$ with Hamming weight 1.

- **The concepts are not** BQP**-hard.** The concepts in $\mathcal{F}_\beta$, binarized as in Eq. (66), involve measurements performed on Gibbs states. For large values of $\beta$, these states closely approximate ground states, which might suggest a connection to tasks where quantum computers are expected to outperform classical ones. However, regardless of how hard it is to estimate expectation values from cold Gibbs states, it is important to clarify that each concept is associated with a *fixed* Gibbs state, and the input $\boldsymbol{x} \in \{0,1\}^n$ simply determines the observable being measured. This is similar to the "flipped concepts" case studied in Molteni et al. (2024). It is easy to see that a P/poly machine can compute these concepts given the expectation values of the (polynomially many) Pauli strings $\{P_i\}_i$ as advice, due to the linearity of the trace. Thus, the concepts we have here are somewhere in the intersection of P/poly and BQP-like problems (depending on what temperature Gibbs states we are dealing with, let us call the corresponding class $A$, where $A$ could be QXC Bravyi et al. (2024) ). However, even this is not sufficient to full explain the apparent gap between our no-go results and the efficiency of Hamiltonian learning. In our proofs we worked towards the (unlikely) implication that BQP was not in a heuristic version of a level of the PH. Here, we can construct fully analogous arguments and obtain the implication that, e.g., P/poly $\cap$ $A$ is in PH, and we also have no reason to believe this to be true. Indeed, proving that this inclusion holds would, to our understanding, constitute a major result in quantum complexity theory. The reasons why the no-go's do not apply are more subtle still.

- **Hamiltonian learning algorithm doesn't satisfy the assumptions of the right type of identification algorithm.** It is clear that the Hamiltonian learning algorithm does not satisfy the conditions of an approximate-verifiable algorithm in Def. 19, as it fails to detect datasets that do not contain enough inputs labeled by a single concept - it always outputs *some guess* for the Hamiltonian. However we could again try to circumvent this issue by employing the constructions from Theorem 13 or Theorem 14, to construct approximate-verifiable identification algorithm somewhere in the PH, which would again suffice for a likely contradiction. However, it is important to note that in normal Hamiltonian learning

settings neither of the two Theorems apply. The theorems require that concept class must either consist of sufficiently distinct concepts (Def.5) or exhibit average-case smoothness (Def.6). On the face of it, neither of these conditions seem to hold for the concept class in Eq. (65), and it is not clear how one would go about attempting to enforce them. This final point is in our view at the crux of the difference between the settings where our no-go's apply and Hamiltonian learning.

While we leave the investigation of the hardness of the Hamiltonian learning task as a potential direction for future work, already at this point an interesting observation emerges. It is a natural question if the conditions of the two Theorems 13 and 14 could be relaxed and generalized, which would lead to the hardness of identification for broader classes of problems. Due to the analysis of the Hamiltonian learning case we see that *if* one could generalize the settings to the point they apply to Hamiltonian learning, since Hamiltonian learning *is* classically tractable, this would imply very surprising results in complexity theory (see second bullet point above). We see it more likely that this is a reason to believe the range of generalization of the settings where identification is intractable is more limited and will not include the standard settings of Hamiltonian learning.

### G.3 THE CASE OF LEARNING OF ORDER PARAMETERS

Another physically meaningful problem that can be framed as an identification task is learning the order parameter that distinguishes between different phases of matter. Consider, for example, a family of Hamiltonians $\{H(\boldsymbol{x})\}_{\boldsymbol{x}}$ parametrized by vectors $\boldsymbol{x} \in [-1, 1]^m$, where the corresponding ground states exhibit distinct phases depending on the value of $\boldsymbol{x}$. In many cases, such as symmetry-breaking phases, there exists a local order parameter of the form $O = \sum_i \alpha_i P_i$, whose expectation value on a given ground state reveals the phase to which it belongs to. The task is to learn the order parameter from a collection of samples $\{(\rho(\boldsymbol{x}_\ell), y_\ell)\}_\ell$, where each $\rho(\boldsymbol{x}_\ell)$ represents the ground state of the Hamiltonian $H(\boldsymbol{x}_\ell)$, and $y_\ell$ denotes the label identifying its associated phase. We can formalize the problem by considering the following concept class:

$$\mathcal{F}_\alpha = \{f^{\boldsymbol{\alpha}}(\boldsymbol{x}) \in \mathbb{R} \mid \boldsymbol{\alpha} \in [-1, 1]^m\} \tag{67}$$

$$\text{with:} \quad f^{\boldsymbol{\alpha}}(\boldsymbol{x}) : \ \boldsymbol{x} \in \mathcal{X} \subseteq \{0, 1\}^n \to \text{Tr}[\rho(\boldsymbol{x})O(\boldsymbol{\alpha})]$$

$$O(\boldsymbol{\alpha}) = \sum_i \boldsymbol{\alpha}_i P_i.$$

Learning the correct order parameter from ground states labeled by their phases can be framed as a machine learning task, where the goal is to identify the underlying labeling function. Specifically, given training data of the form $T = \{(\boldsymbol{x}_\ell, \text{Tr}[\rho(\boldsymbol{x}_\ell)O(\boldsymbol{\alpha}^*)])\}_\ell$, the objective is to recover the correct parameter $\boldsymbol{\alpha}^*$ that enables accurate labeling of the ground state $\rho(\boldsymbol{x}_\ell)$ corresponding to the Hamiltonian $H(\boldsymbol{x}_\ell)$. In other words, the task reduces to identifying the correct order parameter $O(\boldsymbol{\alpha}^*)$. In this setting, the value $\text{Tr}[\rho(\boldsymbol{x}_\ell)O(\boldsymbol{\alpha}^*)]$ acts as a phase indicator. For instance, in systems exhibiting two distinct phases, the sign of $\text{Tr}[\rho(\boldsymbol{x}_\ell)O(\boldsymbol{\alpha}^*)]$ serves to distinguish between them. In case the ground states of the Hamiltonian family $\{H(\boldsymbol{x})\}_{\boldsymbol{x}}$ are computable in BQP, then the quantum algorithm from Molteni et al. (2024), which solves the identification task for the concept class in Eq.(58), can also be employed to solve the learning problem defined in Eq.(67), thus enabling the recovery of the correct order parameter. In the general case, however, the concepts in Eq. (67) compute expectation values on ground states, a task that for a general local Hamiltonian is in QMA-hard, but the situation is actually more involved.

First, we note that the closely related task learning of phases of matter *given the shadows of the ground states* - so where the data consists of pairs $(\sigma(\rho(\boldsymbol{x})), \text{phase}(\boldsymbol{x}))$, where $\sigma(\cdot)$ denotes the classical shadow of a state, and "phase" assigns a binary label specifying the phase - is classically easy, even for many topological phases of matter Huang et al. (2022). In this case, the task is to assign the correct phase to a new datum which is a shadow of a new state given as $\sigma(\rho(\boldsymbol{x}))$. This is different from the setting we consider here, where we explicitly deal with the specification $\boldsymbol{x}$ of the Hamiltonian; that is, pairs $(\boldsymbol{x}, \text{phase}(\boldsymbol{x}))$ [12]. While Huang et al. (2022) also shows how mappings $\boldsymbol{x} \to \sigma(\rho(\boldsymbol{x}))$ can be classically learned as long as the Hamiltonians specified by all $\boldsymbol{x}$ are within the

---

[12]Also, the setting in Huang et al. (2022) do not explicitly find the observable, which is the order parameter, although we suspect this can be computed from the hyperplane found by the linear classifier.

same phase, the identifying of phases requires the crossing of the phase boundaries, and it is not clear how the approaches could be combined.

To analyze the perspectives of classical intractability and then quantum tractability of this task from the lens of the work of this paper, we analyze the key criteria. First, for our hardness results to apply at all, the concepts in the concept class should be sufficiently hard, in a complexity-theoretic sense. That is, the functions that we consider $x \to \text{Tr}[\rho(x)O(\alpha)]$ should be classically intractable, yet quantum easy. In general, this is easy to achieve: by using the standard Kitaev circuit-to-Hamiltonian constructions we can construct Hamiltonians whose ground states encode the output of arbitrary quantum circuits (here encoded in $x$), as was used in Molteni et al. (2024). However, from the perspective of phases of matter identification, it is important to notice that such $\text{BQP}-$hard Hamiltonians are critical: they have an algebraically vanishing gap, and it is an open question whether in this sense $\text{BQP}-$hard Hamiltonians could be gapped at all González-Guillén & Cubitt (2018). At least in the cases of conventional phases of matter (symmetry breaking, even topological), the phases are typically characterized by a constant gap. This suggests that the "hard computations" could only be happening at (or increasingly close to) the phase boundary.

However, we are also reminded that the observable we need to ultimately measure has a meaning: it is the order parameter. This has an interesting implication. We are interested in the setting where the function $x \to \text{Tr}[\rho(x)O(\alpha)]$ corresponds to, intuitively, the characteristic function of some BQP(-hard) language $\mathcal{L}$. But since this function is also the order parameter, the $yes$ instances of the language ($x \in \mathcal{L}_{yes}$) must correspond to Hamiltonians in one phase, whereas the $no$ instances must correspond to the other. This observation allows for a simple cheat, as it highlights the importance of the mapping $x \to H(x)$, which we have the freedom to specify. One can conceal all the hardness in this map and simply choose it such that $H(x)$ is some Hamiltonian in one phase for $x \in \mathcal{L}_{yes}$, and in another for the rest. This is of course unsatisfactory as it involves a highly contrived parametrization.

For natural, smooth parametrizations of the Hamiltonian space, we do need to worry about the constant gap property and so it is not clear whether hard functions emerge in standard settings. If, however, we move to more exotic systems, e.g. general critical systems, systems with complex non-local order parameters[13] and dynamic phases of matter, it becomes more likely that the right theoretical conditions arise even with natural parametrizations.

The hardness of the concepts will ensure the hardness of evaluation-type tasks which is the first step. To achieve the hardness of identification, we need more, i.e., either c-distinct concepts or a smooth family. Learning of order parameters is a closely related task to the learning of observables, and as we discussed in Section G.1, for this more general case it is possible to construct cases that satisfy all the desired assumptions. Whether similar conditions will also be met for some natural settings involving exotic (or standard) phases of matter remains a target of ongoing research.

### G.4 PRACTICAL RELEVANCE OF OUR RESULTS

Although our definition of BQP-complete functions in Def. 13 may seem abstract and distant from what it can be found in practical experiment, we remark that many relevant physical processes have been demonstrated to be BQP-complete. For example, estimating expectation values on time-evolved states is shown to be BQP-complete for many physical Hamiltonians :

- Various variants of the Bose-Hubbard model Childs et al. (2013).

- Stoquastic Hamiltonians Janzing & Wocjan (2006).

- Ferromagnetic Heisenberg model Childs et al. (2013).

- The $XY$ model Piddock & Montanaro (2015).

- Estimating the scattering probabilities in massive field theories Jordan et al. (2018), (more precisely, estimating the vacuum-to-vacuum transition amplitude, in the presence of spacetime-dependent classical sources, for a massive scalar field theory in $(1 + 1)$ dimensions).

- Simulation of topological quantum field theories Freedman et al. (2002).

---

[13]Note that we could encode universal quantum computations in complex enough global measurements.

On top of those, problems from other fields as quantum chemistry and topological data analysis have been proven to be BQP complete, for example:

- Electronic structure problem O'Gorman et al. (2021) for quantum chemistry.

- Computing the persistence of Betti numbers Gyurik et al. (2024)

Any learning problem related to these processes involves BQP-complete functions, to which our results apply.

In addition to the task of learning an order parameter, we highlight several other scenarios that can be naturally modeled within our framework of the identification task.

- First, beyond the case of exactly identifying an unknown order parameter, our framework also applies when the order parameter is known but the corresponding observable is not directly implementable. In such situations, our approach enables the identification of a suitable "proxy" observable that captures the relevant information.

- When considering unitarily parameterized observables, time evolution under a parameterized Hamiltonian can be viewed as part of the measurement process. This allows our framework to encompass cases in which parts of the quantum dynamics are unknown, even though the final measurement setup is fixed, thus covering a broader class of partially unknown observables.

- Identifying an unknown measurement is also crucial in experimental settings where incomplete device characterization means the actual measurements performed may deviate from the intended ones. This is especially relevant in quantum computing, where noise and imperfections in measurement devices are common, making it important to understand and mitigate their impact.

- Finally, we note that the problem of learning an unknown measurement has been extensively studied in the literature under the name of quantum measurement tomography Luis & Sánchez-Soto (1999); Lundeen et al. (2009), further underscoring the practical relevance of this task.

### G.5 BENCHMARKING AGAINST CLASSICAL METHODS

In Appendix G.1, we describe an example of identification task that is classically hard yet solvable by quantum computers. It is natural to ask how the theoretical guarantees manifest when actually solving the task on quantum and classical devices. However, it is important to emphasize that implementing this for BQP-complete functions and, more importantly, benchmarking it against the best classical learning algorithms in a provably rigorous way presents distinct challenges. Specifically, the identification task requires the learner to take an entire dataset as input and output only the description of the labeling function. It is unclear how classical methods could perform such training or which loss function would be appropriate. One possible approach is to design the learning algorithm so that a tentative labeling function is tested by evaluating it on different inputs and comparing the results with the training data. However, this would require the target functions to be classically computable, which is ruled out by our focus on BQP-complete functions. Ultimately, the desired algorithm should be analogous to Hamiltonian learning, where, given a complete dataset of expectation values, the output is a description of the unknown Hamiltonian. Our theoretical results rule out the possibility of the existence of this kind of algorithm for learning problems involving BQP-complete functions. Thus, the most promising approach for designing a classical algorithm is to dequantize the procedure in Appendix G.1 for the identification of the observable. This, however, would require a highly efficient simulator for quantum computations, with runtime scaling as $\text{poly}(n) + 2^{0.23t}t^3w^3$ (where $t$ is the number of T-gates and $w$ the number of measured qubits). In particular, this implies a scaling as $\text{poly}(n) + 2^{0.3t}$. For simulation problems with around 300 qubits, and assuming a linear number of T-gates in the qubit count (which is reasonable for the BQP-hard computations we are interested in our results), this already yields around $2^{10} \sim 10^{31}$ classical steps, clearly infeasible, while a fault-tolerant quantum computer could handle the task without difficulty.

# H DISCUSSION ON THE ASSUMPTIONS OF THE APPROXIMATE-CORRECT ALGORITHM

For convenience, we report here the definition of approximate-correct algorithm stated in the main text in Def. 8.

**Definition 21** (Identification task - non verifiable case ). Let $\mathcal{F} = \{f^\alpha : \{0,1\}^n \to \{0,1\} \mid \alpha \in \{0,1\}^m\}$ be a concept class. An *approximate-correct* identification algorithm is a (randomized) algorithm $\mathcal{A}_B$ such that when given as input a set $T = \{(x_\ell, y_\ell)\}_{\ell=1}^B$ of at least $B$ pairs $(x,y) \in \{0,1\}^n \times \{0,1\}$, an error parameter $\epsilon > 0$ and a random string $r \in R$ satisfies the definition of a proper PAC learner (see Def. 4) along with the following additional properties:

1. If for any $\alpha$ all the $(x_\ell, y_\ell) \in T$ are such that $y_\ell \neq f^\alpha(x_\ell)$ then there exists an $\epsilon_1$ such that for all $\epsilon \leq \epsilon_1$ and all $r \in R$:

$$\mathcal{A}_B(T, \epsilon_1, r) \neq \alpha. \tag{68}$$

   Therefore, for no dataset the algorithm can output a totally wrong $\alpha$, i.e. an $\alpha$ inconsistent with all the inputs in the dataset.

2. If $T = \{(x_\ell, y_\ell)\}_{\ell=1}^B$ is composed of different inputs $x_\ell$ and if there exists an $\alpha$ such that the corresponding labels follow $y_\ell = f^\alpha(x_\ell)$ for all $(x_\ell, y_\ell) \in T$, then there exists a threshold $\epsilon_2$ such that for any $\epsilon \leq \epsilon_2$ there exists at least one $r \in R$ for which:

$$\mathcal{A}_B(T, \epsilon_2, r) = \alpha_2 \tag{69}$$

   With the condition: $\mathbb{E}_{x \sim \text{Unif}(\{0,1\}^n)} |f^\alpha(x) - f^{\alpha_2}(x)| \leq \frac{1}{3}$.
   Therefore, if the dataset is fully consistent with one of the concept $\alpha$, then there is at least one random string for which the identification algorithm will output a $\tilde{\alpha}$ closer than $\frac{1}{3}$ in PAC condition to the true labelling $\alpha$.

We say that $\mathcal{A}_B$ solves the identification task for a concept class $\mathcal{F}$ under the input distribution $\mathcal{D}$ if the algorithm works for any value of $\epsilon, \delta \geq 0$. The required minimum size $B$ of the input set $T$ is assumed to scale as poly($n$,$1/\epsilon$,$1/\delta$), while the running time of the algorithm scales as poly($B, n$). Moreover, the $\epsilon_1$ and $\epsilon_2$ required values scale at most inverse polynomially with $n$.

We observe that an approximate-correct algorithm is, by definition, a proper PAC learner as in Def. 4, and thus guarantees a hypothesis with accuracy within $\epsilon = \frac{1}{\text{poly}(n)}$. In addition to this, we also require it satisfies the two extra conditions specified above. Although the two additional conditions are introduced for technical reasons required by our proof strategy, they are also designed to reflect what one would reasonably expect from any practical learning algorithm. The first property ensures that if the input dataset consists entirely of inputs labeled differently from a given concept, then the algorithm will not output that concept—meaning it is never "totally incorrect". The second property concerns the case where all points in the input dataset are labeled consistently with a particular concept. In this case, the algorithm is required to output a concept that is "close" in the PAC sense for at least one random string, regardless of the distribution from which the training set was drawn. We therefore view the two conditions as natural and well-motivated requirements for a learning algorithm. Moreover, while the second property closely resembles the condition for proper PAC learning in Definition 4—with the key difference being that the training set need not be drawn from a specific distribution—we present a simple example of a concept class where a proper PAC learner would also satisfy the first property with high probability. Consider a concept class $\mathcal{F} = \{f_0, f_1\}$ consisting of two functions such that $f_0(x) \neq f_1(x)$ for every $x \in \{0,1\}^n$. In this case, any dataset that does not contain inputs labeled by $f_0$ is fully consistent with $f_1$, and vice versa. As a result, a proper PAC learner would, with high probability, output the concept consistent with the dataset, rather than the one that is entirely incorrect. This simple scenario changes in the presence of a larger concept class. When multiple concepts are involved, a dataset that is totally inconsistent with one concept may still not be fully consistent with any other. In such cases, a proper PAC learner, without the additional assumption from Def. 8, has no guarantees on the concept outputted.

