# OpenReview forum: "Quantum machine learning advantages beyond hardness of evaluation"
_ICLR.cc/2026/Conference — ICLR 2026 Poster_

### Official Review · Reviewer_Wcsz · 2025-10-25

**Soundness:** 3
**Presentation:** 3
**Contribution:** 3
**Rating:** 4
**Confidence:** 4

**Summary:**

The authors provide a theoretical proof of the hardness of a class of identification problems, thus constructing a classical–quantum separation in the context of PAC learning theory under widely believed complexity conjectures. This result can be considered a generalization of Huang et al. (2021) and Gyurik and Dunjko (2023), and it deepens our understanding of the power of data in quantum learning algorithms. However, for the reasons below (please refer to the Weaknesses section), I cannot recommend accepting this paper in its current form. If the authors adequately address my concerns, I may change the rate.

**Strengths:**

This paper is well written and logically clear. The chosen topic is highly significant, as it helps to characterize the classical–quantum boundary within the framework of computational complexity.

**Weaknesses:**

1. Regarding your first two results (Theorems 1 and 2): these statements seem quite unsurprising. Since the target function f essentially defines a BQP language, it follows that a polynomial-time classical algorithm cannot generate such samples. I do not find these two results surprising.

2. The main claim appears in Theorem 5, which establishes classical hardness. But to demonstrate a classical–quantum separation one must also show an efficient quantum algorithm. In fact, even if f is produced by a quantum device, that does not imply a quantum-machine-learning (QML) algorithm can solve the problem. For example, if the dataset encodes ground-state properties that reveal complex topological order, QML algorithms might still struggle to recognize it. Thus the authors have only proven classical hardness; they have not demonstrated quantum efficiency. As a result, an exponential quantum advantage is not rigorously established.

**Questions:**

Here are some questions:

1. Can the authors give some examples of the average-case-smooth concept class, especially in the quantum computing setting?

2. Regarding Theorem 5: why does L, unif \in BPP^{3NP} lead to a contradiction? To the best of my knowledge, the relationship between these two classes is not known.

3. Could the authors provide an efficient quantum algorithm for learning the classically hard cases indicated in Theorem 5?

---

> ### Author Response · Authors · 2025-11-20
>
> **Regarding your first two results (Theorems 1 and 2): these statements seem quite unsurprising. Since the target function f essentially defines a BQP language, it follows that a polynomial-time classical algorithm cannot generate such samples. I do not find these two results surprising**
>
> We thank the reviewer for this observation, which gives us the opportunity to better clarify the relevance of our results. We agree that the functions f we consider define BQP languages and are therefore believed to be classically hard to compute. However, hardness of computing a function does not in general imply hardness of generating random labeled samples from it.
>
> A standard example is the discrete logarithm problem, which is believed to be hard not only for polynomial-time algorithms (i.e., not in P), but also for heuristic non-uniform algorithms (i.e., not in HeurP/poly). This is derived by its average-case hardness properties and by the cryptographic viewpoint that polynomial-size advice corresponds to non-uniform adversaries, against which discrete logarithm is still expected to be secure. Then, under widely believed assumptions, computing the function x -> log_⁡a(x)  mod p is classically hard for large primes p. Yet, since the inverse function y -> a^y  mod p is efficiently computable, one can classically and efficiently generate random labeled samples (a^y  mod p,y) = (x,DL(x,a,p)) under the uniform distribution. Thus, even though evaluating the discrete logarithm is intractable, producing random samples consistent with it is classically easy.
> Before our work, it was not clear whether a similar phenomenon could occur for quantum (BQP) functions, i.e., whether there could exist efficient classical procedures that generate random labeled examples, despite the functions themselves being hard to compute classically. Our Theorems 1 and 2 rigorously rule out this possibility (under the assumption BQP $\not\in$ PH): we show that random generability of such quantum functions is itself classically hard. Therefore, in this sense, our results are not just a restatement of the hardness of BQP languages. We also emphasize that the failure of the simple “invertible function” argument used in the discrete logarithm example (discussed above) is not by itself sufficient, since invertibility is not a necessary condition for a function to admit efficient random generation. In particular, it can also be proven that random generatability is possible also for graph isomorphism problem [5], which is strictly speaking not in P but also not known to be in BQP either. Given that random generability is not widely studied, we do not see an obvious reason a priori why it should be impossible for BQP functions; establishing this rigorously requires an argument of the type developed in our proofs.
>
> We also note that, subsequent to the submission of our work, the authors of [4] established hardness of random generability for the exact case of pairs (x,f(x)), under the assumption that PostBQP is not contained in PostBPP, which otherwise would imply a collapse of the polynomial hierarchy. Importantly, their proof applies only to the exact random generability setting of Def. 1. In contrast, when errors are allowed (Def. 2), their approach no longer applies, and one must employ different techniques, such as those used in our results.
>
> To avoid confusion, in the revised version of the main text, we clarify that hardness of generating random samples does not follow directly from hardness of computing in the revised Section 3.

---

> ### Author Response · Authors · 2025-11-20
>
> **The main claim appears in Theorem 5, which establishes classical hardness. But to demonstrate a classical–quantum separation one must also show an efficient quantum algorithm. In fact, even if f is produced by a quantum device, that does not imply a quantum-machine-learning (QML) algorithm can solve the problem. For example, if the dataset encodes ground-state properties that reveal complex topological order, QML algorithms might still struggle to recognize it. Thus the authors have only proven classical hardness; they have not demonstrated quantum efficiency. As a result, an exponential quantum advantage is not rigorously established.**
>
> We thank the reviewer for this thoughtful observation. Indeed, establishing classical hardness alone is not sufficient to demonstrate a learning separation. Our focus on classical hardness is precisely motivated by earlier works in the literature where quantum learning was shown to be possible, but the corresponding classical impossibility results concerned only evaluation [1,2], leaving open the possibility that identification might still be easy.
>
> However we do *also* prove quantum learnability for some cases. In particular Theorem 15 does provide an explicit example of an identification task exhibiting a quantum–classical separation. There, we consider a concept class of the type introduced by [2], for which a separation was previously shown only at the level of function evaluation. We then modify this construction using tools from error-correcting codes so that the concepts differ on a large fraction of inputs and hence satisfy our distinctness condition required by our theorems.
> Since the quantum learning algorithm of [2] already PAC-learns the target parameter alpha for this class, our classical hardness result for distinct concept classes implies that, for this identification task, an efficient quantum learner exists but no efficient classical identification algorithm (under our complexity assumptions). This therefore establishes an exponential quantum advantage.
>
> In the revised version, we clarified in the main text that this gives a rigorous example of quantum-classical separation for the identification task, with a comment before Corollary 1.
>
> **Can the authors give some examples of the average-case-smooth concept class, especially in the quantum computing setting?**
>
> We thank the reviewer for this question, which prompted us to think more about this and eventually led us to a new possibly interesting contribution.
>
> We first note that the construction in Appendix G.1 already yields a concept class in which every pair of concepts satisfies Eq. (5) with a constant lower bound on the distance. Nevertheless, it is indeed interesting to have an example where the function distance more directly reflects the distance between parameters alpha’s. We present such a construction below.
>
> In particular, we have derived a simple example of a concept class of functions implementable on a quantum computer that satisfies our definition of average-case-smooth. We outline the construction here, and we have included the full detailed example of this average-case-smooth concept class in the revised version of the paper.
>
> The idea is to consider the concept class F={f^alpha:{0,1}^n -> {0,1} |  alpha $\in${0,1}^m } with f^alpha(x) = alpha_i, where i=int(x[1:log(m)]). In other words, the function f^alpha(x) outputs the i-th bit of the vector alpha, where i is the integer described by the first log(m) bit of x. Then it holds that (see the revised Appendix F2 for a full derivation):
>
> E_{x}|f^alpha(x)-f^alpha’(x)|~d(alpha,alpha’)
>
> where d(alpha,alpha’) is the Hamming distance between the bitstrings alpha and alpha’. Therefore, for the concept class F, the average-case-smooth condition is satisfied. Next, notice that the functions f^alpha can be easily implemented on a quantum circuit by preparing the initial state as |psi_alpha>= |alpha_1>...|alpha_m> and measuring the Z_i(x).
> This provides an example of a concept class satisfying the average-case-smooth condition, with the functions f^\alpha being evidently classically tractable. Nevertheless, by coupling them with a BQP function as we did in the proof of Theorem 15, we can construct a concept class of BQP functions that still respects the average-case-smooth condition, and to which our hardness result then applies.
>
>
> Finally, we expect that similar behavior should hold more generally for natural parametrized quantum models at least when one restricts the parameter space appropriately. For common parametrizations (e.g., via Hamiltonians of PQCs), very different parameter choices alpha can “accidentally” agree on many inputs, for instance due to periodicity or symmetries. By imposing suitable restrictions on the parameter domain, it is then plausible that one can recover average-case-smoothness more generally. We leave a detailed exploration of this direction for future work.

---

> > ### Author Response · Authors · 2025-11-20
> >
> > **Regarding Theorem 5: why does L, unif \in BPP^{3NP} lead to a contradiction? To the best of my knowledge, the relationship between these two classes is not known.**
> >
> > We thank the reviewer for the question. In short, the reviewer is right, we are not claiming a formal contradiction in a technical sense. In particular, if the reviewer is referring to our assumption in the exact case, i.e. BQP $\not\in$ BPP^NP^NP^NP, we just show that the existence of an efficient classical learner for our task would imply the inclusion BQP $\in$BPP^NP^NP^NP (if instead the reviewer is referring to the heuristic version of this assumption, we discuss that case below). We then treat this inclusion as a complexity-theoretic assumption that is widely believed to be false, in close analogy to classical hardness-of-learning results where one shows, for example, that an efficient learner would imply that the discrete logarithm problem lies in P [3]. In those cases, the conclusion “discrete log $\in$ P” is not known to be false, but is considered highly implausible based on cryptographic evidence.
> >
> > Similarly, here we do not claim that BQP$\not\in$ BPP^NP^NP^NP is established, nor that it is as widely believed as “discrete log $\not\in$ P”, but we do point to circumstantial evidence in this direction (e.g., oracle separations between BQP and PH, and collapse consequences for related sampling/functional classes). We do contend that the “majority bet” in the community (agreed not as strong a bet as for DLP not in P)  is that BQP is not contained in such low levels of the (heuristic) polynomial hierarchy, and our hardness results are conditional on that standard type of assumption.
> >
> > The same argument holds in the case the reviewer is asking specifically about the assumption in the heuristic setting. Moreover, as also explained in Appendix C, our classical hardness results can be based on the exact condition BQP $\not\in$ BPP^NP^NP^NP (without heuristics) in the setting where learnability is required for every distribution, rather than the distribution-specific notion of Def. 3.
> >
> > **Could the authors provide an efficient quantum algorithm for learning the classically hard cases indicated in Theorem 5?**
> >
> > We thank the reviewer for this question. Indeed, an example of an efficient quantum algorithm for learning a classically hard identification task is provided in Appendix G1. There, we consider the setting of [2], where the proposed quantum algorithm successfully PAC-learns the target parameter alpha, and we show that the corresponding identification task is classically intractable. Thus, this example provides an efficient quantum algorithm for a classically hard identification problem covered by Theorem 5.
> > In the revised version, we make this link explicit in the main text (immediately before Corollary 1).
> >
> > Finally, as discussed above, we would like to emphasize once more that the main focus of our work is to establish classical hardness of the identification task, rather than to design new quantum learning algorithms.
> >
> > [1]: Gyurik, Casper, and Vedran Dunjko. "Exponential separations between classical and quantum learners." arXiv preprint arXiv:2306.16028 (2023).
> >
> > [2]: Molteni, Riccardo, Casper Gyurik, and Vedran Dunjko. "Exponential quantum advantages in learning quantum observables from classical data." arXiv preprint arXiv:2405.02027 (2024).
> >
> > [3]: Liu, Yunchao, Srinivasan Arunachalam, and Kristan Temme. "A rigorous and robust quantum speed-up in supervised machine learning." Nature Physics 17.9 (2021): 1013-1017.
> >
> > [4]: Huang, Hsin-Yuan, et al. "Generative quantum advantage for classical and quantum problems." arXiv preprint arXiv:2509.09033 (2025).
> >
> > [5]: Arrighi, Pablo, and Louis Salvail. "Blind quantum computation." International Journal of Quantum Information 4.05 (2006): 883-898.

---

### Official Review · Reviewer_3GcN · 2025-10-26

**Soundness:** 3
**Presentation:** 3
**Contribution:** 3
**Rating:** 8
**Confidence:** 3

**Summary:**

This paper proposes a new regime of quantum machine learning that focuses on potential advantages in solely the learning/training step while foregoing the testing steps. The authors analyze the hardness of the learning task itself with theoretical learning frameworks (proper PAC learning) to hypotheses in complexity theory that are believed to be true (BQP not contained in the second level of the polynomial hierarchy). Their main result shows that there exists a broad class of identification tasks of quantum functions that are quantumly-easy and classically hard.

**Strengths:**

1. This paper shows that quantum-hard functions are not generally classically generatable, and certain quantum generated data may only be learned by quantum learners.
2. Formal connections of learning theory and complexity theory regarding this task are constructed in this paper.

**Weaknesses:**

1. While the existence of separations are shown, to my understanding, the paper does not provide a potential path to the construction of a quantum learner that can provide such learnability results.
2. Minor typo in last line of Appendix A.3: $\mathtt{HeurBPP}^{\tt NP}$

**Questions:**

1. How applicable are the main theorems to existing QML problems? The paper shows that Hamiltonian learning does not apply, but how does the results apply to problems like state/process tomography, circuit learning or (quantum-assisted) circuit compiling?
2. Does the paper provide any implications on QAOA-type algorithms, which has been shown to be universal?

---

> ### Author Response · Authors · 2025-11-20
>
> We thank the reviewer for the positive evaluation of our work, here we address the points raised by the reviewer.
>
>
> **While the existence of separations are shown, to my understanding, the paper does not provide a potential path to the construction of a quantum learner that can provide such learnability results.**
>
> We thank the reviewer for this comment. Indeed the reviewer is correct that the main focus of our work is on establishing classical hardness. This focus is motivated by several known examples of quantum learning problems where a quantum computer can learn while a classical one cannot, but only when efficient evaluation of the learned hypothesis is required; it was open whether such separations persist when only identification is demanded [1,2].
>
> In this paper, however, we also show that for some of these cases one can obtain both quantum learnability and classical non-learnability in the sense of identification. Namely we can combine the classical non-learnability of this work with the quantum learning algorithms developed in [2,8] to obtain a separation, and we make it explicit in Corollary 1 and Appendix G1. In particular, in Appendix G.1, we construct an identification task for which there exists an explicit quantum algorithm that successfully performs identification and to which our classical hardness results apply. This yields a formal quantum–classical separation.
>
> In the revised version of the paper, we now comment on this explicitly before Corollary 1 in the main text, and we also mention it in the introduction so that this aspect is not overlooked.
>
> **Minor typo in last line of Appendix A.3**
>
> We thank the reviewer for the spotted typo, we correct it.
>
> **How applicable are the main theorems to existing QML problems? The paper shows that Hamiltonian learning does not apply, but how does the results apply to problems like state/process tomography, circuit learning or (quantum-assisted) circuit compiling?**
>
> We thank the reviewer for this question, which allows us to clarify the scope of our results.
>
> Our hardness theorems apply to learning problems that can be cast as identification tasks over concept classes of BQP functions, and that satisfy our average-case smoothness or distinctness conditions. A first prerequisite, therefore, is that the target problem involves such BQP functions (or, more precisely, functions which are not in the 4th level of the PH). We provide an explicit example of this situation in Appendix G.1, where we construct an identification task for which there exists a concrete quantum algorithm that solves it, and to which our classical hardness results apply.
>
> More broadly, identification-type tasks naturally arise in several settings, such as:
>
> - Identifying a proxy for the order parameter, in case the observable associated with it cannot be implemented.
> - Learning time dynamics with a fixed observable.
> - Device characterization in case of the presence of noise in the measurement apparatus.
> - Studying quantum quenches, which in turn could help us discover and understand new phases of matter, such as topological insulators and superconductors.
> - Quantum measurement tomography [3,4]
>
> In addition, the BQP functions covered by our framework have been shown to arise in problems involving quantum processes, for example:
>
> - Estimating the scattering probabilities in massive field theories [5]
> - Simulation of topological quantum field theories [6]
> - Electronic structure problem for quantum chemistry [7], with application in analyzing quantum chemical reactions
> - Characterization of exotic materials in critical phases,with applications in materials discovery.
>
> Concerning the specific tasks mentioned by the reviewer (state/process tomography, circuit learning, and quantum-assisted circuit compilation), our results would become applicable when the target states, channels, or circuits implement BQP processes of the type above, and when the corresponding learning problem can be modeled as an identification task over an average-case smooth or distinct concept class.
> We also stress that our notion of identification (Def. 8) is weaker than full tomography: we require recovery of a description that is PAC-correct with respect to a distribution, rather than exact reconstruction. For instance, in the state tomography setting, one could view an identification task where the goal is to reproduce expectation values of certain observables up to small error, without necessarily recovering the exact underlying state.

---

> > ### Author Response · Authors · 2025-11-20
> >
> > **Does the paper provide any implications on QAOA-type algorithms, which has been shown to be universal?**
> >
> > We thank the reviewer for raising this connection to QAOA-type algorithms. It is known that circuits of QAOA structure (interleaving fixed ZZ-layers with X-layers but with programmable angles) can be universal. This implies that one can construct families of BQP functions mapping, for example, bitstring inputs to expectation values, where the functions are parametrized by the QAOA angles. For such families, provided the average-case-smoothness conditions are satisfied, our results would imply that classical identification is impossible under our complexity-theoretic assumptions, while a quantum learner could in principle identify the underlying parameters.
> > We agree that this is an interesting direction. We have thought about this proposal and are still unsure how to meaningfully connect it to the standard use of QAOA for optimization tasks such as finding ground states. A key difference is that in our framework the target is a labeling function with an explicit input, whereas QAOA in optimization is typically used without an explicit input register and is not directly framed as realizing such a labeling function, but rather as preparing low-energy states for a given cost Hamiltonian. It is conceivable that our results might constrain more advanced “meta-learning” scenarios with QAOA, for instance, learning QAOA architectures or angles to use to optimize their use for specific families of optimization instances, but this would require careful additional constructions that lie beyond the scope of the present work.
> >
> > [1]: Gyurik, Casper, and Vedran Dunjko. "Exponential separations between classical and quantum learners." arXiv preprint arXiv:2306.16028 (2023).
> >
> > [2]: Molteni, Riccardo, Casper Gyurik, and Vedran Dunjko. "Exponential quantum advantages in learning quantum observables from classical data." arXiv preprint arXiv:2405.02027 (2024).
> >
> > [3]:A. Luis, et al., “Complete characterization of arbitrary quantum measurement processes”, Physical review letters (1999)
> >
> > [4]: J. S. Lundeen,et al., “Tomography of quantum detectors”, Nature Physics (2009)
> >
> > [5]: Jordan, et al., “Bqp-completeness of scattering in scalar quantum field theory,” Quantum (2018)
> >
> > [6]: Freedman, et al., “Simulation of topological field theories by quantum computers,” Communications in Mathematical Physics (2002)
> >
> > [7]: O’Gorman, et al., “Electronic structure in a fixed basis is qma-complete”, arXiv
> >
> > [8]: Barthe, Alice, et al. "Quantum Advantage in Learning Quantum Dynamics via Fourier coefficient extraction." arXiv preprint arXiv:2506.17089 (2025)

---

> > > ### Comment · Reviewer_3GcN · 2025-11-27
> > >
> > > I thank the authors for their response. I will be retaining my positive evaluation of the paper.

---

### Official Review · Reviewer_yvqS · 2025-11-02

**Soundness:** 2
**Presentation:** 1
**Contribution:** 2
**Rating:** 4
**Confidence:** 3

**Summary:**

A central question in quantum machine learning is to determine which learning problems admit a genuine quantum advantage. In recent years, there have been rigorous proofs of quantum advantage for learning properties of functions, typically relying on the classical intractability of evaluating the labeling function. In this paper, the authors instead focus on the identification task—the problem of identifying the correct label from a dataset—which has not been thoroughly explored in prior work.

The authors provide the first rigorous proof of a quantum advantage for identification learning under well-founded complexity-theoretic assumptions. In particular, they show that if BQP is not contained in HeurBPP^NP^NP^NP, then there exists a family of quantumly-samplable functions—computable by a quantum polynomial-time algorithm—that cannot be identified by any classical efficient algorithm. The proof proceeds by contradiction: assuming the existence of a classical efficient identification algorithm, the authors construct an evaluation algorithm in HeurBPP^NP^NP^NP, contradicting the initial assumption.

**Strengths:**

The authors introduce a new task in quantum machine learning: identifying a function within a given concept class. At first glance, this appears easier than standard learning tasks, which typically require the learner not only to infer the target function but also to evaluate it on unseen data points. In contrast, here a classical learner need only output a description of the unknown function. Surprisingly, the authors show that even this seemingly simpler task remains hard for classical learners when the function is quantum-computable—assuming that BQP is not contained in the fourth level of the polynomial hierarchy.

**Weaknesses:**

This work lacks sufficient motivation from quantum physics or quantum information. Although the paper is titled quantum machine learning advantages, the main result is proved entirely using classical complexity-theoretic arguments. Specifically, the hardness argument proceeds by constructing a classical algorithm that evaluates the function in HeurBPP^NP^NP^NP, assuming the function can be efficiently identified classically, and then combining this with the heuristic separation result of Ran and Raz between PH and BQP. The authors include several physics-motivated examples of identification hardness in Appendix G, but the arguments are relatively straightforward: Appendix G.1 closely follows the approach of Molteni et al., while Appendices G.2 and G.3 present heuristic arguments that are not clearly explained or integrated into the main text.

In addition, the treatment of complexity-theoretic assumptions in the paper lacks precision. For example, in Theorem 15, the authors claim the existence of a quantum-computable learning task that is not classically identifiable by any approximately correct algorithm. The proof relies on showing that an efficient classical identification algorithm would imply BQP lies in HeurBPP^NP^NP^NP. However, the theorem’s statement does not explicitly mention any complexity-theoretic assumption, leaving the claim misleading in the standard (unrelativized) world where no such containment is known. Furthermore, while the authors refer to oracle separations between BQP and PH, their proof relies instead on the assumption BQP not in HeurBPP^NP^NP^NP, which is a heuristic class not known to be contained in PH and about which relatively little is understood. Clarifying these assumptions and aligning the theorem statements accordingly would significantly improve the precision and correctness of the presentation.

**Questions:**

Do we have any known separation between HeurBPP^NP^NP^NP and BQP, or we just believe in this separation intuitively?

---

> ### Author Response · Authors · 2025-11-20
>
> **This work lacks sufficient motivation from quantum physics or quantum information... , and then combining this with the heuristic separation result of Ran and Raz between PH and BQP**
>
> We thank the reviewer for this comment. Our understanding is that the main concern is that the core hardness argument is phrased entirely in terms of classical complexity classes, and therefore might seem insufficiently “quantum”. We respectfully disagree that there is no quantum motivation for three reasons.
>
> First, any rigorous demonstration of a quantum advantage must include a classical no-go result, and in practice this classical lower-bound analysis is often the technically most challenging part. The main objective of the paper is precisely to show that the identification task is classically hard when the concept class contains quantum (BQP) functions, under the assumption that BQP is not contained in PH. To obtain such a hardness result, one necessarily has to argue that, if a classical learner could efficiently solve the identification task, then one could construct a classical algorithm within the polynomial hierarchy that efficiently evaluates the underlying BQP function. As correctly noted by the reviewer, this is exactly what we do. In this sense, while the proof technique involves the construction of a classical algorithm, the hardness result itself is intrinsically quantum, as it crucially hinges on the presence of BQP functions in the concept class.
>
> Second, we highlight that several genuinely quantum learning problems, long conjectured to exhibit quantum advantage, can naturally be cast as identification tasks, so our results provide at least a partial answer in these settings. Examples include:
>
> - Identifying a proxy for the order parameter, in case the observable associated with it cannot be implemented.
> - Learning time dynamics with a fixed observable.
> - Device characterization in case of the presence of noise in the measurement apparatus.
> - Studying quantum quenches, which in turn could help us discover and understand new phases of matter, such as topological insulators and superconductors.
> - Quantum measurement tomography [1,2]
>
> Third, we also establish quantum learnability in Appendix G.1, where we use the quantum algorithm from [3] to solve an example of an identification task that is classically hard, thereby obtaining a quantum–classical learnability separation. In the revised version of the paper, we now state this explicitly already in the Introduction.
>
>
>
> **The authors include several physics-motivated examples of identification hardness in Appendix G, but the arguments are relatively straightforward: Appendix G.1..., while Appendices G.2 and G.3 present heuristic arguments that are not clearly explained or integrated into the main text.**
>
> We thank the reviewer for this comment and for acknowledging our effort to present physics-motivated examples. Before addressing the specific points about Appendix G, we would like to stress that the main motivation of our work is exactly the existence of prior results showing quantum learnability in natural settings where identification (recovering a description of the target) is a more appropriate requirement than mere evaluation, yet no classical no-go results were known for identification in those cases. Our goal in this paper is to fill this big gap in the logical picture behind proposed quantum advantages for natural learning problems. Hence, the arguments on the quantum learnability side are naturally closely aligned with those in [3]. By contrast, the arguments for classical non-learnability are substantially different and novel, as we work with a weaker and less demanding notion of learning than [3].
>
> In particular, regarding Appendix G.1, we agree that the quantum learnability aspect closely follows the approach of [3]. However, the classical hardness of identification for this concept class was not established in [3] and is derived exclusively in the present work. In particular, we design a specific family of observables, based on Reed–Solomon codes, precisely so that our general hardness framework applies to the concept class considered in [3], thereby yielding a new identification-hardness result that was not previously known.
> Regarding Appendices G.2 and G.3, in the revised version we have made the heuristic examples more explicit and better integrated into the main text. In particular, in the Discussion section we now provide an overview on the connection between our work and the task of Hamiltonian learning.
>
>
> [1]:A. Luis, et al., “Complete characterization of arbitrary quantum measurement processes”, Physical review letters (1999)
>
> [2]: J. S. Lundeen,et al., “Tomography of quantum detectors”, Nature Physics (2009)
>
> [3]: Molteni, Riccardo, Casper Gyurik, and Vedran Dunjko. "Exponential quantum advantages in learning quantum observables from classical data." arXiv preprint arXiv:2405.02027 (2024).

---

> > ### Author Response · Authors · 2025-11-20
> >
> > **In addition, the treatment of complexity-theoretic assumptions in the paper lacks precision. For example, in Theorem 15, the authors claim the existence of a quantum-computable learning task that is not classically identifiable by any approximately correct algorithm. The proof relies on showing that an efficient classical identification algorithm would imply BQP lies in HeurBPP^NP^NP^NP. However, the theorem’s statement does not explicitly mention any complexity-theoretic assumption, leaving the claim misleading in the standard (unrelativized) world where no such containment is known.**
> >
> >   We thank the reviewer for drawing attention to this aspect of our work. Indeed, the classical hardness part of Theorem 15 relies on the assumption BQP not in HeurBPP^NP^NP^NP. In the revised version, we have updated the statement of Theorem 15 to make this assumption explicit.
> >
> > If there are any other parts of the paper that appear unclear or imprecise, we would be very grateful if the reviewer could point them out so that we can revise and clarify them.
> >
> >
> > **Furthermore, while the authors refer to oracle separations between BQP and PH, their proof relies instead on the assumption BQP not in HeurBPP^NP^NP^NP, which is a heuristic class not known to be contained in PH and about which relatively little is understood. Clarifying these assumptions and aligning the theorem statements accordingly would significantly improve the precision and correctness of the presentation.**
> >
> > The review is right that our theorems and proofs are presented under the assumption BQP not in HeurBPP^NP^NP^NP. However, as discussed in Appendix C, one can obtain classical hardness results under the exact assumption BQP not in BPP^NP^NP^NP, in the setting where one requires learnability with respect to every distribution, rather than the distribution-specific notion of Def. 3. We clarify this aspect in the revised version of the Introduction.
> >
> > Furthermore, although the assumption in the heuristic case is indeed a stronger assumption than working in the exact setting, we consider it not much more unlikely than for the non-heuristic case (see the answer to the next question).
> >
> > Finally, we would like to clarify that we do not, of course, provide new proofs that $BQP\not\subseteq PH$. Rather, we report arguments that are commonly taken as evidence for this conjecture, including oracle separations and conditional separations (based on polynomial-hierarchy collapse) for both sampling and functional variants.
> >
> >
> >
> >
> > **Do we have any known separation between HeurBPP^NP^NP^NP and BQP, or we just believe in this separation intuitively?**
> >
> > We thank the reviewer for this question. Firstly, we would like to remind that one can obtain hardness for the identification task under the exact (non-heuristic) separation BQP $\not\subseteq$ BPP^NP^NP^NP by requiring learnability with respect to every distribution, rather than distribution-specific learnability. We discuss this in Appendix C.
> >
> > To the best of our knowledge, there is currently no formal separation between HeurBPP^NP^NP^NP and BQP, just as there is no known unrelativized separation between BQP and PH in the exact setting. Nonetheless, we view the conjecture BQP$\not\subset$ HeurBPP^NP^NP^NP as a natural and reasonable complexity assumption, as the existence of a not pathological distribution for which BQP would be in HeurBPP^NP^NP^NP would represent a major result in complexity theory. In particular, there are not known examples where passing to heuristic (average-case) variants lowers the known upper bounds for quantum classes into classical ones on natural, non-contrived distributions.
> >
> > Finally, we do agree that this is a stronger assumption, but, as noted above, it is not needed in the setting where learnability is required with respect to all distributions.

---

### Official Review · Reviewer_nGtx · 2025-11-03

**Soundness:** 3
**Presentation:** 3
**Contribution:** 3
**Rating:** 8
**Confidence:** 4

**Summary:**

This paper studies quantum-classical separation for the identification problem. Learning advantages for quantum functions are shown under complexity-theoretic assumptions. This is a problem in the context of PAC in learning theory, where the objective of the learner is to identify if a given labeled dataset is predictable with any member of a concept class of functions.

The authors prove a series of interesting results, for instance, showing that quantum functions are not randomly generable unless BQP is in the second level of the polynomial hierarchy.

**Strengths:**

Overall, this is a solid paper studying an interesting problem regarding the quantum-classical separation under a complexity theoretic framework.
The authors establish a set of theoretical conditions under which such separation arises in learning problems, offering a solid foundation for understanding the boundaries between classical and quantum computational capabilities. In addition, the paper explores a few relevant applications where quantum-classical separation may manifest, providing practical contexts that highlight the significance of the results.

**Weaknesses:**

While the paper presents strong theoretical contributions, its accessibility may be limited due to its specialized focus and technical depth. The exposition appears tailored primarily for experts in quantum computing and theoretical learning, which could pose challenges for the broader ICLR audience. To enhance its impact and reach, the paper would benefit from revisions that clarify key concepts, provide more intuitive explanations, and better contextualize the results within mainstream machine learning frameworks.

**Questions:**

Can elaborate on how your results can contribute to classical quantum separation in the context of the PAC learning framework?

---

> ### Author Response · Authors · 2025-11-20
>
> We sincerely thank the reviewer for the positive comments about our work and the suggestions provided to further increase the quality of our paper.
>
> **While the paper presents strong theoretical contributions, its accessibility may be limited due to its specialized focus and technical depth. The exposition appears tailored primarily for experts in quantum computing and theoretical learning, which could pose challenges for the broader ICLR audience. To enhance its impact and reach, the paper would benefit from revisions that clarify key concepts, provide more intuitive explanations, and better contextualize the results within mainstream machine learning frameworks.|**
>
> We thank the reviewer for this suggestion. To improve the accessibility of our results, in the revised manuscript we now give a simpler, informal explanation of the identification task and BQP functions in the introduction, and we have added one-sentence, plain-language summaries of our main theorems (Theorems 1–2, 5, and 15).
>
>
> **Can elaborate on how your results can contribute to classical quantum separation in the context of the PAC learning framework?**
>
> We thank the reviewer for their question. In the standard PAC learning framework (Def. 3 in our paper), the learner must output a hypothesis function that approximates the target labeling function. In case we allow only classically tractable hypothesis functions, a learning separation can be achieved by just requiring that the labeling function does not admit a polynomial-size classical circuit, since this already rules out the existence of an efficient classical hypothesis (see for example [1], [2]).
> However, the goal of this work is to investigate whether a separation can already appear at an earlier stage, where the learner only has to select a correct hypothesis, without requiring this hypothesis to be efficiently evaluable by a classical algorithm. In particular, we allow the classical learner to output a description of a quantum hypothesis function. [1] showed that if no restriction is imposed on the hypothesis class, then a classical learner can always produce a good hypothesis by outputting the circuit of the quantum learner with the training data hardwired. Hence, to obtain a genuine separation for the identification task, it is necessary to restrict the hypothesis set.
>
> The proper PAC setting gives a natural framework where the hypothesis class is constrained to be identical to the concept class.  As we explain in Appendix H, our notion of an identification task is very close to proper PAC learning, but with the additional requirement that the learner also satisfies the two extra properties in Def. 8. We further show in Appendix H that there are natural concept classes for which a standard PAC learner already fulfills these additional conditions.
> Viewed from the perspective of classical learning theory, our work thus provides a first result on quantum–classical separations for learning concept classes of quantum functions in the setting of proper PAC learning.
>
> In the revised version of the Discussion section, we have added a remark explicitly highlighting the contribution of our results in the context of the classical PAC framework.
>
>
>
>
> [1]: Gyurik, Casper, and Vedran Dunjko. "Exponential separations between classical and quantum learners." arXiv preprint arXiv:2306.16028 (2023).
>
> [2]: Molteni, Riccardo, Casper Gyurik, and Vedran Dunjko. "Exponential quantum advantages in learning quantum observables from classical data." arXiv preprint arXiv:2405.02027 (2024).

---

### Author Response · Authors · 2025-12-03
**Summary of revisions and clarifications  (1/2)**

Dear Area Chair,

We thank all reviewers for their careful and constructive feedback, which helped us strengthen the paper and clarify several important aspects.

During the rebuttal, we addressed each concern in detail. The main issues that led to lower scores (for reviewers yvqS and Wcsz) were primarily due to insufficient clarity on our side, and we have revised the manuscript accordingly.

For reviewer Wcsz, the central concern was that our paper establishes classical hardness but does not provide a corresponding efficient quantum learner, and therefore would not constitute a proper learning separation. However, our paper does establish a proper learning separation as we also provide examples of quantum learnability. Concretely, in Appendix G.1 we present an identification task to which our hardness results apply and for which an explicit quantum learning algorithm exists, yielding a genuine learning separation. To make this clear to readers, we added an explicit discussion in the main text immediately before Corollary 1. Furthermore, we also explain that this connection may not have been sufficiently emphasized in the original version, since the paper primarily focuses on classical lower bounds for identification, while some quantum learnability results were already known, though we still needed new constructions to adapt them to our setting.

The reviewer’s second concern was about the relevance of our results on random generability. The referee raised the relevance of this result, as in their view it was expected. However, we explain how this is not the case. In particular, we clarified that random generability is strictly weaker than evaluation: there are well-known examples (e.g., from cryptography) of functions that are not classically efficiently evaluable but for which labeled samples can still be efficiently generated. Our contribution is to show that this phenomenon does not extend to the quantum functions in our setting: under our assumptions, even random generability of such quantum functions is classically hard. The reviewer’s confusion may have arisen from considering the ability to evaluate a function as equivalent to the ability to generate random labeled samples from it. We now clarify this distinction in Section 3.

We highlight that reviewer Wcsz indicated willingness to revise their score (original grade: 4) if these concerns were resolved, and we believe the clarifications above address their objections directly and warrant an improved evaluation of the work.


For reviewer yvqS, the main concern was that the motivation appeared insufficiently “quantum,” since the proof strategy proceeds via classical complexity-theoretic impossibility statements. We clarified that such an approach is necessary to prove classical hardness, and that our results are intrinsically about quantum concept classes: they apply specifically to BQP functions, which arise from purely quantum processes. In the revision, we better contextualize these connections and integrate additional representative examples into the main text.

The reviewer also raised concerns that some theorem statements did not clearly specify the assumptions being used, and that it was sometimes unclear whether we were working in the average-case (heuristic) or exact setting. In the revised version, we state the required assumptions explicitly in each relevant theorem, and we provide additional discussion motivating them. We also clarify that, in the setting where learnability is required with respect to every distribution, our hardness results can be based on the exact (non-heuristic) assumption BQP  not in BPP^NP^NP^NP, removing the need to rely on the heuristic formulation.

We highlight that the reviewer yvqS stated that making the assumptions explicit and aligning the theorem statements accordingly would substantially improve the precision and correctness of the presentation, which seems to be a main reason for their lower score (original grade: 4). As described above, we have now made these changes in the revised version, and we therefore hope this will support a higher evaluation of our work.

We note that the other reviewers (3GcN and nGtx) were already positive about the paper (original score: 8). Reviewer 3GcN maintained this positive assessment after our response. Reviewer nGtx suggested revisions to clarify key concepts, provide more intuition, and better situate our results within mainstream machine learning frameworks to further broaden the paper’s reach and impact. We have addressed these suggestions by adding several clarifications and additional intuition in the revised manuscript.

---

> ### Author Response · Authors · 2025-12-03
> **Summary of revisions and clarifications (2/2)**
>
> Finally, we would like to highlight a new result that emerged during the discussion period. While it is only indirectly related to the reviewers’ concerns, it strengthens the paper. In particular, reviewer Wcsz asked whether we could provide an explicit example of a concept class of quantum functions satisfying our average-case-smoothness condition, and hence falling within the scope of our hardness theorems. In response, we constructed such a concept class and proved that it satisfies average-case-smoothness, thereby providing a new concrete example where the identification task is classically hard under our assumptions.
>
>
> We hope this summary can be of help to the Area Chair in assessing how we have addressed all reviewer comments and incorporated the corresponding changes into the revised manuscript. We believe these revisions strengthen the contribution and resolve the remaining concerns, leaving the paper in a strong position.
>
>  We are very grateful to the reviewers and to the Area Chair for their time and effort.

---

### Meta-Review · Area_Chair_Dyop · 2025-12-26

**Summary:**

The submission studies quantum–classical separations for identification learning of quantum functions, where the learner must output a description of the target labeling function rather than merely evaluate it. Under complexity-theoretic assumptions (e.g., BQP not contained in low levels of the polynomial hierarchy), the authors show that for broad classes of BQP-induced concept classes, no efficient classical identification algorithm exists. They further prove the hardness of random generability for such quantum functions and, via an instantiation based on prior quantum learners, obtain an explicit example where identification is quantumly easy but classically hard.

Reviewers nGtx and 3GcN were positive, highlighting the conceptual clarity of the identification task and the technical strength of the hardness results, while noting limited accessibility for a general ML audience. Reviewers yvqS and Wcsz raised concerns about the ``quantumness'' of the motivation, the precision and explicitness of the complexity assumptions, and whether a genuine learning separation (including a quantum learner) was established. In their response and revision, the authors clarified the assumptions in each theorem, explicitly connected their setting to proper PAC learning, emphasized the concrete quantum learning separation via Theorem 15 and Appendix G.1, and provided an explicit average-case-smooth concept class, addressing the main technical concerns.

Taking into account the overall positive assessments, the detailed and largely convincing rebuttal, and my own reading of the technical development, I find the work to be a solid and timely theoretical contribution to quantum learning theory. The remaining weaknesses are primarily about accessibility and the strength of the assumed complexity separations, which are standard for this type of result. I therefore recommend acceptance (poster).

**Reviewer Concerns:**

The rebuttal successfully clarified several key points. In particular, the authors made the complexity-theoretic assumptions more explicit theorem-by-theorem, clarified the connection to proper PAC-style identification learning, and explained that Theorem 5 by itself is a classical hardness result while the actual quantum–classical separation is obtained later via Theorem 15 and Appendix G.1 using an explicit quantum learner from prior work. They also provided a concrete average-case–smooth concept class and improved the explanation of why random generatability fails in the quantum setting, which addresses the main technical doubts raised by the more skeptical reviewers.

Some concerns remain only partially resolved. First, the work remains quite dense and difficult to access for a general ML audience; the high technical overhead and compact proofs will still limit readability despite the added clarifications. Second, the results are inherently conditional on relatively strong complexity assumptions, and while these are now clearly stated, their strength and specificity may still leave some readers unconvinced about the robustness of the claimed advantages. Third, the practical relevance for near-term QML remains indirect.

**Reviewer Scores:**

Based on the rebuttal and the ensuing discussion, the reviewers who were initially positive (e.g., nGtx and 3GcN) would likely have maintained their scores, as the clarifications reinforced their view of the work as a solid theoretical contribution. For the more skeptical reviewers (yvqS and Wcsz), the rebuttal addressed their main technical concerns by clearly separating classical hardness from the quantum learning separation and by making the underlying complexity assumptions explicit. As a result, these reviewers might have modestly increased their scores or, at least, softened their reservations.

---

### Decision · Program_Chairs · 2026-01-26

Accept (Poster)